

# Biodegradation by bacteria in clouds:
# An underestimated sink for some organics in the atmospheric multiphase system

Amina Khaled[1], Minghui Zhang[1], Pierre Amato[1], Anne-Marie Delort[1], and Barbara Ervens[1]

[1]Université Clermont Auvergne, CNRS, SIGMA Clermont, Institut de Chimie de Clermont-Ferrand, 63000 Clermont-Ferrand, France

Corresponding author: Amina.Khaled@uca.fr

**Abstract.** Water-soluble organic compounds represent a significant fraction of total atmospheric carbon. The main oxidants towards them in the gas and aqueous phases are OH and $NO_3$ radicals. In addition to chemical solutes, a great variety of microorganisms (e.g. bacteria, viruses, fungi) has been identified in cloud water. Previous lab studies suggested that for some organics, biodegradation by bacteria in water is

comparable to their loss by chemical processes. We perform model sensitivity studies over large ranges of biological and chemical process parameters using a box model with a detailed atmospheric multiphase chemical mechanism and biodegradation processes to explore the importance of biodegradation of organics in the aqueous phase. Accounting for the fact that only a small number fraction of cloud droplets (~0.0001 – 0.001) contains active bacteria cells, we consider only a few bacteria-containing droplets in the model

cloud. We demonstrate that biodegradation might be most efficient for volatile organic compounds (VOC) with intermediate solubility (~$10^4 \leq K_{H(eff)}$ [M atm$^{-1}$] $\leq 10^6$, e.g., formic and acetic acids). This can be explained by the transport limitation due evaporation of organics from bacteria-free droplets to the gas phase, followed by the dissolution into bacteria-containing droplets. For non-volatile organics (NVOC), such as dicarboxylic acids, the upper limit of organic loss by biodegradation can be approximated by the

amount of organics dissolved in the bacteria-containing droplets (< 0.01%). We compare results from this detailed drop-resolved model to simplified model approaches, in which either (i) all cloud droplets are assumed to contain the same cell concentration (0.0001 – 0.001 cell droplet$^{-1}$) or (ii) only droplets with intact bacteria cells are considered in the cloud (liquid water content ~$10^{-11}$ vol/vol). Conclusions based on these approaches generally overestimate of the role of biodegradation, in particular, for highly soluble VOC. Our

model sensitivity studies suggest that current atmospheric multiphase chemistry models are incomplete for organics with intermediate solubility and high bacterial activity.



## 1.     Introduction

Clouds provide a medium for multiphase chemical reactions, in which chemical species from the aqueous, solid and gas phases are transformed and can affect significantly the transport and distribution of chemical species in the atmosphere (Lelieveld and Crutzen, 1991). The chemical composition of cloud water is a complex mixture containing a multitude of organic and inorganic species with a range of chemical and physical properties (e.g., reactivity, solubility, volatility). The organic fraction includes volatile (VOC) and non-volatile organic compounds (NVOC), such as aldehydes, mono- and dicarboxylic acids, organonitrogen and organosulfur compounds. VOC are dissolved from the gas phase to the aqueous phase; NVOC enter the aqueous phase  via nucleation scavenging of condensation nuclei (CCN) (Löflund et al., 2002) (Ervens, 2015). Water-soluble organic carbon (WSOC) constitutes a significant portion of the total atmospheric organic carbon mass, ranging from ~14% to ~64% depending on the sampling location (Decesari et al., 2000) (Gao et al., 2016)(Varga et al., 2001).

In addition to chemical solutes, cloud water contains microorganisms such as bacteria, yeast and fungi (Delort et al., 2010)(Hu et al., 2018). Typical concentrations of bacteria cells are on the order of $10^6$ to $10^8$ cells $L^{-1}$; fungi and yeast cell concentrations are usually lower (~$10^5$ to ~$10^7$ cells $L^{-1}$) (Amato et al., 2007b) (Sattier et al., 2001). The atmosphere is a stressful environment for microorganisms (low temperature, UV exposure, acidic pH, quick hydration/drying cycles) (Sattier et al., 2001) which might limit the survival time of cells in the atmosphere. Several studies have shown that bacteria can grow and be metabolically active in cloud droplets. Marker compounds such as adenosine 5'-triphosphate (ATP) (Amato et al., 2007c) , rRNA (Krumins et al., 2014) or mRNA (Amato et al., 2019) have been used to demonstrate metabolic activity in the atmosphere. Metabolic activity and cell generation of bacteria is likely restricted to the time cells spent in clouds due to the abundance of liquid water (Haddrell and Thomas, 2017) (Ervens and Amato, 2020); bacteria have been found to be dormant at lower relative humidity than in clouds (Kaprelyants and Kell, 1993).

The metabolic activity of bacterial strains identified in cloud water (e.g. *Pseudomonas, Sphingomonas*) has been investigated in lab studies, and it was shown that they can biodegrade organics (e.g., malonate, succinate, adipate, pimelate, formaldehyde, methanol, acetate, formate, phenol and catechol (Delort et al., 2010)(Vaïtilingom et al., 2010, 2011, 2013) (Amato et al., 2007a) (Ariya et al., 2002) (Husárová et al., 2011) (Fankhauser et al., 2019)(Jaber et al., 2020). Based on comparisons of experimentally derived biodegradation rates to chemical rates of oxidation reactions by radicals (e.g., OH, NO$_3$) in the aqueous phase, it was concluded that they might be similar under some conditions, and that, depending on the abundance and metabolic activity of bacteria strains, oxidation and biodegradation processes of organics may compete in clouds.

There are several estimates of WSOC loss by bacteria on a global scale: Sattier et al. (2001) estimated a sink





of 1- 10 Tg yr$^{-1}$, smaller than the estimate by Vaïtilingom et al (2013) (10-50 Tg yr$^{-1}$). However, the latter is likely an overestimate as complete respiration was implied, i.e. total conversion of organics into $CO_2$. More conservatively Ervens and Amato (2020) suggested a global WSOC loss of 8–11 Tg yr$^{-1}$, being comparable to that by chemical processes (8–20 Tg yr$^{-1}$). Similarly, Fankhauser et al. (2019) postulated that the role of biodegradation is likely small but they did not quantify the loss of different organics by bacteria. Current atmospheric multiphase chemistry models include chemical mechanisms of different complexity with up to thousands of chemical reactions describing the transformation of inorganic and organic compounds, e.g (Ervens et al., 2003a) (Mouchel-Vallon et al., 2017) (Tilgner et al., 2013) (Woo and McNeill, 2015). However, they do not include the biodegradation of organics by bacteria despite the available data sets discussed above.

Cloud chemistry models often assume initially identical composition of all cloud droplets. While this might be a reasonable assumption for the chemical droplet composition due to internally mixed CCN and the phase transfer from the gas phase into all droplets, it is not appropriate for the distribution of bacteria. Due to their small number fraction of the total CCN concentration (0.001 – 0.1 %, (e.g.,(Zhang et al., 2020) the fraction of cloud droplets that contain bacteria cells is small (< 0.001). Thus, to explore biodegradation of organics in the atmospheric multiphase system, a realistic distribution within cloud droplet population needs to be assumed.

The aim of our study is to identify conditions, under which biodegradation in clouds is significant in the atmosphere. Using a cloud multiphase box model, we explore the biological and chemical degradation of VOC and NVOC over large parameter ranges of bacterial cell concentrations, biodegradation activities, chemical rate constants and Henry's law constants of the organic substrates. We compare (1) the biodegradation rates in the aqueous phase to the chemical rates in both phases, and (2) the fraction of organics consumed by biodegradation to that by chemical processes. The results of our sensitivity studies elucidate, for which organics biodegradation competes with chemical processes. Our study will give guidance for future experimental and modeling studies to further complete atmospheric models in order to more comprehensively describe organic degradation in the atmosphere.

## 2. Methods

### 2.1 Description of the multiphase box model

We use a multiphase box model with detailed gas and aqueous phase chemistry (75 species, 44 gas phase reactions, 31 aqueous reactions (Ervens et al., 2008). The two phases are coupled by 26 phase transfer processes. Phase transfer is described kinetically based on the resistance model by Schwartz (1986).



In addition to the base chemical mechanism, we define one organic species 'Org' that undergoes chemical radical reactions in the gas and aqueous phases and biodegradation by bacteria only in the aqueous phase in a small subset of the droplets as shown in *Figure 1*.

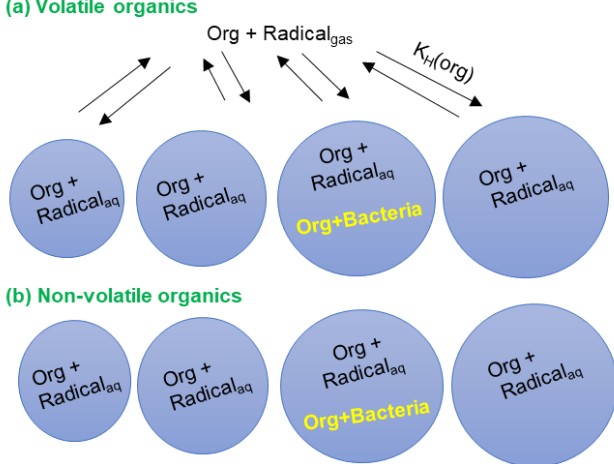

**Figure 1**: Schematic of the multiphase box model including chemical radical reactions in the gas and aqueous phases. Chemical reactions occur in both phases and in all droplets whereas biodegradation processes only occur in the aqueous phase in a small fraction of droplets, depending on the assumed bacteria cell concentration in clouds. (a) VOC that partition between the two phases (b) NVOC that get only processed in the aqueous phase.

We consider a polydisperse droplet population of 263 droplets $cm^{-3}$ in 11 size classes with drop diameters of 5 μm < $D_{droplet}$ < 20 μm and a total liquid water content of LWC = $6.8 \times 10^{-7}$ vol/vol. Only one droplet size class includes bacteria cells ($D_{droplet}$ = 20 μm; $N_{droplet}$ = cell concentration = 0.01 $cm(gas)^{-3}$). Thus, the cell concentration in the cloud water ($C_{cell}$= $1.5 \times 10^{7}$ cell $L(aq)^{-1}$) is similar to that found in ambient clouds and used in some lab experiments (Vaïtilingom et al., 2013). The model simulations are performed for 600s which corresponds approximately to the droplet lifetime during one cloud cycle (Ervens et al., 2004). We assume an initial mixing ratio of the organic compound of 1 ppb.

## 2.2 Kinetic data in the gas and aqueous phases

### 2.2.1 Chemical rates and Henry's law constants

*Table 1* includes chemical rate constants for radical (OH, $NO_3$) reactions in the aqueous and gas phases for organic compounds, for which lab data on their biodegradation rates are available (*Section 2.2.2*). This data covers ranges of $10^3 \leq k_{chemaq} [M^{-1} s^{-1}] \leq 10^{10}$ and $10^{-17} \leq k_{chemgas} [cm^3 s^{-1}] \leq 10^{-10}$, respectively, over which the model sensitivity studies in the following are performed.



**Table 1**. Literature data on chemical rate constants in the aqueous ($k_{OH,aq}$; $k_{NO3,aq}$) and gas phases ($k_{OH,gas}$; $k_{NO3,gas}$) , physical Henry's law constants ($K_H$) and effective Henry's law constants ($K_{Heff}$) for acids at pH = 3 and pH = 6 respectively.

| Organic compounds | $k_{OH,aq}$ (25°C) [$M^{-1}$ $s^{-1}$] | $k_{OH,gas}$ (25°C) [$cm^3$ $s^{-1}$ ] | $k_{NO3,aq}$ [ $M^{-1}s^{-1}$] | $k_{NO3,gas}$ [$cm^3$ $s^{-1}$ ] | $K_H$ [M $atm^{-1}$] | $pK_a$ | $K_{H,eff}$ [M $atm^{-1}$] at pH = 3, pH = 6 |
|---|---|---|---|---|---|---|---|
| **NVOC** | | | | | | | |
| Malonic acid | $1.6 \times 10^7$ (1) | | $5.1 \times 10^4$ (1) | | | | |
| Malonate monoanion | $3.6 \times 10^8$ (1) | | $5.6 \times 10^6$ (1) $2.3 \times 10^7$ (1) | | $1.23 \times 10^{10}$ (12) | 2.8 5.6 | $3.18 \times 10^{10}$ $6.84 \times 10^{13}$ |
| Malonate dianion | $8.0 \times 10^7$ (1) | | | | | | |
| Succinic acid | $3.8 \times 10^8$ (2) | | $5 \times 10^3$ (1) | | $3.5 \times 10^9$ (12) | 4.23 5.64 | $3.70 \times 10^{10}$ $6.81 \times 10^{11}$ |
| Succinate monoanion | $5 \times 10^8$ (1) | | $1.1 \times 10^7$ (1) $1.8 \times 10^7$ (1) | | | | |
| Succinate dianion | $5 \times 10^8$ (5) | | | | | | |
| **VOC** | | | | | | | |
| Acetic acid | $1.6 \times 10^7$ (3) | $6.6 \times 10^{-13}$ (6) | $1.3 \times 10^4$ (8) | - | $7 \times 10^3$ (13) | 4.75 | $7.1 \times 10^3$ $1.3 \times 10^5$ |
| Acetate | $8.5 \times 10^7$ (1) | | $2.3 \times 0^6$ (8) | | | | |
| Formic acid | $1.3 \times 10^8$ (3) | $1.3 \times 10^{-12}$ (7) | $3.3 \times 10^4$ (1) | - | $11 \times 10^3$ (13) | 3.76 | $1.2 \times 10^4$ $1.9 \times 10^6$ |
| Formate | $3.2 \times 10^9$ (1) | | $4.2 \times 10^7$ (1) | | | - | - |
| Formaldehyde | $6.1 \cdot 10^8$ (1) | $8.5 \times 10^{-12}$ (1) | $7.9 \times 10^5$ (11) | | $3 \times 10^3$ (14) | - | - |
| Catechol | $3.8 \cdot 10^8$ (4) | $1 \times 10^{-12}$ (5) | $8.4 \times 10^9$ (9) | | $8.3 \times 10^5$ (4) | - | - |
| Phenol | $1.9 \cdot 10^9$ (1) | $1 \times 10^{-12}$ (5) | $1.9 \times 10^9$ (1) | $5.8 \cdot 10^{-12}$ | $6.47 \times 10^2$ (4) | - | - |
| Methanol | $8.7 \cdot 10^8$ (1) | $7.7 \times 10^{-13}$ (1) | $2.07 \times 10^5$ (10) | - | $2.9 \times 10^2$ (12) | - | - |

(1)(Herrmann, 2003); (2) (Cabelli et al 1985); (3)(Vaïtilingom et al., 2010); (4) (Pillar et al., 2014) (5) (Ervens et al., 2003c) ;(6)(Butkovskaya et al., 2004); (7)(Anglada, 2004); (8)(Exner et al., 1994); (9)(Hoffmann et al., 2018); (10) (Mezyk et al., 2017); (11)(Gaillard De Sémainville et al., 2007); (12) (Sander, 2015); (13) (Johnson et al., 1996); (14)(Allou et al., 2011); (-) : non relevant as no aqueous phase reaction considered.

In order to generalize our results for different radical concentrations, we present in terms of chemical rates $R_{chem}$ [$s^{-1}$] i.e. as the products of the rate constants $k_{chemaq}$ and $k_{chemgas}$ and radical concentrations:

$$R_{chemaq}[s^{-1}] = \frac{-d(org)_{radical}}{dt} = k_{chemaq}[ L\ mol^{-1}\ s^{-1}] \times [radical]_{aq}\ [mol\ L^{-1}] \qquad \text{Eq- 1}$$

$$R_{chemgas}[s^{-1}] = \frac{-d(org)_{radical}}{dt} = k_{chemgas}[cm^3\ s^{-1}] \times [radical]_{gas}\ [cm^{-3}] \qquad \text{Eq- 2}$$





Typical radical concentrations are on the order of $10^{-15}$ mol $L^{-1}$ for OH and $NO_3$ radicals in the aqueous

phase of clouds (Arakaki et al., 2013) (Herrmann, 2003) and $10^6$ $cm^{-3}$ and $10^7$ - $10^8$ $cm^{-3}$ in the gas phase,

respectively (Khan et al., 2008) (Cantrell et al., 1997).

The Henry's law constants for the same organic compounds are also listed **Table 1**. They cover a range of

$10^2 \leq K_H$ [ M $atm^{-1}$] $\leq 10^9$. For carboxylic acids, we also report effective Henry's law constants at pH = 3

and pH = 6 as being typical for cloud water.

### 2.2.2     Biodegradation rates

In the literature, experimental rates for metabolic processes are usually reported in units of [mol $cell^{-1}$ $s^{-1}$]

(**Table 2**). We converted these rates into first order loss rates $k^{1st}_{bact}$ [$s^{-1}$] by dividing them by the ratio of the

concentration of the organic compound [mol $L^{-1}$] to $C_{cell,aq}$ [cell $L^{-1}$] used in the respective experiments. To

obtain a value with units corresponding to chemical rate constants, this rate is divided by the constant model

cell concentration ($C_{cell,aq}$ = $1.5 \times 10^7$ cell $L^{-1}$) resulting in a range of $10^{-18} \leq k_{bact}$ [L $cell^{-1}$ $s^{-1}$] $\leq 10^{-11}$. This cell

concentration is on the same order of magnitude as found in many clouds (Amato et al., 2007c).

Experiments with 17 different cloud bacteria in artificial cloud water with pH = 5.0 and pH = 6.5 showed

also nearly identical results so it can be concluded that biodegradation rates are largely independent of pH

for values typical in cloud water (Vaïtilingom et al., 2011). Similar results were shown by Razika et al.

(2010) who demonstrated that biodegradation rates of phenol by *Pseudomonas aeruginosa* were very similar

when incubated at pH = 5.8, 7.0 and 8.0, respectively. This independence of the biodegradation rate on the

pH of the medium, within certain limits, results from the regulation of the intracellular composition and pH

in bacteria cells (~6.5 -7).

Similar to the chemical processes (Eqs 1 and 2), we express the biological activity in terms of a rate

$$R_{bact}[s^{-1}] = \frac{-d(org)_{bact}}{dt} = k_{bact}[ L\ Cell^{-1}s^{-1}] \times C_{cell,aq}[Cell\ L^{-1}] \qquad \text{Eq- 3}$$

whereas $k_{bact}$ is the bacterial rate constant [L $cell^{-1}$ $s^{-1}$] and $C_{cell,aq}$ is the concentration of bacteria in cloud

water [cell $L^{-1}$].





**Table 2.** Summary of biodegradation rates from the literature and calculated rate constants ($k_{bact}$) for the consumption of small organic species by bacteria isolated from cloud water.

| Organic compound | Bacteria type | Concentration in the experiments | | Experimental rate [mol cell⁻¹ s⁻¹] | | Calculated $k^{1st}_{bact}$ [ s⁻¹] | | Calculated $k_{bact}$ [ cell⁻¹ L s⁻¹] | | Ref |
|---|---|---|---|---|---|---|---|---|---|---|
| | | | | 17°C | 5°C | 17°C | 5°C | 17°C | 5°C | |
| | | Cells [cell mL⁻¹] | Organic [mol L⁻¹] | | | | | | | |
| Form-aldehyde | *P graminis* | $8 \times 10^4$ | 0.02 | $1.9 \times 10^{-20}$ | $8.1 \times 10^{-21}$ | $7.7 \times 10^{-11}$ | $3.2 \times 10^{-11}$ | $4.8 \times 10^{-18}$ | $2.1 \times 10^{-18}$ | 1 |
| | *Pseudomonas* sp. | | | $1.4 \times 10^{-19}$ | $8.6 \times 10^{-20}$ | $5.6 \times 10^{-10}$ | $3.4 \times 10^{-10}$ | $3.7 \times 10^{-17}$ | $2.3 \times 10^{-17}$ | |
| | *Frigoribacterium* sp. | | | $6.4 \times 10^{-21}$ | $6.4 \times 10^{-21}$ | $2.6 \times 10^{-11}$ | $2.6 \times 10^{-11}$ | $1.7 \times 10^{-18}$ | $1.7 \times 10^{-18}$ | |
| | *Bacillus* sp. | | | $2.0 \times 10^{-20}$ | $3.1 \times 10^{-21}$ | $8.1 \times 10^{-11}$ | $1.2 \times 10^{-11}$ | $5.4 \times 10^{-18}$ | $8.4 \times 10^{-19}$ | |
| Formate | *Sphingomonas* sp. | $10^9$ | 0.02 | $9.2 \times 10^{-21}$ | $3.1 \times 10^{-20}$ | $4.6 \times 10^{-7}$ | $1.6 \times 10^{-6}$ | $3.0 \times 10^{-14}$ | $1.0 \text{x} \times 0^{-13}$ | 2 |
| | *P graminis* | | | $1.3 \times 10^{-19}$ | $9.6 \times 10^{-20}$ | $6.5 \times 10^{-6}$ | $4.8 \times 10^{-6}$ | $4.3 \times 10^{-13}$ | $3.2 \times 10^{-13}$ | |
| | *Pseudomonas* sp. | | | $4.6 \times 10^{-20}$ | $8.6 \times 10^{-21}$ | $2.3 \times 10^{-6}$ | $4.3 \times 10^{-7}$ | $1.5 \times 10^{-13}$ | $2.8 \times 10^{-14}$ | |
| | *P viridiflava* | | | $1.6 \times 10^{-19}$ | $4.7 \times 10^{-20}$ | $8.1 \times 10^{-6}$ | $2.3 \times 10^{-6}$ | $5.4 \times 10^{-13}$ | $1.5 \times 10^{-13}$ | |
| | *Rhodococcus* sp. | $10^6$ | $2 \times 10^{-5}$ | $8.0 \times 10^{-19}$ | $4.0 \times 10^{-19}$ | $4.0 \times 10^{-5}$ | $2.0 \times 10^{-5}$ | $2.6 \times 10^{-12}$ | $1.3 \times 10^{-12}$ | 3 |
| | *Pseudomonas* sp. | | | $1.5 \times 10^{-18}$ | $8.0 \times 10^{-19}$ | $7.5 \times 10^{-6}$ | $4.0 \times 10^{-5}$ | $5.0 \times 10^{-13}$ | $3.4 \times 10^{-13}$ | |
| | *P syringae* | | | $2.3 \times 10^{-18}$ | $2.0 \times 10^{-18}$ | $1.1 \times 10^{-4}$ | $1.0 \times 10^{-4}$ | $7.3 \times 10^{-12}$ | $6.6 \times 10^{-12}$ | |
| | *P graminis* | | | $5.0 \times 10^{-18}$ | $1.0 \text{x} \times 10^{-18}$ | $2.5 \times 10^{-4}$ | $5.0 \times 10^{-5}$ | $1.6 \times 10^{-11}$ | $3.3 \times 10^{-12}$ | |
| | Various microorganisms | $8 \times 10^4$ | $43 \times 10^{-6}$ | $2.1 \times 10^{-18}$ | | $4.1 \times 10^{-6}$ | | $2.7 \times 10^{-13}$ | | 4 |
| Acetate | *Sphingomonas* sp. | $10^9$ | 0.02 | $2.7 \times 10^{-20}$ | $1.6 \times 10^{-22}$ | $1.3 \times 10^{-6}$ | $8.2 \times 10^{-9}$ | $1.1 \times 10^{-13}$ | $5.4 \times 10^{-16}$ | 2 |
| | *P graminis* | | | $3.0 \times 10^{-19}$ | $3.0 \times 10^{-20}$ | $1.5 \times 10^{-5}$ | $1.5 \times 10^{-6}$ | $1.0 \times 10^{-12}$ | $1.0 \times 10^{-13}$ | |
| | *Pseudomonas* sp. | | | $2.6 \times 10^{-20}$ | $1.7 \times 10^{-20}$ | $1.3 \times 10^{-6}$ | $8.8 \times 10^{-7}$ | $8.7 \times 10^{-14}$ | $5.3 \times 10^{-14}$ | |
| | *P viridiflava* | | | $5.6 \times 10^{-20}$ | $1.1 \times 10^{-20}$ | $2.8 \times 10^{-6}$ | $6.0 \times 10^{-7}$ | $1.8 \times 10^{-13}$ | $4.0 \times 10^{-14}$ | |
| | *Rhodococcus* sp. | $10^6$ | $2 \times 10^{-5}$ | $5.0 \times 10^{-18}$ | $1.0 \times 10^{-18}$ | $2.5 \times 10^{-4}$ | $5.0 \times 10^{-5}$ | $1.6 \times 10^{-11}$ | $3.3 \times 10^{-12}$ | 3 |
| | *Pseudomonas* sp. | | | $1.3 \times 10^{-18}$ | $6.0 \times 10^{-19}$ | $6.7 \times 10^{-5}$ | $3.0 \times 10^{-5}$ | $4.5 \times 10^{-12}$ | $2.0 \times 10^{-12}$ | |
| | *P syringae* | | | $8.6 \times 10^{-19}$ | $2.0 \times 10^{-19}$ | $4.3 \times 10^{-5}$ | $1.0 \times 10^{-5}$ | $2.8 \times 10^{-12}$ | $6.6 \times 10^{-13}$ | |
| | *P graminis* | | | $4.0 \times 10^{-19}$ | $1.0 \times 10^{-19}$ | $2.0 \times 10^{-5}$ | $5.0 \times 10^{-6}$ | $1.3 \times 10^{-12}$ | $3.3 \times 10^{-13}$ | |
| | Various microorganisms | $8 \times 10^4$ | $2.5 \times 10^{-6}$ | $1.94 \times 10^{-18}$ | | $6.25 \times 10^{-3}$ | | $4.16 \times 10^{-10}$ | | 4 |
| Succinate | *Sphingomonas* sp. | $10^9$ | 0.02 | $2.6 \times 10^{-20}$ | $1.0 \times 10^{-20}$ | $1.3 \times 10^{-6}$ | $5.4 \times 10^{-7}$ | $8.6 \times 10^{-14}$ | $3.6 \times 10^{-14}$ | 2 |
| | *P graminis* | | | $1.0 \times 10^{-19}$ | $9.8 \times 10^{-20}$ | $5.2 \times 10^{-6}$ | $4.9 \times 10^{-6}$ | $3.4 \times 10^{-13}$ | $3.2 \times 10^{-13}$ | |
| | *Pseudomonas* sp. | | | $1.4 \times 10^{-20}$ | $1.8 \times 10^{-20}$ | $7.0 \times 10^{-7}$ | $9.2 \times 10^{-7}$ | $4.7 \times 10^{-14}$ | $6.1 \times 10^{-14}$ | |
| | *P viridiflava* | | | $2.9 \times 10^{-20}$ | $6.1 \times 10^{-20}$ | $1.4 \times 10^{-6}$ | $3.0 \times 10^{-6}$ | $9.7 \times 10^{-14}$ | $2.0 \times 10^{-13}$ | |
| | *Rhodococcus* sp. | $10^6$ | $2 \times 10^{-5}$ | $5.0 \times 10^{-20}$ | $4.0 \times 10^{-20}$ | $2.5 \times 10^{-6}$ | $2.0 \times 10^{-6}$ | $1.6 \times 10^{-13}$ | $1.3 \times 10^{-13}$ | 3 |
| | *Pseudomonas* sp. | | | $1.5 \times 10^{-19}$ | $2.0 \times 10^{-20}$ | $7.5 \times 10^{-6}$ | $1.0 \times 10^{-6}$ | $5.0 \times 10^{-13}$ | $6.6 \times 10^{-14}$ | |
| | *P syringae* | | | $6.8 \times 10^{-19}$ | $1.7 \times 10^{-19}$ | $3.4 \times 10^{-5}$ | $8.8 \times 10^{-6}$ | $2.2 \times 10^{-12}$ | $5.8 \times 10^{-13}$ | |
| | *P graminis* | | | $5.0 \times 10^{-19}$ | $1.0 \times 10^{-19}$ | $2.5 \times 10^{-5}$ | $5.0 \times 10^{-6}$ | $1.6 \times 10^{-12}$ | $3.3 \times 10^{-13}$ | |
| | Various microorganisms | $8 \times 10^4$ | $3.1 \times 10^{-6}$ | $5.6 \times 10^{-19}$ | | $1.4 \times 10^{-5}$ | | $9.6 \times 10^{-13}$ | | 4 |
| Malonate | Various microorganisms | $8 \times 10^4$ | $3.1 \times 10^{-6}$ | $5.2 \times 10^{-19}$ | | $1.3 \times 10^{-5}$ | | $9.0 \times 10^{-13}$ | | 4 |
| Catechol | *R enclensis* | $10^7$ | 0.0001 | $4.1 \times 10^{-19}$ | | $4.1 \times 10^{-8}$ | | $2.7 \times 10^{-15}$ | | 5 |
| Phenol | *R enclensis* | $10^9$ | 0.0001 | $5.0 \times 10^{-20}$ | | $5.0 \times 10^{-4}$ | | $3.3 \times 10^{-11}$ | | 5 |
| Methanol | *P graminis* | $8 \times 10^4$ | 0.02 | $5.6 \times 10^{-22}$ | - | $2.2 \times 10^{-12}$ | - | $1.5 \times 10^{-19}$ | 0 | 1 |
| | *P syringae* | | | $5.7 \times 10^{-21}$ | $5.8 \times 10^{-22}$ | $2.3 \times 10^{-11}$ | $2.3 \times 10^{-12}$ | $1.5 \times 10^{-18}$ | $1.5 \times 10^{-19}$ | |
| | *Frigoribacterium* sp. | | | $3.5 \times 10^{-23}$ | $2.5 \times 10^{-23}$ | $1.4 \times 10^{-13}$ | $1.0 \times 10^{-13}$ | $9.4 \times 10^{-21}$ | $6.7 \times 10^{-21}$ | |
| | *Bacillus* sp. | | | $2.9 \times 10^{-21}$ | - | $1.1 \times 10^{-11}$ | - | $7.8 \times 10^{-19}$ | 0 | |

(1)(Husárová et al., 2011); (2)(Vaïtilingom et al., 2010); (3)(Vaïtilingom et al., 2011); (4)(Vaïtilingom et al., 2013); (5)(Jaber et al., 2020). Several experiments showed that temperature only had a small influence on the biodegradation rates that differed at most by a factor of 2 in experiments at 5°C or 17°C (Husárová et al., 2011) (Vaïtilingom et al., 2010, 2011).





### 2.3 Definition of model output parameters

#### 2.3.1 Relative contributions to biodegradation and chemical loss rates

Several previous studies compared biodegradation and chemical rates in order to conclude on the potential importance of biological processes in clouds (Ariya et al., 2002) (Vaïtilingom et al., 2011, 2013)(Husárová et al., 2011)(Jaber et al., 2020). Similarly, we define the relative contributions of the bacterial and the chemical processes in the two phases to the total loss rate of an organic compound

$$R'_{bact}[mol\ L^{-1}s^{-1}] = R_{bact}[s^{-1}] \times C_{org,aq}\ [mol\ L^{-1}] \qquad \text{Eq-4}$$

$$R'_{chemaq}[mol\ L^{-1}s^{-1}] = R_{chemaq}[\ s^{-1}] \times C_{org,aq}\ [mol\ L^{-1}] \qquad \text{Eq-5}$$

$$R'_{chemgas}[\ cm^{-3}\ s^{-1}] = R_{chemgas}[\ s^{-1}] \times C_{org,gas}[cm^{-3}] \qquad \text{Eq-6}$$

where $C_{org,aq}$ and $C_{org,gas}$ are the concentration of VOC in the aqueous [mol $L^{-1}$] and gas phases [$cm^{-3}$], respectively. Their relative contributions are then expressed as

$$fr_{bact}[\%] = \frac{R'_{bact}}{R'_{bact} + R'_{chemaq} + R'_{chemgas}} \times 100 \qquad \text{Eq- 7}$$


$$fr_{chemaq}[\%] = \frac{R'_{chemaq}}{R'_{bact} + R'_{chemaq} + R'_{chemgas}} \times 100 \qquad \text{Eq- 8}$$

$$fr_{chemgas}[\%] = \frac{R'_{chemgas}}{R'_{bact} + R'_{chemaq} + R'_{chemgas}} \times 100 \qquad \text{Eq- 9}$$

whereas these fractions always add up to 100%. As NVOC are only in the aqueous phase, $fr_{chemgas,NVOC} = 0$.

#### 2.3.2 Contributions to total loss of organics by bacterial and chemical processes

While $fr_{bact}$, $fr_{chemaq}$ and $fr_{chemgas}$ define the relative importance of the bacterial and chemical loss rates, they do not give quantitative information on the total loss of the organics. To define this sink strength, we introduce the parameter $L_t$ [%] as:

$$L_t[\%] = 100 - \left[\frac{[Org]_t}{[Org]_0} \times 100\right] = L_{bact} + L_{chemaq} + L_{chemgas} \qquad \text{Eq-10}$$

whereas $[Org]_0$ is the initial organic concentration (1 ppb) and $[Org]_t$ is the organic concentration at time t (for most simulations: t = 600 s). Accordingly, $L_{bact}$, $L_{,chemaq}$ and $L_{chemgas}$ are the fractions [%] of organics consumed by bacteria and radicals in the aqueous and gas phases, respectively.

$$L_{bact}[\%] = fr_{bact}[\%] \times L_t[\%] \div 100\% \qquad \text{Eq-11}$$
$$L_{chemaq}[\%] = fr_{chemaq}[\%] \times L_t[\%] \div 100\% \qquad \text{Eq-12}$$
$$L_{chemgas}[\%] = fr_{chemgas}[\%] \times L_t[\%] \div 100\% \qquad \text{Eq-13}$$

As both the fractions 'fr' and $L_t$ are expressed in percent, we divided by 100% in Equations 11-13. Unlike





the sum of $fr_{bact}$, $fr_{chemaq}$ and $fr_{chemgas}$ that always yields 100% (Eqs 7- 9), by definition, $L_t$ does not have to reach 100%. As we only consider one cloud cycle in our simulations (t = 600 s ), the values of $L_t$, $L_{bact}$, $L_{chemgas}$ and $L_{chemaq}$ are rather small (a few percent at most); however, it should be kept in mind that particles

likely undergo multiple cloud cycles during their residence time in the atmosphere. Thus, the contribution of chemical and biological processes to the total loss for a specific organic can be extrapolated for longer time scales based on our results. However, as one of the main goals of our study is to compare microbial activity to the better constrained chemical losses, our conclusions will be independent of the time scales.

## 3.  Results and discussion

### 3.1  Relative loss rate of organics by biological processes ($fr_{bact}$)

#### 3.1.1  $fr_{bact}$ of VOC

We first compare the contributions of biodegradation and chemical losses to the total loss rate for VOC. In order to cover a representative parameter range for the physicochemical properties of the organic compound, we performed five simulations, in all of which the full ranges of $K_H$ and of $R_{chemaq}$ over eight orders of

magnitude were explored (***Section 2.2***). The simulations differ by the assumptions of $R_{chemgas}$ and $R_{bact}$ that are kept constant in the individual model runs: For the results shown in ***Figure 2a, b,*** and ***c***, we applied $R_{bact}$ = $10^{-8}$ s$^{-1}$, $10^{-6}$ s$^{-1}$ and $10^{-4}$ s$^{-1}$ and $R_{chemgas}$ = $10^{-6}$ s$^{-1}$, respectively. For the three sets of simulations shown in ***Figure 2b, d,*** and ***e,*** $R_{bact}$ = $10^{-6}$ s$^{-1}$ and $R_{chemgas}$ equal to $10^{-6}$ s$^{-1}$, $10^{-5}$ s$^{-1}$ and $10^{-4}$ s$^{-1}$, were assumed respectively. Thus, in total three values each for $R_{chemgas}$ and $R_{bact}$ are discussed in the following.

In general, for all combinations of $R_{chemgas}$ and $R_{bact}$, the highest $fr_{bact}$ is predicted for organics with the highest solubility ($K_H$ [M atm$^{-1}$] $\geq 10^8$) and lowest chemical reaction rate in the aqueous phase ($R_{chemaq}$ [s$^{-1}$] $\leq 10^{-11}$). For the lowest biological activity ($R_{bact}$ = $10^{-8}$ s$^{-1}$, ***Figure 2a***), $fr_{bact}$ reaches a maximum value of ~100%. For higher biological activity ($R_{bact}$ = $10^{-6}$ s$^{-1}$ and $10^{-4}$ s$^{-1}$), $fr_{bact}$ is always smaller and only reaches at most ~80% (***Figure 2b and 2c***). This trend seems counterintuitive as for the highest $R_{bact}$, the highest importance of

biodegradation may be expected. We will explore the reasons for this further in ***Section 3.1.2*** where we compare loss rates in individual droplets as a function of time.

The comparison of ***Figures 2b*** and ***2c*** shows that the parameter ranges, for which $fr_{bact}$ has the maximum value, are broader for higher $R_{bact}$: When $R_{bact}$ = $10^{-6}$ s$^{-1}$ (***Figure 2b***), $fr_{bact}$ ~80% within the ranges of ~$10^7 \leq$ $K_H$ [M atm$^{-1}$] $\leq$ ~$10^9$ and $10^{-12} \leq R_{chemaq}$ [s$^{-1}$] $\leq 10^{-10}$, i.e. both $R_{chemaq}$ and $K_H$ span about two orders of

magnitude. For $R_{bact}$ = $10^{-4}$ s$^{-1}$ (***Figure 2c***), these ranges are ~$10^5 \leq K_H$ [ M atm$^{-1}$] $\leq$ ~$10^9$ and $10^{-12} \leq R_{chemaq}$ [s$^{-1}$] $\leq 10^{-9}$, i.e., wider by about two and one order of magnitude, respectively.





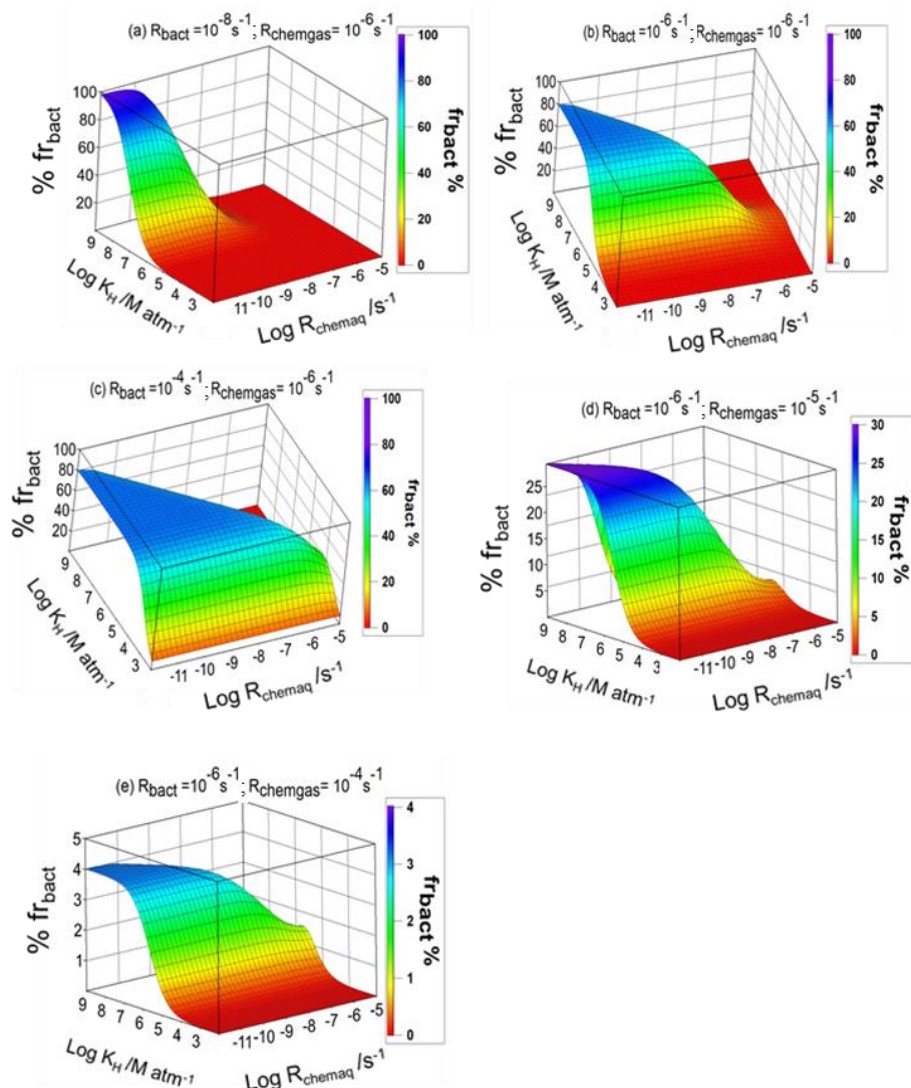

**Figure 2**. Relative contribution of bacteria($fr_{bact}$) to the total loss rate of organics as a function of the full ranges of $R_{chemaq}$ and $K_H$. $R_{chemgas} = 10^{-6}$ s$^{-1}$ and a) $R_{bact} = 10^8$ s$^{-1}$, b) $R_{bact} = 10^{-6}$ s$^{-1}$, c) $R_{bact} = 10^{-4}$ s$^{-1}$ and for $R_{bact} = 10^{-6}$ s$^{-1}$ with d) $R_{chemgas} = 10^{-5}$ s$^{-1}$ and e) $R_{chemgas} = 10^{-4}$ s$^{-1}$.

This widening of the parameter spaces can be explained by the contributions of the chemical processes to the total organic loss rate, i.e. $fr_{chemaq}$ and $fr_{chemgas}$ (**Figures S1** and **S2** in the supplement): For the two values of $R_{bact}$, $fr_{chemaq}$ is highest (> 80%) for a combination of (i) ~$10^5 \leq K_H$ [ M atm$^{-1}$] $\leq 10^9$ and ~$10^{-9} \leq R_{chemaq}$ [s$^{-1}$] $\leq 10^{-5}$, or of (ii) low/intermediate solubility (~$10^3 \leq K_H$ [M atm$^{-1}$] $\leq$ ~$10^5$) and high chemical aqueous phase reactivity ($10^{-6} \leq R_{chemaq}$ [s$^{-1}$] $\leq 10^{-5}$). These trends are due to the predominant partitioning of highly



soluble organics to the aqueous phase (> 90% for $K_H \geq 10^6$ M atm$^{-1}$). If $R_{chemaq}$ is low, the organics do not undergo efficient chemical processes in the aqueous phase. Therefore, $fr_{bact}$ is highest for this parameter combination. The comparison of the results for $R_{bact} = 10^{-6}$ s$^{-1}$ and three values of $R_{chemgas}$ (**Figure 2b, d, e**)

shows a decrease of the maximum value of $fr_{bact}$ from 80% (**Figure 2b**) to 4% (**Figure 2e**) for similar ranges of $K_H$ and $R_{chemaq}$ because of the dependence of $fr_{bact}$ on $fr_{chemgas}$: for compounds with highest $R_{chemgas}$, the dominant loss is the gas phase reaction, leading to a high $fr_{chemgas}$ and consequently to a lower $fr_{bact}$ (**Figure 2e**). Therefore, the parameter ranges of $K_H$ and $R_{chemaq}$, where $fr_{bact}$ is maximum, do not change for different $R_{chemgas}$, but idecrease when the gas phase chemistry dominates the loss of the organic.

Overall, the variation in $fr_{bact}$ as a response to changes in $R_{chemaq}$, $R_{chemgas}$ and $K_H$ shows different sensitivities: For example, for organics with $K_H = 10^6$ M atm$^{-1}$, $R_{chemaq} = 10^{-9}$ s$^{-1}$ and $R_{chemgas} = 10^{-6}$ s$^{-1}$, $fr_{bact}$ is ~8% when $R_{bact} = 10^{-8}$ s$^{-1}$ (red range in **Figure 2a**). This fraction increases to $fr_{bact}$ ~ 73%, i.e. by a factor of ~ 9, when $R_{bact} = 10^{-6}$ s$^{-1}$ (blue part in **Figure 2b**) and approaches ~80% when $R_{bact} = 10^{-4}$ s$^{-1}$ (light blue part in **Figure 2c**). Similarly, for organics with $K_H \sim 10^6$ M atm$^{-1}$, $R_{chemaq} = 10^{-9}$ s$^{-1}$ and $R_{bact} = 10^{-6}$ s$^{-1}$, an increase in $R_{chemgas}$

from $10^{-6}$ s$^{-1}$ (**Figure 2b**) to $10^{-5}$ s$^{-1}$ (**Figure 2d**) and $10^{-4}$ s$^{-1}$ (**Figure 2e**) decreases $fr_{bact}$ from 73% to 23% and 3%. Based on these non-linear trends, one can hypothesize that (i) a change $R_{bact}$ and/or $R_{chemgas}$ translates into a less than proportional change in $fr_{bact}$, and (ii) an increase of $R_{bact}$ might translate into a larger change in $fr_{bact}$ than an increase of $R_{chemgas}$ by the same factor. Therefore, $fr_{bact}$ is more sensitive to a change in $R_{bact}$ than in $R_{chemgas}$. Given that $R_{chemgas}$ only differs by about two orders of magnitude for most organics relevant

in the atmospheric multiphase system (**Table 1**), we conclude that $fr_{bact}$ may be largely independent of the gas phase chemical reactivity. Additional sensitivity studies (not shown) reveal that using combinations of $R_{bact}$ and $R_{chemgas}$ other than those in **Figure 2**, result in slightly different locations of the maximum of $fr_{bact}$ but in similar shapes and widths of parameter spaces, for which $fr_{bact}$ is maximum. Therefore, our conclusions on the sensitivities seem robust for wide parameters ranges and combinations.

### 245 3.1.2 $fr_{bact}$ of NVOC and comparison to $fr_{bact}$ of VOC

For NVOC, the analysis of $fr_{bact}$ is limited to exploring the ranges of $R_{bact}$ and $R_{chemaq}$ (**Figure 1**). Similar to the findings based on **Figure 2a, b and c** for VOC, highest values of $fr_{bact}$ are predicted for intermediate values of $10^{-9} \leq R_{bact}$ [s$^{-1}$] $\leq 10^{-6}$ and $R_{chemaq}$ [s$^{-1}$] $\leq 10^{-10}$; for the highest biological activity ($R_{bact} = 10^{-4}$ s$^{-1}$), $fr_{bact}$ is nearly zero (**Figure 3**). In order to understand the reasons for these trends, we explore in the following

the variables included determining $fr_{bact}$ (**Equation 7**).



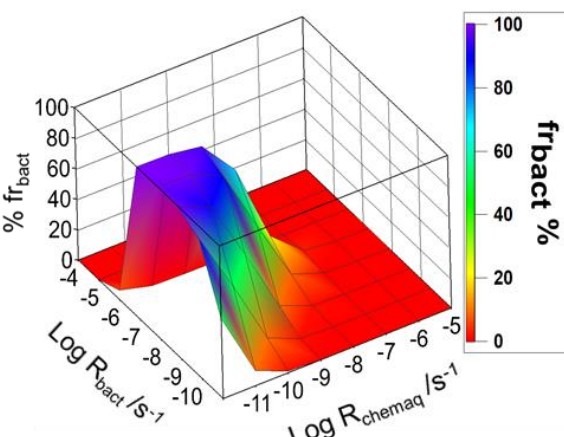

**Figure 3:** Relative contribution of bacteria to the total loss rate of organics ($fr_{bact}$) for the full range of $10^{-9} \leq R_{bact} [s^{-1}] \leq 10^{-6}$ and $10^{-12} \leq R_{chemaq} [s^{-1}] \leq 10^{-5}$.

As shown in *Figure 1*, only one drop size class in the model contains bacteria; in all other droplets, the bacterial activity is zero and only chemical processes occur. In *Figure 4a*, we show the evolution of the organic concentration of NVOC in the bacteria-containing droplets ([Org]$_{bact}$) over 600 s, $R_{chemaq} = 10^{-11}$ s$^{-1}$ and $R_{chemgas} = 10^{-6}$ s$^{-1}$, i.e. for relatively low chemical reactivity in both phases. It is obvious that for the highest $R_{bact}$ ($10^{-4}$ s$^{-1}$), the organics are immediately depleted whereas for lower biological activity ($R_{bact} = 10^{-6}$ s$^{-1}$), [Org]$_{bact}$ decreases more slowly or even stays constant ($R_{bact} = 10^{-8}$ s$^{-1}$). As for the highest $R_{bact}$, the concentration approaches zero after ~10 seconds, R'$_{bact}$ (*Equation-4*) becomes negligible and $fr_{bact}$ approaches 0% (Eq-7, *Figure 4c*). In *Figure 4b*, we show the equivalent to *Figure 4a*, but for VOC and for three $K_H$ values ($10^2$, $10^5$ and $10^9$ M atm$^{-1}$). For the lowest $R_{bact}$ and all $K_H$ values, [Org]$_{bact}$ stays also nearly constant at the initial value as the organics are not efficiently consumed in any droplet. For the intermediate and highest $R_{bact}$ ($10^{-6}$ and $10^{-4}$ s$^{-1}$) and $K_H = 10^9$ M atm$^{-1}$, [Org]$_{bact}$ remains higher than for the NVOC and levels off at t ~300 seconds. For these two $R_{bact}$, [Org]$_{bact}$ significantly drops by several orders of magnitude. However, unlike for the NVOC, $fr_{bact}$ does not drop to ~0% but levels off at ~70% for $K_H = 10^9$ M atm$^{-1}$ (*Figure 4d*). For these two $R_{bact}$ and $K_H = 10^5$ M atm$^{-1}$, [Org]$_{bact}$ stays also nearly constant over the whole simulation time (600s) and is higher for lowest $R_{bact}=10^{-6}$ s$^{-1}$ (*Figure 4b*). However, $fr_{bact}$ is higher for the highest $R_{bact}=10^{-4}$ s$^{-1}$ (*Figure 4d*). *Figure S3a* and *S3b* show the ratio of the organic concentrations in bacteria-containing and bacteria-free droplets of the same diameter. For these conditions of low $R_{chemgas}$ and $R_{chemaq}$, the concentration ratio is near unity for both NVOC and VOC ($K_H= 10^5$ and $10^9$ M atm$^{-1}$) when $R_{bact} = 10^{-8}$ s$^{-1}$. For higher $R_{bact}= 10^{-6}$ s$^{-1}$ and $K_H = 10^5$ and $10^9$ M atm$^{-1}$, the concentration ratio for the VOC is near unity and $10^{-3}$, respectively and for the NVOC it is much lower (~$10^{-6}$). For $R_{bact} = 10^{-4}$ s$^{-1}$, this ratio is also higher for $K_H = 10^5$ M atm$^{-1}$ than for $K_H = 10^9$ M atm$^{-1}$ (~$10^{-1}$ and $10^{-5}$ respectively) and << $10^{-11}$ for





NVOC. It can be summarized that (1) for VOC, the concentration in bacteria-containing droplets is higher for $K_H = 10^5$ M atm$^{-1}$ than for $K_H = 10^9$ M atm$^{-1}$ and (2) the concentration in bacteria-containing droplets is predicted to be always smaller for the NVOC than for the VOC with at least intermediate solubility.

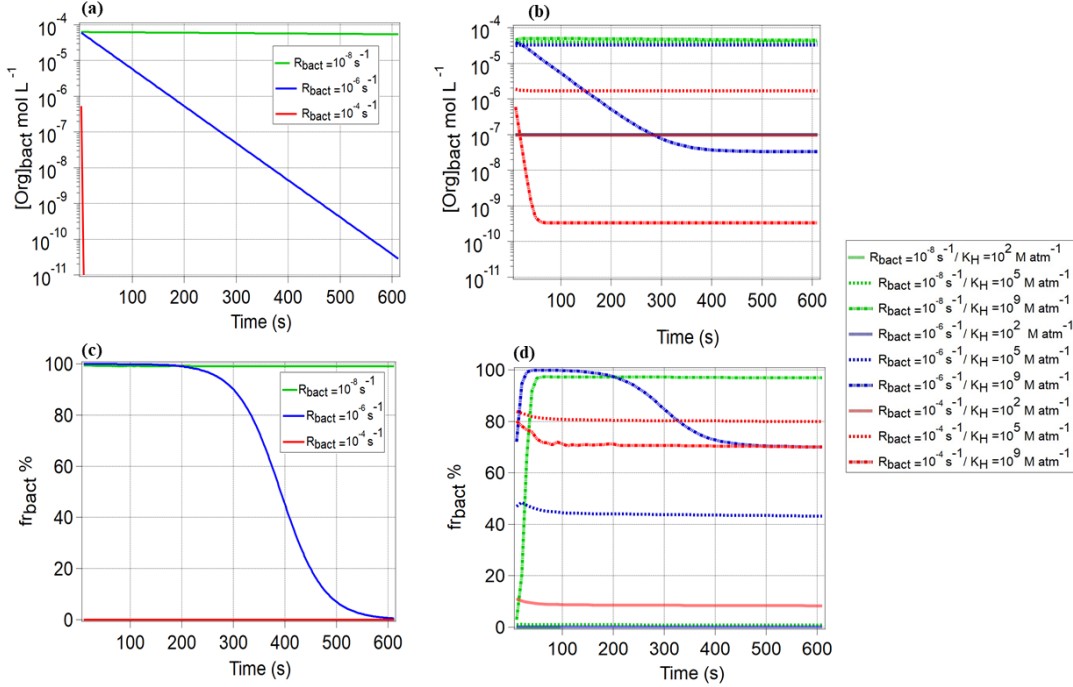

**Figure 4.** Comparison of the organic concentration in the bacteria-containing droplets [Org]$_{bact}$ [mol L$^{-1}$] for (a) NVOC and (b) VOC ($K_H = 10^2$, $10^5$ and $10^9$ M atm$^{-1}$), and the resulting fr$_{bact}$ for these compounds (c, d). Results are shown for R$_{bact}$ =$10^{-8}$, $10^{-6}$ and $10^{-4}$ s$^{-1}$ , and R$_{chemgas}$=$10^{-6}$ s$^{-1}$ ,R$_{chemaq}$ = $10^{-11}$ s$^{-1}$.

This difference between VOC and NVOC can be explained by the schematic in *Figure 5*: The insets in the droplets schematically depict the temporal evolution of the organic concentrations as shown in *Figure 4*. The efficient consumption of organics by bacteria in the bacteria-containing droplets (I) leads to a significant decrease of the organic concentration in these droplets. For VOC, this results in a strong deviation from thermodynamic equilibrium of the gas and aqueous phase concentrations, as defined by Henry's law (II).

As a consequence, organics diffuse from the gas phase into the bacteria-containing droplets (III). As this diffusion leads to a decrease of the gas phase concentration, thermodynamic equilibrium between the gas phase and the bacteria-free droplets is not fulfilled anymore (IV) resulting in a concentration gradient between these droplets and the gas phase. Finally, organics from the bacteria-free droplets evaporate to replenish the gas phase concentration (V) and eventually the organic concentration in the bacteria-containing


droplets. These coupled equilibria between the gas phase and droplets lead to a continuous replenishment of VOC in the bacteria-containing droplets. However, since all processes are associated with time scales (evaporation, gas phase diffusion, transport through the gas/water interface, loss by chemical and biological processes in the aqueous phase; (Schwartz, 1986) thermodynamic equilibrium is not instantaneously established. For organics with low $K_H$ (~$10^2$ M atm$^{-1}$), the fraction of dissolved material is very small,

therefore, organics are mostly consumed in the gas phase and only a small fraction is partitioned to the droplets. For intermediate $K_H$ (~$10^5$ M atm$^{-1}$), the consumption of the organics in the bacteria-containing droplets is fast leading to large concentration gradients between the phases. Consequently, the evaporation and diffusion rates are high and thus the replenishment of the bacteria containing droplets by the organics is most efficient. Highly soluble organics ($K_H = 10^9$ M atm$^{-1}$) are nearly completely dissolved in the aqueous

phase and thus, only small fraction will be consumed by the bacteria leading to relatively smaller concentration gradient between gas and bacteria-containing droplets (Process I and ***Figure 4b***) and therefore to less efficient replenishment and longer time scales to establish thermodynamic equilibrium. Thus for such organics, the organic consumption by bacteria is relatively less efficient. For NVOC, only Process I occurs as there is no coupling to the gas phase and thus, only the amount of organics that is initially dissolved in

the bacteria-containing droplets can be consumed.

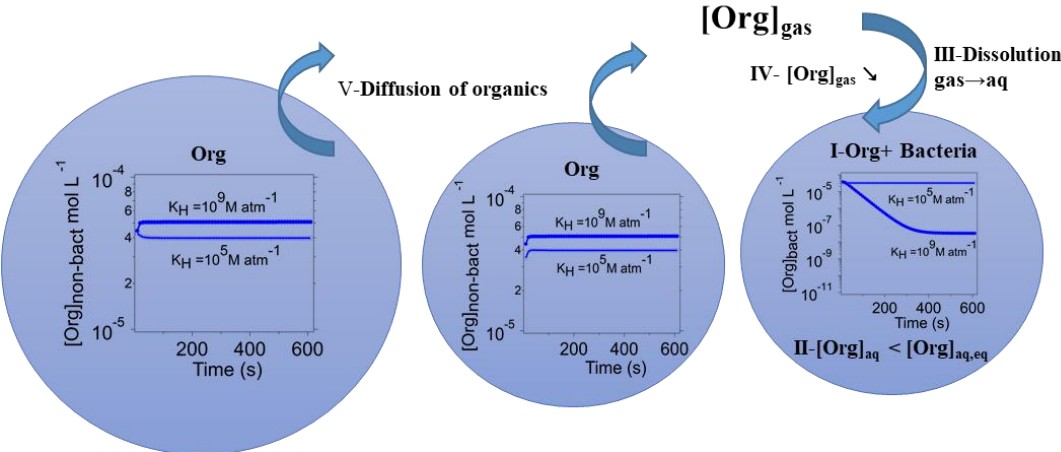

**Figure 5**. **:** Schematic of the partitioning of VOC between the bacteria-containing and bacteria-free droplets and the gas phase: The inset figures show the temporal evolution of the organic concentration in the bacteria-containing droplets ([Org]$_{bact}$) and bacteria free-droplets ([Org]$_{non-bact}$) for VOC with $K_H = 10^5$ M atm$^{-1}$ and $10^9$ M atm$^{-1}$.



## 3.2 Total loss of the organic compound ($L_t$) and sinks of organics by bacteria ($L_{bact}$)

### 3.2.1 $L_t$ for VOC

While the analysis of $fr_{bact}$ in **Section 3.1** quantifies the relative importance of the biological and chemical processes for the organic loss, we explore in the following the absolute loss of these processes ($L_t$, Eq-10). Unlike the sum of $fr_{bact}$, $fr_{chemaq}$, and $fr_{chemgas}$ that always yields 100%, the ranges of $L_t$ that will be discussed in the following are rather small, i.e. $10^{-4} \leq L_t[\%] \leq 12$, given that we only simulate approximately one cloud cycle (600s). **Figure 6** shows $L_t$ for the same parameters ($R_{chemgas}$, $R_{bact}$) as in **Figure 2**. For all combinations

of $R_{bact}$ and $R_{chemgas}$, the highest values of $L_t$ are predicted for organics with highest $K_H$ and $R_{chemaq}$. Generally, all panels show the highest values for $L_t$ for the highest $R_{chemaq}$ but in all panels there is a small maximum, for low to intermediate $K_H$ and nearly independent of $R_{chemaq}$. This feature becomes more prominent with increasing $R_{chemgas}$ ($10^{-6}$ s$^{-1}$, $10^{-5}$ s$^{-1}$, $10^{-4}$ s$^{-1}$, **Figure 6 c-e**), with the overall highest $L_t$ (~12%) for the highest $R_{chemgas}$ and lowest $K_H$. Comparing **Figure 6a, b, and c**, for which the model conditions only differ in $R_{bact}$,

a small increase in $L_t$ from 1.2% to 1.6% at intermediate $K_H$ values can be seen. We focus on these trends in **Section 3.2.2** where the contribution of the bacterial process to $L_t$ (i.e. $L_{bact}$) will be explored. Comparing the shapes of the three panels, shows that $L_t$ is additive as with increasing $R_{bact}$, $L_t$ reaches a maximum (1.6%) at $K_H \sim 10^4$ M atm$^{-1}$ and $R_{chemaq} \sim 10^{-5}$ s$^{-1}$ (**Figure 6c**). Similar to our findings for $fr_{bact}$ (**Section 3.1.1.**), we also see in the trends of $L_t$ that a change of several orders of magnitude in $R_{bact}$ or $R_{chemgas}$ translates into

a smaller change in $L_t$. Thus, one can conclude that $L_t$ has a similarly low sensitivity to the various parameters as $fr_{bact}$.






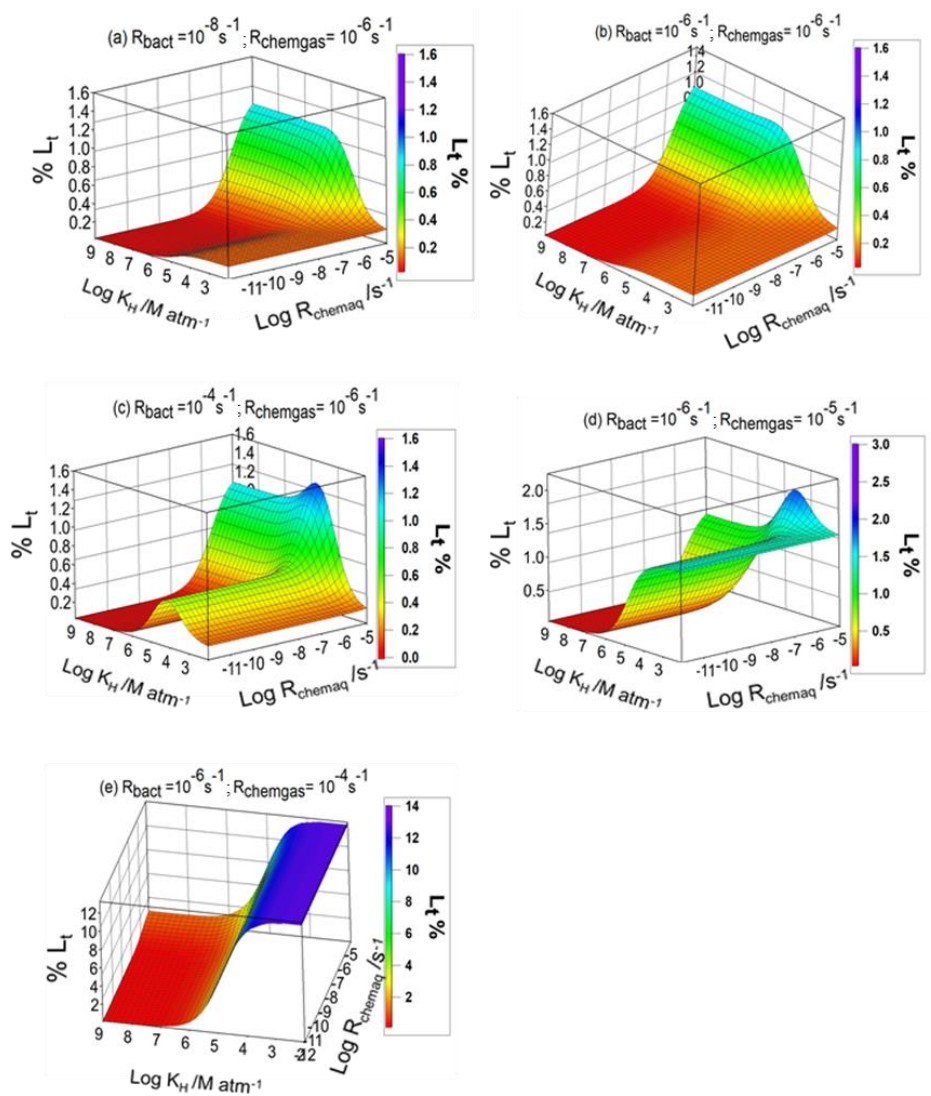

**Figure 6:** The total loss of VOC ($L_t$) as a function of the full ranges of $R_{chemaq}$ and $K_H$. $R_{chemgas} = 10^{-6}$ s$^{-1}$ and a) $R_{bact} = 10^8$ s$^{-1}$, b) $R_{bact} = 10^{-6}$ s$^{-1}$, c) $R_{bact} = 10^{-4}$ s$^{-1}$ and for $R_{bact} = 10^{-6}$ s$^{-1}$ with d) $R_{chemgas} = 10^{-5}$ s$^{-1}$ and e) $R_{chemgas} = 10^{-4}$ s$^{-1}$.





### 3.2.2 $L_{bact}$ for VOC


To understand the contribution of bacteria ($L_{bact}$) in the total consumption of the organics ($L_t$), we explore $L_{bact}$ for the same values of $R_{bact}$ and $R_{chemgas}$ as in *Figures 2* and *6* (*Figure 7*). As suggested in the comparison of *Figures 6a-c*, the contribution of the organic loss by bacteria increases with increasing $R_{bact}$ (*Figure 7a-c*), i.e. $L_{bact}$ increases from 0.025% to 0.45%.

The different shape of *Figure 7* compared to other panels is somewhat misleading as the scales of the z-axes differ. For all conditions, there is a contribution of $L_{bact} \sim 0.025\%$ for the lowest $R_{chemaq}$ and highest $K_H$, i.e. when chemical activity in the aqueous phase is lowest and solubility is highest. However, when $R_{bact} \geq 10^{-6}$ s$^{-1}$, the maximum value of $L_{bact}$ is predicted for a narrow range of $\sim 10^4 \leq K_H$ [M atm$^{-1}$] $\leq 10^6$, nearly independent of $R_{chemaq}$. The highest value of $L_{bact}$ (0.4%) is observed for highest $R_{bact}$ ($R_{bact} = 10^{-4}$ s$^{-1}$; *Figure 7c*). This corresponds to nearly a quarter of the total loss ($L_t \leq 1.4\%$, *Figure 6c*). To further explore why $L_{bact}$ has a maximum for this intermediate values of $K_H$, we compare $L_{bact}$ for $K_H = 10^2$ M atm$^{-1}$, $10^5$ M atm$^{-1}$ and $10^9$ M atm$^{-1}$ for $R_{bact} = 10^{-6}$ s$^{-1}$, $R_{chemgas} = 10^{-6}$ s$^{-1}$ and $R_{chemaq} = 10^{-11}$ s$^{-1}$, i.e. for the same conditions as the examples in *Figure 4b* and *d*. When $K_H$ is low ($10^2$ M atm$^{-1}$), a very small fraction (0.16%) of the organics is dissolved and, thus, $L_{bact}$ is lowest. For $K_H = 10^9$ M atm$^{-1}$ (*Figure 4b*, blue line), [Org]$_{aq}$ is low

because - as mentioned before - it is consumed quickly and the transport from the gas phase is not fast enough resulting in a low $L_{bact}$. For $K_H \sim 10^5$ M atm$^{-1}$ (*Figure 4b*, blue line), [Org]$_{aq}$ is relatively high because the thermodynamic equilibrium can be more quickly established and, thus, $L_{bact}$ is high.

For each set of conditions, i.e. in the various panels of *Figure 7*, the maximum value $L_{bact}$ is independent of $R_{chemaq}$ and does not greatly vary for the same $R_{bact}$ ($L_{bact,max} \sim 0.05\%$) (*Figure 7b, d, e*). This is different than

the trends of $fr_{bact}$ (*Figure 2*) that show a decrease with increasing $R_{chemaq}$. By definition (Eq-7), $fr_{bact}$ and $fr_{chemaq}$ are coupled and thus an increase in one value causes a decrease in the other. Contrary, $L_{bact}$ is independent of the chemical contributions as it describes the absolute consumption rate related to the initial organic concentration. Comparing the trends in *Figures 2 and 7*, it is evident that the maximum of $L_{bact}$ (intermediate $K_H$, independent of $R_{chemaq}$) does not coincide with the parameter ranges for which $fr_{bact}$ is

maximum (highest $K_H$, lowest $R_{chemaq}$). This finding highlights that previous estimates on the importance of biodegradation that were solely based on comparing rates (Jaber et al., 2020) are misleading. In these studies, it was concluded that biodegradation for highly soluble compounds is likely most important.






**Figure 7.** The loss of VOC by bacteria ($L_{bact}$) as a function of the full ranges of $R_{chemaq}$ and $K_H$. $R_{chemgas}$ = $10^{-6}$ s$^{-1}$ and a) $R_{bact}$ = $10^8$ s$^{-1}$, b) $R_{bact}$ = $10^{-6}$ s$^{-1}$, c) $R_{bact}$ = $10^{-4}$ s$^{-1}$ and for $R_{bact}$ = $10^{-6}$ s$^{-1}$ with d) $R_{chemgas}$ = $10^{-5}$ s$^{-1}$ and e) $R_{chemgas}$ = $10^{-4}$ s$^{-1}$.





### 3.2.3 $L_t$ and $L_{bact}$ for NVOC

For NVOC, the analysis of trends in $L_t$ is simpler as it is the sum of $L_{bact}$ and $L_{chemaq}$ only. $L_t$ increases with
increasing $R_{chemaq}$, nearly independently of $R_{bact}$ up to a maximum value of ~1.1% (*Figure 8a*). Comparison
of $L_t$ to $L_{bact}$ (*Figure 8b*) shows that the consumption by bacteria is smaller by several orders of magnitude
($L_{bact,max} = 0.005\%$) and thus it is not a major contribution to the total loss. *Figures 3 and 8b* exhibit similar
shapes, i.e. after 600 seconds simulation time, the highest relative and absolute contributions are predicted
for NVOC with intermediate bacterial and low aqueous phase chemical activity. The trend in *Figure 8b* can
be also explained by the findings in *Figure 4*: After 600 seconds, the NVOC in the bacteria-containing
droplets are completely depleted for high $R_{bact}$ and thus both $fr_{bact}$ and $L_{bact}$ approach zero.

Since there is no exchange with the gas phase or other replenishment of NVOC into bacteria-containing
droplets, $L_{bact,maximum}$ of NVOC can be estimated: If one assumes in a first approximation that the mass
fraction of NVOC in the initial CCN population scales with the liquid water content associated with the
resulting droplets, one can conclude that at most 0.0065% (=LWC(bacteria-containing droplets/total LWC)
can be consumed.

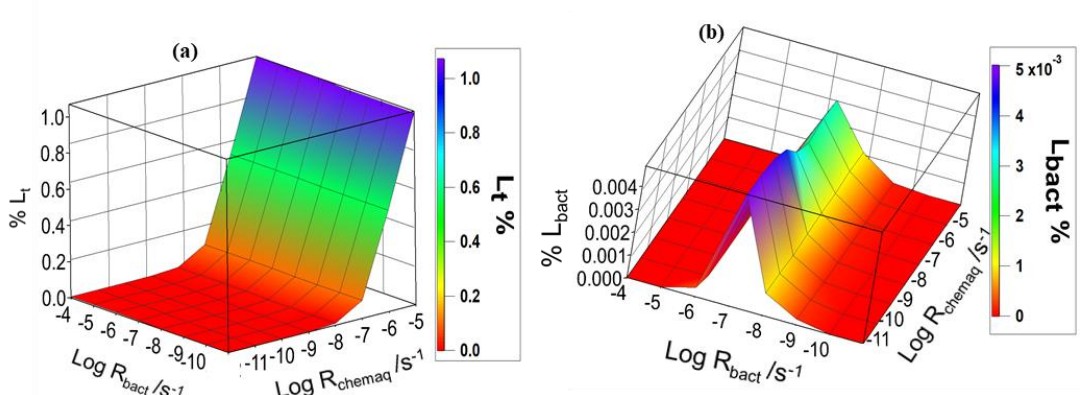

**Figure 8**. Predicted loss of NVOC as a function of chemical ($R_{chemaq}$) and microbial ($R_{bact}$) activity in the
aqueous phase, a) total loss ($L_t$), b) loss by microbial processes ($L_{bact}$).

## 3.3 Comparison of our model approach to simplified assumptions

### 3.3.1 Simplified model approaches

The results in *Figures 4,* demonstrate that the coupled equilibria between the gas phase and the droplets
with and without bacteria need to be considered, in order to correctly predict the organic sink strength by
bacterial processes in the multiphase system. Previous estimates of the role of biodegradation in the
multiphase system have been performed assuming simplified assumptions: (i) Biodegradation and chemical





rates were compared in a bulk aqueous phase (Vaïtilingom et al., 2013) or in a population of droplets with identical composition (Jaber et al., 2020). However, since the bacteria cell concentration (~0.01 cells cm$^{-3}$) in clouds in much smaller than the droplet (= 263 droplets cm$^{-3}$, in our model), this results in a ratio of ~4.10$^{-5}$ bacteria cells per droplet. (ii) A multiphase system is considered with only droplets that contain an intact bacteria cell (Fankhauser et al., 2019) resulting in a liquid water content of ~ 10$^{-11}$ vol/vol. In **Figure 9**, we schematically contrast these approaches. While it is clear, that none of the two approaches reflect the conditions as encountered in real clouds, we will analyze in the following the extent to which these simplified model approaches lead to similar results as predicted in our detailed model discussed so far (**Figure 1**).

### 3.3.2    Comparison of fr$_{bact}$, L$_t$ and L$_{bact}$ using simplified model approaches

For VOC, we compare fr$_{bact}$ and L$_{bact}$ for the three model appraoches for the full range of K$_H$ values, R$_{bact}$=10$^{-6}$ s$^{-1}$, R$_{chemaq}$=10$^{-7}$ s$^{-1}$ and R$_{chemgas}$=10$^{-6}$ s$^{-1}$ (**Figure 9 a, b**). For $10^2 \leq K_H$ [M atm$^{-1}$] $\leq 10^4$, only a small difference in fr$_{bact}$ and L$_{bact}$ is predicted for a given R$_{bact}$ because of the low organic fraction in the aqueous phase. For all K$_H$ values, fr$_{bact}$ is the highest for the low LWC model approach, because it only considers bacteria-containing droplets and, thus, neglects the aqueous phase chemistry in a majority of droplets; therefore, fr$_{chemaq}$ is significantly underestimated. Consequently, the competition between biodegradation and chemical degradation takes place only in the bacteria-containing droplets and so fr$_{bact}$ is the highest.

For intermediate $10^5 \leq$ K$_H$ [M atm$^{-1}$] $\leq 10^7$, L$_{bact}$ based on the bulk and low LWC models are higher than for the detailed model. For example, for K$_H$ = $10^6$ M atm$^{-1}$, L$_{bact}$ in the low LWC model is 4.3 and 12.7 higher than in the bulk and our detailed models. For the highest K$_H$ ($10^7 \leq$ K$_H$ [M atm$^{-1}$] $\leq 10^9$), the comparison of L$_{bact}$ for R$_{bact}$ = $10^{-6}$ s$^{-1}$ shows that both simplified model approaches overestimate L$_{bact}$ by up to several orders of magnitude (**Figure 9b**) for K$_H$ = $10^8$ M atm$^{-1}$ (L$_{bact, lowLWC}$ = $5.8 \times 10^{-1}$; L$_{bact, bulk}$ = $1.07 \times 10^{-1}$; L$_{bact, detailed}$ = $6.12 \times 10^{-4}$). For organics with intermediate or low solubility, the differences between the three models are smaller. Given that we identified biodegradation to be most important for organics (**Section 3.2.2**), we compare fr$_{bact}$ and L$_{bact}$ for K$_H$ = $10^5$ M atm$^{-1}$, R$_{chemgas}$= $10^{-6}$ s$^{-1}$, R$_{chemaq}$ = $10^{-7}$ s$^{-1}$ as a function of R$_{bact}$ (**Figure 9 c, d**). Similar to the results from the detailed model (**Section 3.1.2**), the low LWC approach leads to fr$_{bact}$ < 100% at t = 600 s because organics are efficiently consumed in the bacteria-containing droplets. However, for the bulk model, fr$_{bact}$ ~ 100% for the highest R$_{bact}$, even after 600 s, because bacterial processes take place in all droplets with a reduced efficiency as compared to the processes in the single droplet in the detailed model. Consequently, the concentration of organics in all droplets remains relatively high for extended time scales, even for the highest R$_{bact}$. **Figure 9d** shows the dependence of L$_{bact}$ between the three models on R$_{bact}$: for R$_{bact} \leq 10^{-6}$ s$^{-1}$, L$_{bact}$ is similar for the detailed and the bulk models (~5 $\times 10^{-4}$ %) whereas it is twice as high for the low LWC model (~1$\times 10^{-3}$ %). However, for the highest R$_{bact}$ ($10^{-4}$ s$^{-1}$), the




highest $L_{bact}$ is predicted by the bulk approach (~5%) which is more than two orders of magnitude higher than predicted from the detailed model (~0.2 %). The similarity of $L_{bact}$ between the detailed and the bulk models for $R_{bact}=10^{-6}$ s$^{-1}$ is due to the efficient replenishment of organics with intermediate $K_H$ (**Figure 5**), i.e. the amount of organics available to bacteria are similar. However, for the highest $R_{bact}$, $L_{bact}$ is much

higher for the bulk model because the consumption of the organics occurs without any kinetic limitations of the various transport processes that are only presented by the detailed model.

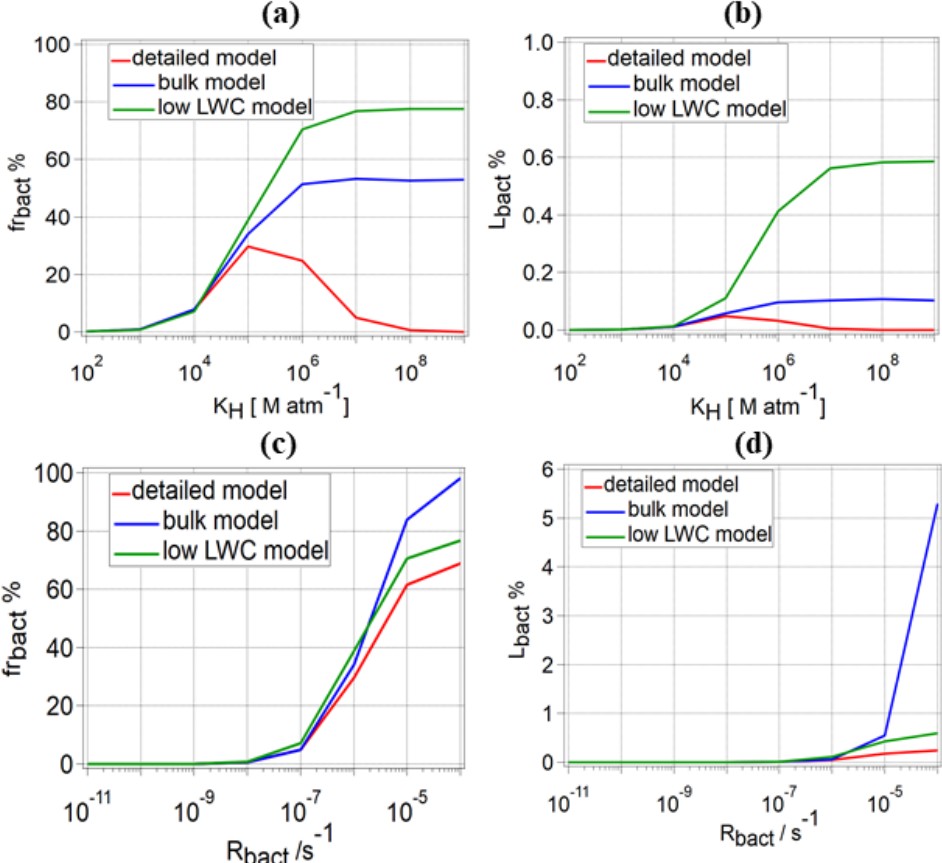

**Figure 9:** Comparison of (a) fr$_{bact}$, (b) $L_{bact}$ for volatile organics for three the different model approaches: detailed model (red line), bulk approach (blue line) and low LWC model (green line) for different $K_H$, $R_{chemaq}=10^{-7}$ s$^{-1}$, $R_{bact}=10^{-6}$ s$^{-1}$ and $R_{chemgas}=10^{-6}$ s$^{-1}$ ; (c) fr$_{bact}$ and (d) $L_{bact}$ for volatile organics for the same approaches for different $R_{bact}$ , one $K_H= 10^5$ M atm$^{-1}$, two $R_{chemaq}$: $10^{-7}$ s$^{-1}$ ,$R_{chemgas}=10^{-6}$ s$^{-1}$

For the NVOC, we compare fr$_{bact}$ and $L_{bact}$ for different $R_{bact}$ and two $R_{chemaq}$ ($10^{-11}$ and $10^{-7}$ s$^{-1}$) (**Figure S4 a, b**). For $R_{bact} = 10^{-8}$ s$^{-1}$ and $R_{chemaq} = 10^{-11}$ s$^{-1}$, fr$_{bact}$ is about 99% for all three approaches which is not the





case for highest $R_{chemaq}$ ($10^{-7}$ s$^{-1}$) (***Figure S4a***). However, for $R_{bact} = 10^{-4}$ s$^{-1}$, fr$_{bact}$ is ~0% (at t = 600 s) in the
       detailed model for both $R_{chemaq}$ for the reasons explained in ***Section 3.1.2*** whereas fr$_{bact}$ ~ 99% for the bulk
       model because of continuous bacteria activity in all droplets. Moreover, fr$_{bact}$ and $L_{bact}$ are always 100% for
       the low LWC model. The role of aqueous phase chemistry in the multiphase system is predicted to be
       negligible by the low LWC approach as it is equally reduced as the LWC, leading to the dominance of the
bacterial processes. If we compare $L_{bact}$ for the bulk (blue line in ***Figure S4b***) and the detailed models (red
       line), one finds an increasing discrepancy between the model results: For the lowest $R_{bact}$ considered here
       ($10^{-8}$ s$^{-1}$) and the two $R_{chemaq}$: $10^{-11}$ s$^{-1}$ and $10^{-7}$ s$^{-1}$ respectively, $L_{bact}$ for the bulk model is predicted to be ~
       9.3 and 11.6 higher than for the detailed model. For $R_{bact} = 10^{-6}$ s$^{-1}$ and the two $R_{chemaq}$, this factor increases
       to $2.5 \times 10^4$ and $1.3 \times 10^6$. For the highest $R_{bact} = 10^{-4}$ s$^{-1}$, $L_{bact}$ between the bulk and our detailed model after
600s of simulation cannot be reasonably compared as in the detailed model after < 100 seconds, the NVOC
       in the bacteria-containing droplets are completely depleted for high $R_{bact}$ and, thus, no further consumption
       occurs ($L_{bact}$ ~ 0%) (***Section 3.2.3***). In the bulk model, the organic consumption by bacteria is predicted to
       occur continuously as the bacteria could, in theory, consume all organics on 'infinite' time scales.
       Our model comparison shows that for VOC and NVOC, both the bulk and low-LWC models overestimate
the importance of biodegradation, i.e. both fr$_{bact}$ and $L_{bact}$. The biases are highest in the bulk approach for
       high $R_{bact}$ and high $K_H$. Also, the comparison shows that the bulk approach leads to wrong conclusions in
       terms of the importance of biodegradation as a function of the solubility of organics. Our model analysis
       emphasizes that a detailed model is needed in order to correctly represent the partitioning of VOC between
       the gas phase and the droplets with and without bacteria.

## 4.    Atmospheric implications: How important is biodegradation for organic compounds identified in cloud water?

### 4.1    $L_{bact}$ of organic cloud water constituents

Based on our model sensitivity studies discussed in the previous sections and the literature data for cloud
water organics summarized in ***Tables 1*** and ***2***, we assess the importance of biodegradation of these
compounds. The 3-d representations in ***Figure 10*** are the same as in Figures ***7 a-c*** (VOC) and ***Figure 8b***
       (NVOC), respectively, with slightly shifted viewing angles for better clarity. The added symbols correspond
       to $L_{bact}$ of the compounds as listed in ***Table 1*** for a single $R_{bact}$ in each figure panel. Given that $R_{chemgas}$ only
       differs by about two orders of magnitude for the organics in ***Table 1***, we present the results with $R_{chemgas} = 10^{-6}$ s$^{-1}$. The reasoning of the assumed single $R_{bact}$ is further discussed in ***Section 4.2***.
For the carboxylic acids, two values are shown, i.e. $K_{H(eff)}$ and $R_{chemaq}$ at pH = 3 and 6 (***Table 3***). $R_{chemaq}$ at
       the two pH values are calculated based rate constants $k_{chemaq}$ as a function of the proportion of the acidic and
       anionic forms depending on pH and pK$_a$ and the rate constants of the OH reactions for the acid and anion.





Similarly, rate constants for other (e.g. $NO_3$) reactions could be derived using data summarized in **Table 1**.
As discussed in **Section 2.2.2** and by Vaïtilingom et al. (2013), the bacterial activity does not show any

systematic difference in this pH range; therefore, we only show one $R_{bact}$ value for each acid. For the lowest
$R_{bact}$ ($10^{-8}$ $s^{-1}$), the physicochemical properties ($K_{H(eff)}$, $R_{chemaq}$) of the VOC (acetic acid/acetate, formic
acid/formate, formaldehyde, catechol, phenol and methanol) are in the range where $L_{bact}$ has its minimum
values $\sim 10^{-6} \leq L_{bact}$ [%] $\leq 10^{-3}$ (**Figure 10a**).

For higher $R_{bact}$ ($10^{-6}$ $s^{-1}$, **Figure 10b**), which implies higher cell concentration in the cloud, highly soluble
compounds, such as acetate ($a_2$) and catechol (C), are in the range where $L_{bact}$ shows a maximum ($\sim 0.04 \leq$
$L_{bact}$ [%] $\leq 0.06$) and less soluble compounds such as formic acid ($F_1$), formate ($F_2$) and formaldehyde (Fo)
($10^4 < K_{H(eff)}$ [M atm$^{-1}$] $< 10^6$) are in the area where $\sim 0.02 \leq L_{bact}$ [%] $\leq 0.04$. Less soluble compounds ($K_{H(eff)}$
$< 10^3$ M atm$^{-1}$), such as acetic acid ($a_1$), phenol (P) and methanol (M) are in the area of much lower $L_{bact}$
($\sim 10^{-5} \leq L_{bact}$ [%] $\leq 10^{-3}$). As discussed in **Section 3.2**, the maximum value of $L_{bact}$ increases with higher $R_{bact}$
and is predicted for compounds with moderate $K_{H(eff)}$ (**Figure 5**). Therefore, at the highest assumed $R_{bact}$ ($10^{-4}$ $s^{-1}$), the less soluble compounds $F_1$, Fo and $a_1$, are in the range where $L_{bact}$ has its maximum values ($0.3 <$
$L_{bact}$ [%] $< 0.4$; **Figure 11c**).

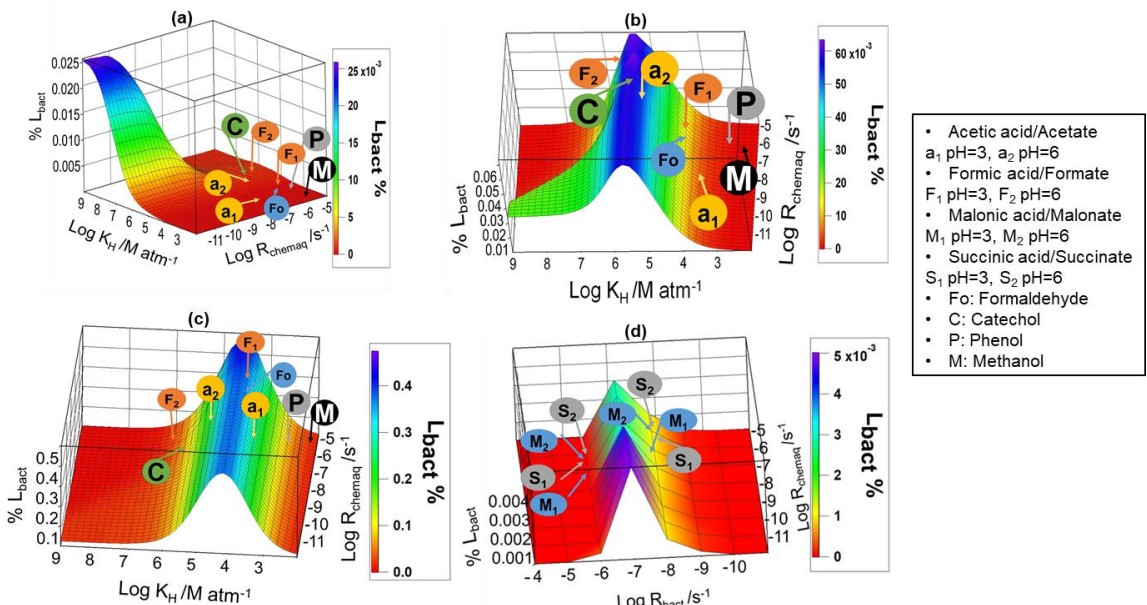

**Figure 10:** $L_{bact}$ for VOC for a specific combination of $K_H$ and $R_{chemaq}$ (a,b,c) and for NVOC (d) for a specific combination of $R_{bact}$ and $R_{chemaq}$ at 2 pH (3 and 6) for the acidic compounds.

**Figure 10d** repeats the full parameter ranges of $R_{bact}$ and $R_{chemaq}$. We discuss $L_{bact}$ for the NVOC (malonic
acid ($M_1$), malonate ($M_2$), succinic acid ($S_1$), succinate ($S_2$)) for the same $R_{bact}$ as in **Figure 10 a-c**: The



maximum consumption of NVOC by bacteria corresponds to the fraction of the organics present in bacteria-containing droplets (~0.0065% in our simulations).  Therefore, the predicted $L_{bact}$ [%] are very similar for all compounds considered here (malonate/malonic acid and succinate/succinic acid) and does not differ independently of $R_{bact}$.

### 4.2    Dependence of $L_{bact}$ on bacteria cell concentration in cloud water

According to Eq-3, $R_{bact}$ is the product of the biodegradation rate constant $k_{bact}$ [L Cell$^{-1}$ s$^{-1}$] (**Table 1**) and the cell concentration $C_{cell}$ [Cell L$^{-1}$]. Based on the three $R_{bact}$ values and the $k_{bact}$ data for the various organics, we calculate $C_{cell,theoretical}$ that would be needed to reach these $R_{bact}$ :

$$C_{cell,theoretical} \ [Cell \ L^{-1}] = \frac{R_{bact} \ [s^{-1}]}{k_{bact} \ [L \ Cell^{-1} \ s^{-1}]} \qquad \text{Eq-14}$$

The resulting $C_{cell,theoretical}$, [Cell L$^{-1}$] are listed in **Table 3**, together with $k_{bact}$ [L Cell$^{-1}$ s$^{-1}$]. Among the VOC,
$C_{cell,theoretical}$ is predicted to be ~$10^2 \leq C_{cell,theoretical}$ [Cell L$^{-1}$] $\leq 10^7$ for acetic acid, formic acid and phenol while it is much higher (~$10^7 \leq C_{cell,theoretical}$ [Cell L$^{-1}$] $\leq 10^{15}$) for catechol, formaldehyde and methanol that have much lower $k_{bact}$ than the former compounds. Typical bacteria cell concentration in cloud water are in the range of $10^6$ - $10^8$ Cell L$^{-1}$ (Amato et al., 2007b). Comparing this range to $C_{cell,theoretical}$ reveals that for several compounds $C_{cell,theoretical}$ falls into a realistic range (grey shaded cells in **Table 3**): for ambient cell
concentrations in cloud water, malonic acid/malonate, succinic acid/succinate, acetic acid/acetate, formic acid/formate, cathecol and phenol might fall into a range where the maximum consumption of organics by biodegradation can be expected, while for formaldehyde and methanol $C_{cell,theoretical}$ is unrealistic ($\geq 10^9$ cells L$^{-1}$) based on the available studies of bacteria cell concentrations in cloud water.

**Table 3.** $k_{bact}$ [L Cell$^{-1}$ s$^{-1}$] calculated from literature data, theoretical $C_{cell}$ [Cell L$^{-1}$] based on Eq-14 and the ratio $L_{bact}/L_t$ for three $R_{bact}$ ($10^{-8}$ s$^{-1}$, $10^{-6}$ s$^{-1}$ and $10^{-4}$ s$^{-1}$) for VOC and NVOC.

| Organic compound | $k_{bact}$ [L Cell$^{-1}$ s$^{-1}$] | $C_{cell,theoretical}$ [Cell L$^{-1}$] based on Eq-14 | | | $L_{bact,max}$ [%] | | |
|---|---|---|---|---|---|---|---|
| | | $R_{bact}$ | | | $R_{bact}$ | | |
| | | $10^{-8}$ s$^{-1}$ | $10^{-6}$ s$^{-1}$ | $10^{-4}$ s$^{-1}$ | $10^{-8}$ s$^{-1}$ | $10^{-6}$ s$^{-1}$ | $10^{-4}$ s$^{-1}$ |
| **NVOC** | | | | | | | |
| Malonic acid **M₁** Malonate **M₂** | $9.0 \times 10^{-13}$ | $3.3 \times 10^4$ | $3.3 \times 10^6$ | $1.1 \times 10^8$ | $\leq 0.0065\%$ (= LWC$_{bact}$ / / LWC$_{total}$ | | |
| Succinic acid **S₁,** Succinate **S₂** | $3.0 \times 10^{-13}$ | $3.3 \times 10^4$ | $3.3 \times 10^6$ | $3.3 \times 10^8$ | | | |
| **VOC** | | | | | | | |
| | | | | | $L_{bact}/L_t$ | $L_{bact}/L_t$ | $L_{bact}/L_t$ |
| Acetic acid **a₁** Acetate **a₂** | $4.7 \times 10^{-11}$ | $2.1 \times 10^2$ $7.1 \times 10^3$ | $2.1 \times 10^4$ $7.1 \times 10^5$ | $2.1 \times 10^6$ | $5.8 \times 10^{-4}$ $8.0 \times 10^{-2}$ | $5.8 \times 10^{-2}$ $3.3 \times 10^{-1}$ | $7.0 \times 10^{-1}$ $6.7 \times 10^{-1}$ |





| | | | | | | | |
|---|---|---|---|---|---|---|---|
| Formic acid $\mathbf{F_1}$ | $1.4\times10^{-12}$ | $7.1\times10^3$ | $7.1\times10^5$ | $7.1\times10^7$ | $6.2\times10^{-4}$ | $7.4\times10^{-2}$ | $6.5\times10^{-1}$ |
| Formate $\mathbf{F_2}$ | | $1.4\times10^9$ | $1.4\times10^{11}$ | | $1.2\times10^{-3}$ | $2.6\times10^{-2}$ | $3.4\times10^{-2}$ |
| Formaldehyde $\mathbf{Fo}$ | $7.0\times10^{-18}$ | $3.6\times10^6$ | $3.6\times10^8$ | $1.4\times10^{13}$ | $8.4\times10^{-4}$ | $7.4\times10^{-2}$ | $6.5\times10^{-1}$ |
| Catechol $\mathbf{C}$ | $2.7\times10^{-15}$ | $3.0\times10^2$ | $3.0\times10^4$ | $3.6\times10^{10}$ | $3.5\times10^{-3}$ | $1.2\times10^{-1}$ | $2.0\times10^{-1}$ |
| Phenol $\mathbf{P}$ | $3.3\times10^{-11}$ | $1.2\times10^{11}$ | $1.2\times10^{13}$ | $3.0\times10^6$ | $5.4\times10^{-5}$ | $5.4\times10^{-3}$ | $3.2\times10^{-1}$ |
| Methanol $\mathbf{M}$ | $8.1\times10^{-20}$ | $1.2\times10^{11}$ | $1.2\times10^{13}$ | $1.2\times10^{15}$ | $2.3\times10^{-5}$ | $2.3\times10^{-3}$ | $2.4\times10^{-1}$ |

## 4.3 For which organics is biodegradation an efficient sink in the atmosphere?

The maximum value of $L_{bact}$ is ~ 0.7% but we should keep in mind that also the predicted $L_t$ is not higher than a few percent (***Figure 6 a-c***) as we restrict our simulations to the time scale of approximate one cloud cycle. However, in a relative sense, our results allow us to compare the importance of biodegradation to chemical loss for the compounds shown in ***Figure 10***. To quantify the contribution of bacteria in this total consumption ($L_t$) we list the ratio $L_{bact}/L_t$ in ***Table 3***. Our results clearly show that biodegradation might add significantly to the loss of formic acid/formate ($L_{bact}/L_t$= 0.65) and acetic acid/acetate ($L_{bact}/L_t$ =0.70) at cell concentrations of ~$7\cdot10^7$ and $2\cdot10^6$ Cells $L^{-1}$, respectively. These acids contribute on average ~68% to the total dissolved carbon in cloud and fog water (Herckes et al., 2013). Their removal by dry and wet deposition are considered the major loss processes in the atmosphere (Khare et al., 1999). However, several studies also suggested that the oxidation for formic acid/formate in cloud water may be a net sink (Jacob, 1986). Measurements during different seasons in the Amazon showed that indeed formic and acetic acids have stronger sinks during the wet season (Herlihy et al., 1987). While wet deposition was described to contribute to a large extent the observed removal, chemical and possibly bacterial processes were suggested to act as additional sinks. While the latter was regarded as being inefficient due to long incubation time as observed in lab experiments, more recent experiments suggest that such incubation times are likely not occurring in the atmosphere where bacteria cells are continuously exposed to water and substrates.

We conclude that biodegradation of these major cloud water organics may be a significant sink under ambient conditions, possibly even comparable to the loss by chemical reactions. While phenol is not a major contributor to the WSOC content in cloud water (5-95 nM ) (Jaber et al., 2020) as compared to 10μm (Ervens et al., 2003b) (Löflund et al., 2002) (Sun et al., 2016) for formic and acetic acids respectively, its degradation processes in the atmosphere might be of interest due to its toxic properties. Overall, our results are in agreement with previous findings that neither chemical processes nor biodegradation are major WSOC losses as compared to deposition (Ervens and Amato, 2020) (Fankhauser et al., 2019). However, in order to comprehensively describe the loss processes and time scales of organic degradation and residence time scales in the atmosphere, both chemical and biological processes should be considered. Hence, we suggest that biological processes of organics with properties similar to those of formic and acetic acids and phenol



($!10^3 < K_{H,eff}$ [M atm$^{-1}$] $< \sim\!10^6$, $k_{bact} = 10^{-11}$) should be included in atmospheric multiphase models.

The importance of biodegradation of NVOC is limited by the number fraction of cloud droplets that contain bacteria. Malonate/malonic acid and succinate/succinic acid contribute on average to < 5% to the total organic aerosol mass in ambient particles, e.g.(Kawamura and Ikushima, 1993) (Fu et al., 2013),. Their loss by chemical and biological processes will neither affect the total carbon budget to a large extent nor the

budget of the individual compounds. These conclusions can be generalized for other NVOC aerosol constituents, for which biodegradation has been suggested to occur in the atmosphere. Our assumption of static cloud droplets in the box model is certainly a simplified representation of cloud microphysics. Droplets might experience collision/coalescence in clouds leading to mixing of the cloud water constituents in the resulting larger droplets. However, such processes are unlikely to add significantly to the loss of NVOC by

bacteria due to (i) the number small fraction of bacteria-containing droplets (0.001 – 0.0001) and (ii) the limited atmospheric residence time of large droplets which are efficiently removed by precipitation as a function of drop size (Beard and Ochs, 1984).

## 5.    Summary and Conclusions

Our model sensitivity study is the first comprehensive analysis of the importance of biodegradation of

organics by bacteria in the atmospheric multiphase system in comparison to chemical loss for wide ranges of chemical and biodegradation kinetic data. We use a box model with drop-size-resolved aqueous phase chemistry and additional biological processes that only occur in a small number of cloud droplets, in agreement with ambient ratios of cell and droplet concentrations. We neglect the fact that bacteria cells may form agglomerates in the atmosphere and consequently, there might more than one cell per droplet. This

effects could be included in our model by multiplying the biological activity in the respective droplets with the number of cells.

We compare the predicted loss rates of chemical processes in both phases to those of biological processes in the aqueous phase only. In addition to presenting the relative loss rates ($fr_{bact}$, $fr_{chem}$) as in previous studies (Jaber et al., 2020) (Vaïtilingom et al., 2010), we discuss the relative amounts of organics ($L_t$) consumed by

chemical ($L_{chem}$) and biological processes ($L_{bact}$). We find that the relative loss rate of organics by biological processes ($fr_{bact}$) is generally higher for VOC than for NVOC. However, the total loss of the organics ($L_t$) is predicted not to reach any value higher than ~12% because our simulations were restricted to a period of 600 s (~ drop lifetime within one cloud cycle); it would be higher if the total particle processing time during multiple cloud cycles in the atmosphere were considered. The contribution of bacteria ($L_{bact}$) to the total loss

is predicted to be highest for VOC with intermediate solubility ($\sim\!10^4 \le K_H$ [M atm$^{-1}$] $\le \sim\!10^6$). This can be explained by the replenishment of VOC in the bacteria-containing droplets upon uptake from the gas phase and evaporation from the bacteria-free droplets, in which less efficient consumption of the organics occurs.



Less soluble organics ($K_H < 10^4$ M atm$^{-1}$) that partition to a smaller extent (< 1%) to the aqueous phase are mostly consumed by chemical processes in the gas phase; more soluble compounds ($K_H > 10^6$ M atm$^{-1}$) are

predominately partitioned to the aqueous phase and, thus, the evaporation to the gas phase and consequently the redistribution from the bacteria-free to the bacteria-containing droplets is kinetically more limited and thus less efficient. The ratio of the consumption of VOC by bacteria to the total loss ($L_{bact}/L_t$) might be as high as 0.7 for high biological activity and cell concentrations (~ $10^8$ cells L$^{-1}$). These values suggest that biological processes might add significantly (>70%) to the loss processes in the atmospheric multiphase

system for organics with intermediate solubility such as formic acid/formate, acetic acid/acetate or phenol. For NVOC, the amount of organics consumed by bacteria is restricted to the fraction of the organic dissolved in bacteria-containing droplets (~ 0.001%) as no efficient replenishment from the gas phase or from the other droplets occurs. Thus, biodegradation of NVOC does not significantly affect their atmospheric budget. In addition to our detailed model with a realistic bacteria cell distribution within a cloud droplet population,

we apply simpler model approaches: (i) Similar to many chemical model studies, bacteria cells are distributed equally in all droplets ('bulk approach') resulting in cell concentrations of $10^{-5}$ cells per droplet, clearly an unphysical assumption for intact bacteria cells. (ii) A multiphase system with only cloud droplets which contain bacteria resulting in a liquid water content of ~$10^{-11}$ vol/vol as compared to ~$10^{-7}$ vol/vol in clouds ('low LWC approach'). Comparing $L_{bact}$ predicted from these approaches to results of our detailed

model shows that all approaches agree in predicting $L_{bact}$ for organics of low solubility ($K_{H(eff)} < 10^4$ M atm$^{-1}$). However, for such species the importance of biodegradation is low due to their inefficient partitioning to the aqueous phase. The bulk approach increasingly overestimates $L_{bact}$ of organics with higher $K_H$; the greatest discrepancy is predicted for highly soluble compounds ($K_{H,eff} > 10^6$ M atm$^{-1}$) as the bulk approach does not take into account the kinetic limitation due to the organic redistribution between the bacteria-free

and bacteria-containing droplets. As the bulk approach implies organics in all droplets, it does not allow limiting on the organic consumption by biodegradation. Predictions of the relative role of biodegradation as compared to chemical processes by the 'low LWC approach' are biased high because the loss due to aqueous phase processes is only considered in an unrealistically small fraction of droplets.

The current data sets for microbial rates of organic compounds are limited to very few compounds. Our

model sensitivity study shows that biodegradation by bacteria in clouds is most efficient for compounds with intermediate (effective) Henry's law constants (~$10^4$ M atm$^{-1}$ < $K_{H(eff)} < 10^6$ M atm$^{-1}$) as found for common cloud water constituents such as formic acid/formate and acetic acid/acetate but also for less abundant species such as phenol, largely dependent of their chemical reactivity. Our framework allows to estimate the potential importance of biodegradation of organics, in comparison to chemical processes. It

also gives guidance to future lab and model studies to further explore the role of biodegradation of specific organics in the multiphase system.



**Data availability:** Additional model results can be obtained from the corresponding author upon request.

**Author contributions:** AK and BE designed and carried out the model studies and wrote the manuscript.
MZ, PA, AMD contributed by fruitful discussions and gave constructive feedback on the manuscript.

**Competing interests:** The authors declare no competing interests.

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
