# Peer review of "Figure S1. Relative contributions of the aqueous phase chemical processes to the total loss rate of organics ( $fr_{chemaq}$ ) for three $R_{bact}$ : (a) $10^{-8}$ , (b) $10^{-6}$ and (c) $10^{-4} \text{ s}^{-1}$ and $R_{chemgas}$ ( $10^{-6} \text{ s}^{-1}$ ) and the full ranges of $R_{"

_Atmospheric Chemistry and Physics, 2020_

## Referee Comment (RC1) · Anonymous Referee #1 · 12 Oct 2020

This work investigated biodegradation of bacteria in clouds with aqueous-phase modeling, demonstrating that biodegradation might be most efficient for volatile organic compounds with intermediate solubility. Most of current atmospheric models do not treat biodegradation by bacteria and this study suggests its potential importance for some cases. I found that the manuscript is well-written: motivation is clear, the methods are mostly described thoroughly, results and discussions are discussed well with many sensitivity studies, model analysis is comprehensive, and conclusions are justified. With that said, I am happy to support publication in ACP after the below comments are considered and implemented.

[Figure]

- Throughout the manuscript I found terminology of VOC vs. NVOC somewhat confusing. It seems that you categorize compounds based on solubility ($K\_H$) but not with vapor pressures (no vapor pressures are given for treated compounds). For organic aerosol research, VOC is often used for compounds mostly in the gas phase, while NVOC for compounds mostly in the condensed phase. As you know, compounds which may exist both phases comparably are termed semi-volatile organic compounds (SVOC), so most compounds treated in this study appear better termed as SVOC. Some more clarifications with clear definition would help improve presentation quality.

- It was not clear to me how you treat phase transfer in your model. It is just simply stated that it is based on the resistance model (P3), but unclear how exactly you treat (Fuchs-Sutugin correction?) and what values you use for critical parameters such as gas diffusivity and mass accommodation coefficient.

- I wonder what is cell viability and how long bacteria would survive in cloud droplets. Would this be something should be discussed and considered in the model with some sensitivity studies? Cloud droplets contain oxidants (e.g., OH, $H_2O_2$, etc.), so I wonder if they experience oxidative stress and eventually degraded? In other words, would k\_bact be time-dependent?

---

## Referee Comment (RC2) · Anonymous Referee #2 · 17 Nov 2020

The authors have investigated via a model the degradation of dissolved organic compounds by bacteria in cloud droplets. The results suggest that this could be appreciable for some organic compounds with intermediate solubility and high bacterial activity. The manuscript is generally well-written and can be recommended for publication in ACP after the minor comments and suggestions below are addressed.

1) Section 2.1: The model description is too concise and could be improved a bit. Which 26 species are transferred between the gas and aqueous phases? It would be worth showing the coupled mass transfer ODEs with Schwartz's treatment.

2) Is the size class same as the size bin? On line 101, it is stated that the 5 um < $D_{droplet}$ < 20 um, but then one droplet size class has $D_{droplet}$ = 20 um. Should the range be changed to 5 um < $D_{droplet}$ ≤ 20 um? Also, why is only the last size class allowed to have bacteria cells?

3) Line 105-106: Which organic compound? Is the model run separately for each individual compound? What other inorganic species were considered in the model? What was the cloud water pH in the model simulations discussed in section 3?

4) While the degradation rate constant remains constant between pH values 5 and 8, could it change at lower pH?

5) Figures 2, 3, 6, 7, 8, 10 are presently displayed as three-dimensional plots, which I found a little difficult to read as parts of the plots are obscured by curved surfaces. Since the Z-axis and colors represent the same dimension, I suggest replotting these figures as two-dimensional color contour plots. This would greatly improve the quality and readability of the figures.

6) Table 2: There's an extra multiplication symbol in the 5 C Experimental rate column on 12th row.

---

## Author Comment (AC1) · 18 Dec 2020

We thank the referee for their constructive and positive comments on our manuscript. We respond to all comments in detail below. Our responses are in black, manuscript text is in italic with new/modified text is marked in blue. The line numbers cited here refer to those in the clean manuscript version.

**Referee comment**: - Throughout the manuscript I found terminology of VOC vs. NVOC somewhat confusing. It seems that you categorize compounds based on solubility ($K_H$) but not with vapor pressures (no vapor pressures are given for treated compounds). For organic aerosol research, VOC is often used for compounds mostly in the gas phase, while NVOC for compounds mostly in the condensed phase. As you know, compounds which may exist both phases comparably are termed semi-volatile organic compounds (SVOC), so most compounds treated in this study appear better termed as SVOC. Some more clarifications with clear definition would help improve presentation quality.

**Author response**: We thank the referee for pointing out this inconsistency in our terminology. We agree that this was misleading as volatility is not the main criterion for the partitioning of species between the gas and aqueous phases, respectively. The categorization into VOC, NVOC and SVOC is mostly relevant for the partitioning of compounds between the gas and a condensed phase, such as present in aerosol. However, as the volatility does not (necessarily) correlate with water-solubility (e.g. alkanes and small aldehydes exhibit similar vapor pressures but vastly different Henry's law constants), these categories do not seem useful for the categorization of water-soluble compounds into the aqueous phase of cloud droplets.

Therefore, we categorize the various compounds now exclusively based on their solubility in water, i.e. on their Henry's law constant. We distinguish now four compound groups and we will add the following text at the end of the section 2.1.:

[revised manuscript text omitted]

*d) $L_{bact}$ for CCN-derived organics (Table 1). The contour plot is identical to Figure 8 b) .Note that $R_{chemaq}$ and $R_{bact}$ for succinate (pH = 6) is nearly identical to the values for succinic acid (pH = 3) and were therefore omitted from the figure.*

*- Header of Section 3.1.1:  $fr_{bact}$ for*  *: $fr_{bact}$ for water-soluble organic gases*

*- Header of section 3.1.2: $fr_{bact}$ of*  *and comparison to $fr_{bact}$ of*  *: $fr_{bact}$ of CCN-derived compounds and comparison to $fr_{bact}$ of water-soluble organic gases.*

*- Header of Section 3.2.1:  $L_t$ for*  *: $L_t$ for water-soluble organic gases*

*- Header od Section 3.2.2.: $L_{bact}$ for*  *: $L_{bact}$ for water-soluble organic gases*

*- Header od Section 3.2.3: $L_t$ and $L_{bact}$ for*  *: $L_t$ and $L_{bact}$ for CCN-derived compounds*

**Referee comment**: - It was not clear to me how you treat phase transfer in your model. It is just simply stated that it is based on the resistance model (P3), but unclear how exactly you treat (Fuchs-Sutugin correction?) and what values you use for critical parameters such as gas diffusivity and mass accommodation coefficient.

**Author response:** Since also the other referee asked for more details on the model, we added three tables into the supplement with details of our aqueous phase chemical mechanism (Table S1), phase transfer parameters (Table S2) and initial mixing ratios and concentrations (Table S3). Please find all tables at the end of this response. In addition, we added the following set of equations for the description of the phase transfer to the supplement:

*The mass transfer coefficient $k_{mt}$ [$s^{-1}$] can be expressed as (Seinfeld and Pandis, 1998)*

$$k_{mt} = \left[ \frac{r_d^2}{3D_g} + \frac{r_d}{3\alpha} \sqrt{\frac{2\pi M_g}{RT}} \right]^{-1} \qquad \text{(Eq-R1)}$$

*Whereas*

*$r_d$  = cloud droplet radius [cm]*

*$D_g$ = gas phase diffusion coefficient [cm $s^{-1}$]*

*$\alpha$ = mass accommodation coefficient (dimensionless)*

*$M_g$ = molecular weight of gas [g mol$^{-1}$]*

*$R$ = constant for ideal gases (8.314 ·10$^7$ erg mol$^{-1}$ K$^{-1}$)*

*$T$ = temperature [K]*

*The mass transfer coefficient is then applied to determine the sink and source terms of soluble species:*

$$\frac{dc_g}{dt} = k_{mt} \cdot LWC \cdot \left[ \frac{c_{aq}}{LWC \cdot K_{H(eff)} \cdot R' \cdot T} - c_g \right] + (P_{gaschem} - L_{gaschem}) \qquad \text{(Eq-R2)}$$

$$\frac{dc_{aq}}{dt} = k_{mt} \cdot LWC \cdot \left[ c_g - \frac{c_{aq}}{LWC \cdot K_{H(eff)} \cdot R' \cdot T} \right] + (P_{aqchem} - L_{aqchem}) \qquad \text{(Eq-R3)}$$

*whereas both the gas and the aqueous phase concentrations have units of mol g(air)$^{-1}$; LWC is the liquid water content in g/m$^3$, $K_{H(eff)}$ is the effective Henry's law constant in M atm$^{-1}$, R' is the ideal-gas constant (8.314.10$^7$ erg mol$^{-1}$ K$^{-1}$), T is the temperature [K] and P and L are the rates of the chemical production and loss reactions in the gas and aqueous phases, respectively.*

In brief, we do not use the Fuchs and Sutugin correction as this applies for the transition regime of transport of gas molecules towards particles, i.e. when the mean free path length of molecules ($\lambda$) is approximately equal to the particle diameter (Dp), i.e. when the Knudsen number is Kn ~ 1. The transport of gas molecules towards droplets with typical diameters of 10 μm or larger falls into the continuum regime. For this regime, the various processes that affect the transport of gas molecules into droplets were expressed in terms of 'resistances' by (Schwartz, 1986). These processes generally include gas phase diffusion, interfacial mass transport, aqueous phase diffusion and aqueous phase chemical reaction. In previous sensitivity studies, we have shown that aqueous phase diffusion is usually sufficiently fast compared to the interfacial mass transfer chemical reaction in cloud droplets so its term can be neglected  (Ervens et al., 2003b, 2014).

We add the following text at the beginning of Section 2.1:

*We use a multiphase box model with detailed gas and aqueous phase chemistry (75 species, 44 gas phase reactions, 31 aqueous reactions). The chemical aqueous phase mechanism with rate constants is listed in **Table S1**. The chemical gas phase mechanism is based on the NCAR Master mechanism (Aumont et al., 2000; Madronich and Calvert, 1989). The two phases are coupled by 26 phase transfer processes which is described kinetically based on the resistance model by Schwartz (1986). The parameters describing the phase transfer of the soluble species are presented in **Table S2**. In addition, the initial mixing ratios of gas phase species are included in **Table S3**.*

*The equations for the mass transfer coefficient k$_{mt}$ and the differential equations for the aqueous and gas phase concentrations were added to the supplement **(Eq-R1-R3;** Section 'Description of phase transfer) (Seinfeld and Pandis, 1998).*

**Referee Comment**: - I wonder what is cell viability and how long bacteria would survive in cloud droplets. Would this be something should be discussed and considered in the model with some sensitivity studies?

**Author response:** We thank the referee for pointing to the uncertainties associated with the biological activity of bacteria in clouds vs lab studies. We did not perform any sensitivity studies on the viability and survival time of bacteria cells in cloud water due to the large uncertainties associated with their estimates. Viable cells have been isolated from aerosol samples (Bovallius et al., 1978; Lighthart, 1997). Given that

living bacteria cells have been found in the atmosphere at many different places around the world, not only in clouds but also in particles, it seems reasonable to assume that cells survive for several hours or days (Fahlgren et al., 2010; Lighthart and Shaffer, 1994). Metabolic activity in cloud water was confirmed by assaying ATP (Adenosine 5'triphosphate) in cloud water samples (Amato et al., 2007b). ATP is considered as a key molecule of the energetic cell metabolism. We also would like to refer to a discussion on uncertainties in determining viable, cultivable and living cell fractions in our previous publication (Ervens and Amato, 2020). We will add some information on this topic throughout the manuscript; see our response to the next comment.

**Referee Comment**: Cloud droplets contain oxidants (e.g., OH , $H_2O_2$, etc.), so I wonder if they experience oxidative stress and eventually degraded? In other words, would $k_{bact}$ be time-dependent?

**Author response:** The referee is correct that high oxidant concentrations in the atmosphere and in particular in cloud water lead to stressful conditions for the bacteria cells, in addition to other factors such as UV exposure, low pH and temperature (Sattler et al., 2001). In addition, other effects such as substrate limitation may impact $k_{bact}$. Thus, given these different aspects, we respond to this referee comment in three parts:

a) Previous experiments in our research group have shown that cloud microorganisms can resist the oxidative stress due to the presence of reactive oxygen species, such as OH. The ADP/ATP ratio was monitored during such experiments which showed that even after exposure to oxidants over several hours, biodegradation rates were not significantly affected (Vaïtilingom et al., 2013). Based on these experiments, we can conclude that the presence of oxidants at typical levels as found in cloud water probably does not significantly affect $k_{bact}$ during the cloud cycles.

We add to the introduction (l.48, 52):

*The atmosphere is a stressful environment for microorganisms (low temperature, UV exposure, acidic pH, quick hydration/drying cycles and the presence of oxidants such as OH, $H_2O_2$) which might limit the survival time of cells in the atmosphere. Marker compounds such as adenosine 5'-triphosphate (ATP) (Amato et al., 2007c) , rRNA 50 (Krumins et al., 2014) or mRNA (Amato et al., 2019) have been used to demonstrate metabolic activity in the atmosphere. The ADP/ATP ratio was monitored during such experiments which showed that even after exposure to oxidants over several hours, biodegradation rates were not significantly affected (Vaïtilingom et al., 2013).*

b) Recent experiments of airborne bacteria suggested that metabolic rates are a function of substrate availability (Krumins et al., 2014). Cloud water can be considered an oligotrophic medium. As the biodegradation experiments by (Husárová et al., 2011; Vaïtilingom et al., 2010, 2013) were performed in real or artificial cloud water with realistic solute concentrations, it is assumed that biodegradation occurs at the same rates in clouds. In the case of the CCN-derived water-soluble substrates that are not replenished within a cloud cycle, such substrate limitation might occur. However, as we do not have any data on the change of biodegradation rates as a function of substrate limitation, we do not perform any additional sensitivity studies. Our conclusions would not change, namely that the amount of CCN-derived organics that can be consumed by biodegradation is constrained by the fraction of cloud droplets that contain bacteria cells. Soluble gases are continuously replenished in the cloud water due to uptake processes from the gas phase (cf our discussion of Figure 5 in Section 3.1.2). If the availability of substrates with intermediate water-solubility decreases, $k_{bact}$ may decrease as well. However, as a response, the substrate concentration might increase due to a relatively more efficient replenishment from the gas phase, which may, in turn, lead to an increase in $k_{bact}$ again. Currently, no data is available on the absolute changes in $k_{bact}$ at different levels of substrate availability and/or on the time scales during which $k_{bact}$ might adjust to such conditions. Therefore, we do not think sensitivity studies on would be useful at this point due to the lack of parameter

constraints. We will add some discussion of such potential feedbacks in Section 4, highlighting the need of future experiments.

We add in the introduction (line 66):

*The biodegradation rate likely depends on the availability of the substrate in the aqueous phase. In the case of soluble substrates, it can be expected that uptake of soluble substrates from the gas phase leads to a continuous replenishment of the organics.*

We add in Section 5 (l. 591):

*Recent experiments of airborne bacteria suggested that metabolic rates are a function of substrate availability in oligotrophic media such as cloud droplets (Krumins et al., 2014). In the case of CCN-derived organics or those that are inefficiently replenished by uptake from the gas phase, substrate limitation might result in lower metabolic rates. Currently, no data is available on the absolute changes in $k_{bact}$ at different levels of substrate availability and/or on the time scales during which $k_{bact}$ might adjust to such conditions.*

c) The initial biodegradation rates of acetic and formic acid in lab experiments were observed to be very small if detectable at all (Herlihy et al., 1987). Only after incubation of several hours or days, biodegradation was observed. However, such incubation periods are likely due to the adjustment of the bacteria cells to the ambient conditions in the lab experiments. As bacteria cells are continuously exposed to water, solutes, substrates etc in the atmosphere, such incubation periods are unlikely to occur in clouds. Indeed, experiments in our lab performed in real cloud water did not exhibit such lag periods (e.g. (Amato et al., 2007a; Vaïtilingom et al., 2010, 2013) which supports this hypothesis. While we touched briefly on this topic in Section 4, we will extend this discussion in the revised manuscript adding the latter references.

While some text on the incubation effects was already included in Section 4.3, the following references will be added there (line 535):

*While the latter was regarded as being inefficient due to long incubation time as observed in lab experiments, more recent experiments suggest that such incubation times are likely not occurring in the atmosphere where bacteria cells are continuously exposed to water and substrates (Amato et al., 2007a; Vaïtilingom et al., 2010, 2013).*

**Table S1**. Aqueous phase chemical mechanism : aqueous phase irreversible reactions with rate constants k and temperature dependencies (Ea/R) where available and aqueous phase equilibria (Ervens et al., 2003, 2008) .

| Reactions | Reactants | Products | k [$M^{-1}$ $s^{-1}$] | Ea/R [K] |
|---|---|---|---|---|
| Aqueous phase irreversible reactions | | | | |
| $R_1$ | $SO_2 + O_3$ | $S(VI) + O_2$ | $2.4\times10^4$ | |
| $R_2$ | $HSO_3^- + O_3$ | $S(VI) + O_2$ | $3.7\times10^5$ | 5530 |
| $R_3$ | $SO_3^{2-} + O_3$ | $S(VI) + O_2$ | $1.5\times10^9$ | 5280 |
| $R_4$ | $H_2O_2 + HSO_3$ | $S(VI) + H_2O$ | $7.2\times10^7$ | 14000 |
| $R_5$ | $HO_2 + HO_2$ | $H_2O_2 + O_2$ | $8.3\times10^5$ | 2720 |
| $R_6$ | $O_2^- + HO_2$ | $H_2O_2 + O_2$ | $9.7\times10^7$ | 1060 |
| $R_7$ | $OH + CH_2O$ | $HO_2 + HCOOH$ | $1\times10^9$ | 1020 |
| $R_8$ | $OH + CH_3OOH$ | $CH_3O_2 + H_2O$ | $2.4\times10^7$ | 1680 |
| $R_9$ | $OH + CH_3OOH$ | $HO_2 + HCOOH$ | $6\times10^6$ | 1680 |
| $R_{10}$ | $O_3 + O_2^- (+ H^+)$ | $OH + 2\ O_2$ | $1.5\times10^9$ | 2200 |
| $R_{11}$ | $OH + CHOCHO$ | $HO_2 + CHOCOOH$ | $1.1\times10^9$ | 1516 |
| $R_{12}$ | $OH + CHOCOOH$ | $HO_2 + H_2C_2O_4$ | $3.6\times10^8$ | 1000 |
| $R_{13}$ | $OH + CHOCOO^-$ | $H_2C_2O_4^-$ | $2.9\times10^9$ | 4300 |
| $R_{14}$ | $OH + CH_3COCHO$ | $HO_2 + 0.92\ CH3COCOOH + 0.08\ CHOCOOH$ | $1.1\times10^9$ | 1600 |
| $R_{15}$ | $OH + CH_2(OH)CHO$ | $HO_2 + COOHCH_2OH$ | $1.2\times10^9$ | |
| $R_{16}$ | $OH + COOHCH_2OH$ | $HO_2 + CHOCOOH$ | $1.2\times10^9$ | |
| $R_{17}$ | $OH + C_2O_4^{2-}$ | $O_2^- + 2\ CO_2 + OH^-$ | $1.6\times10^8$ | 4300 |
| $R_{18}$ | $OH + HC_2O_4^-$ | $HO_2 + 2\ CO_2 + OH^-$ | $1.9\times10^8$ | 2800 |
| $R_{19}$ | $OH + HOOCCOOH$ | $HO_2 + 2\ CO_2 + H_2O$ | $1.4\times10^6$ | |
| $R_{20}$ | $OH + CH_3C(O)COOH$ | $HO_2 + CO_2 + CH_3COO^-$ | $7\times10^8$ | |
| $R_{21}$ | $OH + CH_3COCOOH$ | $HO_2 + H_2O + CH_3COOH$ | $1.2\times10^8$ | |
| $R_{22}$ | $OH + HCOO^-$ | $HO_2 + CO_2 + H_2O$ | $3.2\times10^9$ | 1000 |

| $R_{23}$ | OH + HCOOH | $HO_2 + CO_2 + H_2O$ | $1.3\times10^8$ | 1000 |
|---|---|---|---|---|
| $R_{24}$ | $OH + CH_3COO^-$ | $HO_2 + OH^- + 0.15\ CH_2O + 0.85$ CHOCOOH | $1\times10^8$ | 1800 |
| $R_{25}$ | $OH + CH_3COOH$ | $HO_2 + H_2O + 0.15\ CH_2O + 0.85$ CHOCOOH | $1.5\times10^7$ | 1330 |
| $R_{26}$ | $CH_3O_2 + CH_3O_2$ | $CH_2O + CH_3OH + HO_2$ | $1.7\times10^8$ | 2200 |
| $R_{27}$ | $H_2O_2 + OH$ | $HO_2 + H_2O$ | $3\times10^7$ | 1680 |
| $R_{28}$ | $OH + CH_3CHO$ | $HO_2 + H_2O + CH_3COOH$ | $3.6\times10^9$ | 580 |
| $R_{29}$ | $O_2^- + CH_3(CO)OO$ | $CH_3COOH$ | $1\times10^9$ | |
| $R_{30}$ | $CH_3C(O)OO + CH3C(O)OO$ | $2\ CH_3O_2 + 2\ CO_2$ | $1.5\times10^8$ | |
| $R_{31}$ | $OH + HO\text{-}CH_2\text{-}CHO$ | $HO_2 + CH_2(OH)COOH$ | $5\times10^8$ | |
| $R_{32}$ | OH + WSOC | $WSOC + HO_2$ | $3.8\times10^8$ | |
| Aqueous phase equilibria | | | | |
| E | Reactants | products | $K_a$ [M] | |
| $E_1$ | $H_2O$ | $OH^- + H^+$ | $1.0\times10^{-14}$ | |
| $E_2$ | $HO_2$ | $O_2^- + H^+$ | $1.60\times10^{-5}$ | |
| $E_3$ | CHOCOOH | $CHOCOO^- + H^+$ | $6.60\times10^{-4}$ | |
| $E_4$ | HCOOH | $HCOO^- + H^+$ | $1.77\times10^{-4}$ | |
| $E_5$ | $CH_3COCOOH$ | $CH_3COCOO^- + H^+$ | $4.07\times10^{-3}$ | |
| $E_6$ | $CH_3COOH$ | $CH_3COO^- + H^+$ | $1.77\times10^{-5}$ | |
| $E_7$ | $H_2C_2O_4$ | $HC_2O_4^- + H^+$ | $6.40\times10^{-2}$ | |
| $E_8$ | $HC_2O_4^-$ | $C_2O_4^{2-} + H^+$ | $5.25\times10^{-5}$ | |
| $E_9$ | $HNO_3$ | $NO_3^- + H^+$ | 22 | |
| $E_{10}$ | $HOCH_2CH_2OH$ | $HOCH_2CH_2O^- + H^+$ | $1.54\times10^{-4}$ | |
| $E_{11}$ | $SO_2 + H_2O$ | $HSO_3^- + H^+$ | 0.013 | |
| $E_{12}$ | $HSO_3^-$ | $SO_3^{2-} + H^+$ | $6.60\times10^{-8}$ | |
| $E_{13}$ | $H_2SO_4$ | $HSO_4^- + H^+$ | 1000 | |
| $E_{14}$ | $HSO_4^-$ | $SO_4^{2-} + H^+$ | 0.102 | |
| $E_{15}$ | $CO_2(aq) + H_2O$ | $HCO_3^- + H^+$ | $7.70\times10^{-7}$ | |
| $E_{16}$ | $HCO_3^-$ | $CO_3^{2-} + H^+$ | $4.84\times10^{-11}$ | |
| $E_{17}$ | $NH_3$ | $NH_4^+ + OH^-$ | $1.76\times10^{-5}$ | |
| $E_{18}$ | $H_2O$ | $H^+ + OH^-$ | $1\times10^{-14}$ | |

**Table S2.** Phase transfer parameters of soluble species to calculate the mass transfer coefficient (Eq-S1) and Henry's law constants $K_H$. ($M_g$: molecular weight, $\alpha$: mass accommodation coefficient *(dimensio less)*, $D_g$: gas phase diffusion coefficient

| Species | $M_g$ [g mol$^{-1}$] | $\alpha$[1] | $D_g$ [cm s$^{-1}$][1] | $K_H$ [M atm$^{-1}$] |
|---|---|---|---|---|
| $O_3$ | 48 | 0.05 | 0.148 | $1.14\times10^{-2}$[1] |
| $H_2O_2$ | 34 | 0.1 | 0.11 | $1.02\times10^{5}$[1] |
| HO | 17 | 0.05 | 0.153 | 25[1] |
| $HO_2$ | 33 | 0.01 | 0.104 | $9\times10^{3}$[1] |
| HCHO | 30 | 0.02 | 0.164 | $4.99\times10^{3}$[1] |
| $CH_3O_2$ | 47 | 0.0038 | 0.135 | 310[1] |
| $CH_3OOH$ | 48 | 0.0038 | 0.135 | 310 |
| $HNO_3$ | 63 | 0.054 | 0.132 | $2.1\times10^{5}$[1] |
| $N_2O_5$ | 108 | 0.0037 | 0.110 | 1.4[1] |
| Hydroxyaldehyde | 60 | 0.03 | 0.195 | $4.1\times10^{4}$[3] |
| Glyoxal | 58 | 0.023 | 0.115 | $4.19\times10^{5}$[1] |

| | | | | |
|---|---|---|---|---|
| Methylglyoxal | 72 | 0.1 | 0.115 | $3.2\times10^{4\,[3]}$ |
| HCOOH | 74 | 0.012 | 0.153 | $1.77\times10^{-4[1]}$ |
| Acetic acid | 46 | 0.1 | 0.1 | $4\times10^{3[4]}$ |
| Glyoxylic acid | 60 | 0.019 | 0.124 | $9\times10^{3[2]}$ |
| Glycolic acid | 76 | 0.1 | 0.1 | $9\times10^{3[2]}$ |
| Pyruvate | 88 | 0.1 | 0.1 | $3.11\times10^{5[3]}$ |
| Oxalate | 90 | 0.1 | 0.1 | $9\times10^{3[3]}$ |
| Hydroxyketone | 88 | 0.1 | 0.1 | $100^{[3]}$ |
| $CH_3O_3$ | 75 | 0.1 | 0.1 | $669^{[3]}$ |
| Aldehyde | 44 | 0.1 | 0.1 | $11.4^{[3]}$ |
| $SO_2$ | 64 | 0.035 | 0.128 | $1.23^{[3]}$ |
| $CO_2$ | 44 | $2.e^{-4}$ | 0.155 | $3.11\times10^{-2[1]}$ |
| Glycolaldehyde | 58 | 0.1 | 0.1 | $4.1\times10^{4[3]}$ |
| $C_2H_5OOH$ | 62 | 0.1 | 0.1 | $310^{[1]}$ |
| Organic compound that reacts with OH and is consumed by bacteria | 150 | 0.1 | 0.1 | $10^2$ - $10^9$ |

[1]: (Ervens et al., 2003a), [2]: (Ip et al., 2009), [3]: (Sander, 2015), [4]: (Johnson et al., 1996)

**Table S3**. Initial gas phase mixing ratios of gas phase species and the concentration of species only in the aqueous phase; all other compounds considered in the mechanism were not initialized

| Species formula | Species name | Mixing ratio [ppb] |
|---|---|---|
| Gases with phase transfer into the aqueous phase | | |
| $O_3$ | ozone | 39 |
| $H_2O_2$ | hydrogen peroxide | 1.202 |
| OH | hydroxyl radical | $3.05\times10^{-11}$ |
| $HO_2$ | hydroperoxyl radical | $9.07\times10^{-3}$ |
| HCHO | formaldehyde | 2.519 |
| $CH_3O_2$ | methylperoxy radical | $1.38\times10^{-3}$ |
| $CH_3OOH$ | methyl hydrogen peroxide | 0.211 |
| $HNO_3$ | nitric acid | 0.397 |
| $N_2O_5$ | dinitrogen pentoxide | $4.806\times10^{-4}$ |
| $CH_2(OH)CHO$ | hydroxy acetaldehyde | 0.437 |
| CHOCHO | glyoxal | 0.218 |
| $CH_3COCHO$ | methyl glyoxal | 0.190 |
| HCOOH | formic acid | $2.239\times10^{-3}$ |
| HAc | Acetic acid | 0.198 |
| $CH_3CO(OO)$ | acetylperoxy radical | $5.5\times10^{-5}$ |
| $SO_2$ | sulfur dioxide | 0.150 |
| $CH_3CHO$ | acetaldehyde | 0.409 |
| $CO_2$ | carbon dioxide | $3.96\times10^5$ |
| $CH_2(OH)CHO$ | glycolaldehyde | 0.4273 |
| $CH_3CH_2(OOH)$ | ethyl hydrogen peroxide | $2.423\times10^{-3}$ |
| Organic compound | | 1 |
| compounds without phase transfer into the gas phase | | |
| NO | nitric oxide | 0.0429 |
| $NO_2$ | nitrogen dioxide | 0.179 |
| $NO_3$ | nitrate radical | $3.5\times10^{-4}$ |

| | | |
|---|---|---|
| HNO$_4$ | Peroxynitric acid | $8.83 \times 10^{-3}$ |
| CO | carbon monoxide | 140.3 |
| C$_5$H$_8$ | isoprene | 1.031 |
| MACR | methacrolein | 0.282 |
| MVK | methyl vinyl ketone | 0.113 |
| C$_2$H$_6$ | ethane | 0.846 |
| C$_2$H$_4$ | ethene | 0.469 |
| C$_3$H$_6$ | propene | 0.118 |
| H$_2$ | hydrogen | 550 |
| CH$_3$CO(OOH) | peracetic acid | 0.240 |
| C$_4$H$_{10}$ | butane | 0.142 |
| CH$_3$CH$_2$(OO) | ethylperoxy radical | $7.144 \times 10^{-6}$ |
| PO$_2$ | Other peroxy radicals (C$_2$, C$_3$) | $1.414 \times 10^{-4}$ |
| POOH | Hydroperoxides of PO$_2$ | $2.076 \times 10^{-3}$ |
| PAN | peroxy acetyl nitrate | 0.586 |
| Isop-OO | | $4.181 \times 10^{-3}$ |

---

## Author Comment (AC2) · 18 Dec 2020

We would like to thank the referee for their positive and insightful comments on the manuscript. Below is our point-by-point response to the comments. Our responses are in black, manuscript text is in italic with new/modified text is marked in blue. The line numbers cited here refer to those in the clean manuscript version.

**Referee comment:** Section 2.1: The model description is too concise and could be improved a bit. Which 26 species are transferred between the gas and aqueous phases? It would be worth showing the coupled mass transfer ODEs with Schwartz's treatment.

**Author response:** We agree with the referee that the model description in the section 2.1 was very short. Since also the other referee asked for more detail on the model, we added three tables into the supplement with details on our aqueous phase chemical mechanism (Table S1), phase transfer parameters (Table S2) and initial mixing ratios and concentrations (Table S3). Please find all tables at the end of this response.

We will add the following text at the beginning of Section 2.1:

*We use a multiphase box model with detailed gas and aqueous phase chemistry (75 species, 44 gas phase reactions, 31 aqueous reactions). The chemical aqueous phase mechanism with rate constants is listed in **Table S1.** The chemical gas phase mechanism is based on the NCAR Master mechanism (Aumont et al., 2000; Madronich and Calvert, 1989). The two phases are coupled by 26 phase transfer processes which is described kinetically based on the resistance model by Schwartz (1986). The parameters describing the phase transfer of the soluble species are presented in **Table S2**. In addition, the initial gas phase mixing ratios are listed in **Table S3**. The equations for the mass transfer coefficient $k_{mt}$ and the differential equations for the aqueous and gas phase concentrations can be found in the supplement **(Eq-R1-R3; Section 'Description of phase transfer)** (Seinfeld and Pandis, 1998).*

In addition, we add the following set of equations for the description of the phase transfer to the supplement:

$$k_{mt} = \left[\frac{r_d^2}{3D_g} + \frac{r_d}{3\alpha}\sqrt{\frac{2\pi M_g}{RT}}\right]^{-1} \tag{Eq-R1}$$

*Whereas*

*$r_d$ = cloud droplet radius [cm]*

*$D_g$ = gas phase diffusion coefficient [cm s$^{-1}$]*

*$\alpha$ = mass accommodation coefficient (dimensionless)*

*$M_g$ = molecular weight of gas [g mol$^{-1}$]*

*$R$ = constant for ideal gases (8.314 ·10$^7$ erg mol$^{-1}$ K$^{-1}$)*

*$T$ = temperature [K]*

*The mass transfer coefficient is then applied to determine the sink and source terms due to phase transfer of soluble species:*

$$\frac{dc_g}{dt} = k_{mt} \cdot LWC \cdot \left[\frac{c_{aq}}{LWC \cdot K_{H(eff)} \cdot R' \cdot T} - c_g\right] + (P_{gaschem} - L_{gaschem}) \tag{Eq-R2}$$

$$\frac{dc_{aq}}{dt} = k_{mt} \cdot LWC \cdot \left[c_g - \frac{c_{aq}}{LWC \cdot K_{H(eff)} \cdot R' \cdot T}\right] + (P_{aqchem} - L_{aqchem}) \qquad (Eq\text{-}R3)$$

*whereas both the gas and the aqueous phase concentrations have units of mol g(air)$^{-1}$; LWC is the liquid water content in g/m$^3$, $K_{H(eff)}$ is the effective Henry's law constant in M atm$^{-1}$, $R'$ is the ideal-gas constant (8.314.10$^7$ erg mol$^{-1}$ K$^{-1}$), T is the temperature [K] and P and L are the rates of the chemical production and loss reactions in the gas and aqueous phases, respectively.*

**Referee comment:** Is the size class same as the size bin? On line 101, it is stated that the 5 um < Ddroplet < 20 um, but then one droplet size class has $D_{droplet}$ = 20 um. Should the range be changed to 5 um < $D_{droplet}$ ≤ 20 um? Also, why is only the last size class allowed to have bacteria cells?

**Author response:** The referee is correct that the size range of the droplets should be written as 5 μm ≤ $D_{droplet}$ ≤ 30 μm .

In our box model, we chose a drop size spectrum with diameters of 5 to 30 μm. The bacteria are only in one of the drop classes (diameter 20 μm). Note that other box model studies usually only consider a single drop size (Deguillaume et al., 2004; Ervens et al., 2003; Tilgner et al., 2013). However, as we have shown previously that the drop size may impact the OH(aq) concentration and distribution (Ervens et al., 2014), we used a polydisperse drop size distribution. While also the uptake of organic compounds may be drop-size-dependent, we did not further explore this effect as it would not add significantly to our conclusions.

**Referee comment:** Line 105-106: Which organic compound? Is the model run separately for each individual compound? What other inorganic species were considered in the model? What was the cloud water pH in the model simulations discussed in section 3?

**Author response:** We removed the sentence in line 105/106 (now line: 116) and clarified now which species are initialized in our model. We provide more detail on our model in Table S1 (detailed aqueous phase chemical mechanism) ,Table S2 (phase transfer parameters) and Table S3 (initial concentration of different species).

**Referee comment**: While the degradation rate constant remains constant between pH values 5 and 8, could it change at lower pH?

**Author response:**

The referee is correct to point out the possible effects of acidity on bacterial activity. Several studies investigated the acid tolerance mechanism of different microorganisms, e.g., (Casal et al., 2016; Patel et al., 2006) by decarboxylation, deamination (Noh et al., 2018), or cell membrane modification (Zhang et al., 2011). In addition, some bacteria can develop an acid resistance system to survive under acidic conditions (pH=2.5) (Lu et al., 2013). However, these strategies do not necessarily imply that the bacteria maintain the same biodegradation activities at high acidity but they allow survival of the cells in the atmosphere.

When exposed to very broad ranges of external pHs, bacteria can control their intracellular pH (~6.5 -7) by internal buffering (Delort et al., 2017). As biodegradation occurs inside the cell, it takes place at these (nearly) neutral conditions. The efficiency of buffering decreases at extreme conditions, e.g., pH < 2 or pH > 10 (Guan and Liu, 2020). However, such pH range is not representative for cloud water where more moderate pH values (~ 3 – 6) are typically found (Deguillaume et al., 2014).

We will modify the text in Section 2.2.2. as follows:

*Experiments with 17 different cloud bacteria in artificial cloud water with pH = 5.0 and pH = 6.5 showed also nearly identical results* (Vaïtilingom et al., 2011). . *Similar results were shown by Razika et al. (2010) who demonstrated that biodegradation rates of phenol by Pseudomonas aeruginosa were very similar when incubated at pH = 5.8, 7.0 and 8.0, respectively. When exposed to very broad ranges of external pHs, bacteria can control their intracellular pH (~6.5 -7) by internal buffering (Delort et al., 2017). As biodegradation occurs inside the cell, it takes place at these (nearly) neutral conditions. The efficiency of buffering decreases at extreme conditions, e.g., pH < 2 or pH > 10 (Guan and Liu, 2020). However, such pH range is not representative for cloud water where more moderate pH values (~ 3 – 6)(Deguillaume et al., 2014) are typically found.*

In addition, we add in the conclusion section (l.596):

*In addition, the biodegradation rates may be affected under highly acidic conditions. Some studies demonstrated that some bacteria can develop an acid resistance to survive under acidic conditions ((Lu et al., 2013). However, these strategies do not necessarily imply that the bacteria maintain the same biodegradation activities at high acidity but they allow survival of the cells in the atmosphere. It can be expected that internal buffering of bacteria cells allows them to maintain their metabolic activity over wide pH ranges (~ 3 to 6) as found in cloud water. Therefore, we do not consider a potential pH dependency of biodegradation rates in our model studies.*

**Referee comment:** Figures 2, 3, 6, 7, 8, 10 are presently displayed as three-dimensional plots, which I found a little difficult to read as parts of the plots are obscured by curved surfaces. Since the Z-axis and colors represent the same dimension, I suggest replotting these figures as two-dimensional color contour plots. This would greatly improve the quality and readability of the figures.

**Author response:** Thank you for this suggestion, we agree with this suggestion and show in the revised version the figures 2,3,6,7,8 and 10 and Figures S1 and S2 as two-dimensional color contour plots to improve the quality and readability of the figures as suggested.

**Referee comment**: Table 2: There's an extra multiplication symbol in the 5 C Experimental rate column on 12th row.

**Author response:** Corrected

Aumont, B., Madronich, S., Bey, I. and Tyndall, G.: Contribution of Secondary VOC to the Composition of Aqueous Atmospheric Particles: A Modeling Approach, J. Atmos. Chem., 35(1), 59–75, doi:10.1023/a:1006243509840, 2000.

Casal, M., Queirós, O., Talaia, G., Ribas, D. and Paiva, S.: Carboxylic acids plasma membrane transporters in saccharomyces cerevisiae, in Advances in Experimental Medicine and Biology., 2016.

Deguillaume, L., Leriche, M., Monod, A. and Chaumerliac, N.: The role of transition metal ions on HOx radicals in clouds: a numerical evaluation of its impact on multiphase chemistry, Atmos. Chem. Phys., doi:10.5194/acp-4-95-2004, 2004.

Deguillaume, L., Charbouillot, T., Joly, M., Vaïtilingom, M., Parazols, M., Marinoni, A., Amato, P., Delort, A. M., Vinatier, V., Flossmann, A., Chaumerliac, N., Pichon, J. M., Houdier, S., Laj, P., Sellegri, K., Colomb, A., Brigante, M. and Mailhot, G.: Classification of clouds sampled at the puy de Dôme (France) based on 10 yr of monitoring of their physicochemical properties,

Atmos. Chem. Phys., doi:10.5194/acp-14-1485-2014, 2014.

Delort, A.-M., Deguillaume, L., Renard, P., Vinatier, V., Canet, I., Vaïtilingom, M. and Chaumerliac, N.: Impacts on Cloud Chemistry, in Microbiology of Aerosols., 2017.

Ervens, B., George, C., Williams, J. E., Buxton, G. V., Salmon, G. A., Bydder, M., Wilkinson, F., Dentener, F., Mirabel, P., Wolke, R. and Herrmann, H.: CAPRAM 2.4 (MODAC mechanism): An extended and condensed tropospheric aqueous phase mechanism and its application, J. Geophys. Res. Atmos., 108(D14), doi:10.1029/2002jd002202, 2003.

Ervens, B., Carlton, A. G., Turpin, B. J., Altieri, K. E., Kreidenweis, S. M. and Feingold, G.: Secondary organic aerosol yields from cloud-processing of isoprene oxidation products, Geophys. Res. Lett., 35(2), doi:10.1029/2007GL031828, 2008.

Ervens, B., Sorooshian, A., Lim, Y. B. and Turpin, B. J.: Key parameters controlling OH-initiated formation of secondary organic aerosol in the aqueous phase (aqSOA), J. Geophys. Res. - Atmos., 119(7), 3997–4016, doi:10.1002/2013JD021021, 2014.

Guan, N. and Liu, L.: Microbial response to acid stress: mechanisms and applications, Appl. Microbiol. Biotechnol., 51–65, doi:10.1007/s00253-019-10226-1, 2020.

Ip, H. S. S., Huang, X. H. H. and Yu, J. Z.: Effective Henry's law constants of glyoxal, glyoxylic acid, and glycolic acid, Geophys. Res. Lett., doi:10.1029/2008GL036212, 2009.

Johnson, B. J., Betterton, E. A. and Craig, D.: Henry's Law coefficients of formic and acetic acids, J. Atmos. Chem., 24(2), 113--119, doi:10.1007/BF00162406, 1996.

Lu, P., Ma, D., Chen, Y., Guo, Y., Chen, G. Q., Deng, H. and Shi, Y.: L-glutamine provides acid resistance for Escherichia coli through enzymatic release of ammonia, Cell Res., doi:10.1038/cr.2013.13, 2013.

Madronich, S. and Calvert, J. G.: No Title, NCAR Tech. Note TN-474+STR, 1989.

Noh, M. H., Lim, H. G., Woo, S. H., Song, J. and Jung, G. Y.: Production of itaconic acid from acetate by engineering acid-tolerant Escherichia coli W, Biotechnol. Bioeng., doi:10.1002/bit.26508, 2018.

Patel, M. A., Ou, M. S., Harbrucker, R., Aldrich, H. C., Buszko, M. L., Ingram, L. O. and Shanmugam, K. T.: Isolation and characterization of acid-tolerant, thermophilic bacteria for effective fermentation of biomass-derived sugars to lactic acid, Appl. Environ. Microbiol., doi:10.1128/AEM.72.5.3228-3235.2006, 2006.

Sander, R.: Compilation of Henry's law constants (version 4.0) for water as solvent, Atmos. Chem. Phys., 15(8), 4399–4981, doi:10.5194/acp-15-4399-2015, 2015.

Seinfeld, J. H. and Pandis, S. N.: Atmospheric Chemistry and Physics, John Wiley & Sons, New York., 1998.

Tilgner, A., Bräuer, P., Wolke, R. and Herrmann, H.: Modelling multiphase chemistry in deliquescent aerosols and clouds using CAPRAM3.0i, J. Atmos. Chem., 70(3), 221–256, doi:10.1007/s10874-013-9267-4, 2013.

Zhang, J. G., Liu, X. Y., He, X. P., Guo, X. N., Lu, Y. and Zhang, B. run: Improvement of acetic

acid tolerance and fermentation performance of Saccharomyces cerevisiae by disruption of the FPS1 aquaglyceroporin gene, Biotechnol. Lett., doi:10.1007/s10529-010-0433-3, 2011.

**Table S1**. Aqueous phase chemical mechanism : aqueous phase irreversible reactions with rate constants k and temperature dependencies (Ea/R) where available and aqueous phase equilibria (Ervens et al., 2003, 2008) .

| Reactions | Reactants | Products | $k$ [$M^{-1} s^{-1}$] | Ea/R [K] |
|---|---|---|---|---|
| Aqueous phase irreversible reactions | | | | |
| $R_1$ | $SO_2 + O_3$ | $S(VI) + O_2$ | $2.4 \times 10^4$ | |
| $R_2$ | $HSO_3^- + O_3$ | $S(VI) + O_2$ | $3.7 \times 10^5$ | 5530 |
| $R_3$ | $SO_3^{2-} + O_3$ | $S(VI) + O_2$ | $1.5 \times 10^9$ | 5280 |
| $R_4$ | $H_2O_2 + HSO_3$ | $S(VI) + H_2O$ | $7.2 \times 10^7$ | 14000 |
| $R_5$ | $HO_2 + HO_2$ | $H_2O_2 + O_2$ | $8.3 \times 10^5$ | 2720 |
| $R_6$ | $O_2^- + HO_2$ | $H_2O_2 + O_2$ | $9.7 \times 10^7$ | 1060 |
| $R_7$ | $OH + CH_2O$ | $HO_2 + HCOOH$ | $1 \times 10^9$ | 1020 |
| $R_8$ | $OH + CH_3OOH$ | $CH_3O_2 + H_2O$ | $2.4 \times 10^7$ | 1680 |
| $R_9$ | $OH + CH_3OOH$ | $HO_2 + HCOOH$ | $6 \times 10^6$ | 1680 |
| $R_{10}$ | $O_3 + O_2^- (+ H^+)$ | $OH + 2 O_2$ | $1.5 \times 10^9$ | 2200 |
| $R_{11}$ | $OH + CHOCHO$ | $HO_2 + CHOCOOH$ | $1.1 \times 10^9$ | 1516 |
| $R_{12}$ | $OH + CHOCOOH$ | $HO_2 + H_2C_2O_4$ | $3.6 \times 10^8$ | 1000 |
| $R_{13}$ | $OH + CHOCOO^-$ | $H_2C_2O_4^-$ | $2.9 \times 10^9$ | 4300 |
| $R_{14}$ | $OH + CH_3COCHO$ | $HO_2 + 0.92 CH3COCOOH + 0.08 CHOCOOH$ | $1.1 \times 10^9$ | 1600 |
| $R_{15}$ | $OH + CH_2(OH)CHO$ | $HO_2 + COOHCH_2OH$ | $1.2 \times 10^9$ | |
| $R_{16}$ | $OH + COOHCH_2OH$ | $HO_2 + CHOCOOH$ | $1.2 \times 10^9$ | |
| $R_{17}$ | $OH + C_2O_4^{2-}$ | $O_2^- + 2 CO_2 + OH^-$ | $1.6 \times 10^8$ | 4300 |
| $R_{18}$ | $OH + HC_2O_4^-$ | $HO_2 + 2 CO_2 + OH^-$ | $1.9 \times 10^8$ | 2800 |
| $R_{19}$ | $OH + HOOCCOOH$ | $HO_2 + 2 CO_2 + H_2O$ | $1.4 \times 10^6$ | |
| $R_{20}$ | $OH + CH_3C(O)COOH$ | $HO_2 + CO_2 + CH_3COO^-$ | $7 \times 10^8$ | |
| $R_{21}$ | $OH + CH_3COCOOH$ | $HO_2 + H_2O + CH_3COOH$ | $1.2 \times 10^8$ | |
| $R_{22}$ | $OH + HCOO^-$ | $HO_2 + CO_2 + H_2O$ | $3.2 \times 10^9$ | 1000 |
| $R_{23}$ | $OH + HCOOH$ | $HO_2 + CO_2 + H_2O$ | $1.3 \times 10^8$ | 1000 |
| $R_{24}$ | $OH + CH_3COO^-$ | $HO_2 + OH^- + 0.15 CH_2O + 0.85 CHOCOOH$ | $1 \times 10^8$ | 1800 |
| $R_{25}$ | $OH + CH_3COOH$ | $HO_2 + H_2O + 0.15 CH_2O + 0.85 CHOCOOH$ | $1.5 \times 10^7$ | 1330 |
| $R_{26}$ | $CH_3O_2 + CH_3O_2$ | $CH_2O + CH_3OH + HO_2$ | $1.7 \times 10^8$ | 2200 |
| $R_{27}$ | $H_2O_2 + OH$ | $HO_2 + H_2O$ | $3 \times 10^7$ | 1680 |
| $R_{28}$ | $OH + CH_3CHO$ | $HO_2 + H_2O + CH_3COOH$ | $3.6 \times 10^9$ | 580 |
| $R_{29}$ | $O_2^- + CH_3(CO)OO$ | $CH_3COOH$ | $1 \times 10^9$ | |
| $R_{30}$ | $CH_3C(O)OO + CH3C(O)OO$ | $2 CH_3O_2 + 2 CO_2$ | $1.5 \times 10^8$ | |
| $R_{31}$ | $OH + HO-CH_2-CHO$ | $HO_2 + CH_2(OH)COOH$ | $5 \times 10^8$ | |
| $R_{32}$ | $OH + WSOC$ | $WSOC + HO_2$ | $3.8 \times 10^8$ | |
| Aqueous phase equilibria | | | | |
| E | Reactants | products | $K_a$ [M] | |
| $E_1$ | $H_2O$ | $OH^- + H^+$ | $1.0 \times 10^{-14}$ | |
| $E_2$ | $HO_2$ | $O_2^- + H^+$ | $1.60 \times 10^{-5}$ | |
| $E_3$ | $CHOCOOH$ | $CHOCOO^- + H^+$ | $6.60 \times 10^{-4}$ | |

| E4 | HCOOH | HCOO⁻ + H⁺ | $1.77\times10^{-4}$ | |
|---|---|---|---|---|
| E₄ | HCOOH | $HCOO^- + H^+$ | $1.77\times10^{-4}$ | |
| E₅ | $CH_3COCOOH$ | $CH_3COCOO^- + H^+$ | $4.07\times10^{-3}$ | |
| E₆ | $CH_3COOH$ | $CH_3COO^- + H^+$ | $1.77\times10^{-5}$ | |
| E₇ | $H_2C_2O_4$ | $HC_2O_4^- + H^+$ | $6.40\times10^{-2}$ | |
| E₈ | $HC_2O_4^-$ | $C_2O_4^{2-} + H^+$ | $5.25\times10^{-5}$ | |
| E₉ | $HNO_3$ | $NO_3^- + H^+$ | 22 | |
| E₁₀ | $HOCH_2CH_2OH$ | $HOCH_2CH_2O^- + H^+$ | $1.54\times10^{-4}$ | |
| E₁₁ | $SO_2 + H_2O$ | $HSO_3^- + H^+$ | 0.013 | |
| E₁₂ | $HSO_3^-$ | $SO_3^{2-} + H^+$ | $6.60\times10^{-8}$ | |
| E₁₃ | $H_2SO_4$ | $HSO_4^- + H^+$ | 1000 | |
| E₁₄ | $HSO_4^-$ | $SO_4^{2-} + H^+$ | 0.102 | |
| E₁₅ | $CO_2(aq) + H_2O$ | $HCO_3^- + H^+$ | $7.70\times10^{-7}$ | |
| E₁₆ | $HCO_3^-$ | $CO_3^{2-} + H^+$ | $4.84\times10^{-11}$ | |
| E₁₇ | $NH_3$ | $NH_4^+ + OH^-$ | $1.76\times10^{-5}$ | |
| E₁₈ | $H_2O$ | $H^+ + OH^-$ | $1\times10^{-14}$ | |

**Table S2.** Phase transfer parameters of soluble species to calculate the mass transfer coefficient (Eq-S1) and Henry's law constants $K_H$. ($M_g$: molecular weight, α: mass accommodation coefficient *(dimensio less)*, $D_g$: gas phase diffusion coefficient

| Species | $M_g$ [g mol⁻¹] | $\alpha^{[1]}$ | $D_g$ [cm s⁻¹][1] | $K_H$ [M atm⁻¹] |
|---|---|---|---|---|
| O₃ | 48 | 0.05 | 0.148 | $1.14\times10^{-2[1]}$ |
| H₂O₂ | 34 | 0.1 | 0.11 | $1.02\times10^{5[1]}$ |
| HO | 17 | 0.05 | 0.153 | $25^{[1]}$ |
| HO₂ | 33 | 0.01 | 0.104 | $9\times10^{3[1]}$ |
| HCHO | 30 | 0.02 | 0.164 | $4.99\times10^{3[1]}$ |
| CH₃O₂ | 47 | 0.0038 | 0.135 | $310^{[1]}$ |
| CH₃OOH | 48 | 0.0038 | 0.135 | 310 |
| HNO₃ | 63 | 0.054 | 0.132 | $2.1\times10^{5[1]}$ |
| N₂O₅ | 108 | 0.0037 | 0.110 | $1.4^{[1]}$ |
| Hydroxyaldehyde | 60 | 0.03 | 0.195 | $4.1\times10^{4[3]}$ |
| Glyoxal | 58 | 0.023 | 0.115 | $4.19\times10^{5[1]}$ |
| Methylglyoxal | 72 | 0.1 | 0.115 | $3.2\times10^{4[3]}$ |
| HCOOH | 74 | 0.012 | 0.153 | $1.77\times10^{-4[1]}$ |
| Acetic acid | 46 | 0.1 | 0.1 | $4\times10^{3[4]}$ |
| Glyoxylic acid | 60 | 0.019 | 0.124 | $9\times10^{3[2]}$ |
| Glycolic acid | 76 | 0.1 | 0.1 | $9\times10^{3[2]}$ |
| Pyruvate | 88 | 0.1 | 0.1 | $3.11\times10^{5[3]}$ |
| Oxalate | 90 | 0.1 | 0.1 | $9\times10^{3[3]}$ |
| Hydroxyketone | 88 | 0.1 | 0.1 | $100^{[3]}$ |
| CH₃O₃ | 75 | 0.1 | 0.1 | $669^{[3]}$ |
| Aldehyde | 44 | 0.1 | 0.1 | $11.4^{[3]}$ |
| SO₂ | 64 | 0.035 | 0.128 | $1.23^{[3]}$ |
| CO₂ | 44 | 2.e⁻⁴ | 0.155 | $3.11\times10^{-2[1]}$ |
| Glycolaldehyde | 58 | 0.1 | 0.1 | $4.1\times10^{4[3]}$ |
| C₂H₅OOH | 62 | 0.1 | 0.1 | $310^{[1]}$ |

| Organic compound that reacts with OH and is consumed by bacteria | 150 | 0.1 | 0.1 | $10^2 - 10^9$ |
|---|---|---|---|---|

[1]: (Ervens et al., 2003a), [2]: (Ip et al., 2009), [3]: (Sander, 2015), [4]: (Johnson et al., 1996)

**Table S3**. Initial gas phase mixing ratios of gas phase species and the concentration of species only in the aqueous phase; all other compounds considered in the mechanism were not initialized

| Species formula | Species name | Mixing ratio [ppb] |
|---|---|---|
| <td colspan="3" align="center">**Gases with phase transfer into the aqueous phase**</td> | | |
| $O_3$ | ozone | 39 |
| $H_2O_2$ | hydrogen peroxide | 1.202 |
| OH | hydroxyl radical | $3.05\times10^{-11}$ |
| $HO_2$ | hydroperoxyl radical | $9.07\times10^{-3}$ |
| HCHO | formaldehyde | 2.519 |
| $CH_3O_2$ | methylperoxy radical | $1.38\times10^{-3}$ |
| $CH_3OOH$ | methyl hydrogen peroxide | 0.211 |
| $HNO_3$ | nitric acid | 0.397 |
| $N_2O_5$ | dinitrogen pentoxide | $4.806\times10^{-4}$ |
| $CH_2(OH)CHO$ | hydroxy acetaldehyde | 0.437 |
| CHOCHO | glyoxal | 0.218 |
| $CH_3COCHO$ | methyl glyoxal | 0.190 |
| HCOOH | formic acid | $2.239\times10^{-3}$ |
| HAc | Acetic acid | 0.198 |
| $CH_3CO(OO)$ | acetylperoxy radical | $5.5\times10^{-5}$ |
| $SO_2$ | sulfur dioxide | 0.150 |
| $CH_3CHO$ | acetaldehyde | 0.409 |
| $CO_2$ | carbon dioxide | $3.96\times10^5$ |
| $CH_2(OH)CHO$ | glycolaldehyde | 0.4273 |
| $CH_3CH_2(OOH)$ | ethyl hydrogen peroxide | $2.423\times10^{-3}$ |
| Organic compound | | 1 |
| <td colspan="3" align="center">**compounds without phase transfer into the gas phase**</td> | | |
| NO | nitric oxide | 0.0429 |
| $NO_2$ | nitrogen dioxide | 0.179 |
| $NO_3$ | nitrate radical | $3.5\times10^{-4}$ |
| $HNO_4$ | Peroxynitric acid | $8.83\times10^{-3}$ |
| CO | carbon monoxide | 140.3 |
| $C_5H_8$ | isoprene | 1.031 |
| MACR | methacrolein | 0.282 |
| MVK | methyl vinyl ketone | 0.113 |
| $C_2H_6$ | ethane | 0.846 |
| $C_2H_4$ | ethene | 0.469 |
| $C_3H_6$ | propene | 0.118 |
| $H_2$ | hydrogen | 550 |
| $CH_3CO(OOH)$ | peracetic acid | 0.240 |
| $C_4H_{10}$ | butane | 0.142 |
| $CH_3CH_2(OO)$ | ethylperoxy radical | $7.144\times10^{-6}$ |
| $PO_2$ | Other peroxy radicals ($C_2$, $C_3$) | $1.414\times10^{-4}$ |
| POOH | Hydroperoxides of $PO_2$ | $2.076\times10^{-3}$ |
| PAN | peroxy acetyl nitrate | 0.586 |
| Isop-OO | | $4.181\times10^{-3}$ |

---

## Author Response (AR1)

We thank the referee for their constructive and positive comments on our manuscript. We respond to all comments in detail below. Our responses are in black, manuscript text is in italic with new/modified text is marked in blue. The line numbers cited here refer to those in the clean manuscript version.

**Referee comment**: - Throughout the manuscript I found terminology of VOC vs. NVOC somewhat confusing. It seems that you categorize compounds based on solubility ($K_H$) but not with vapor pressures (no vapor pressures are given for treated compounds). For organic aerosol research, VOC is often used for compounds mostly in the gas phase, while NVOC for compounds mostly in the condensed phase. As you know, compounds which may exist both phases comparably are termed semi-volatile organic compounds (SVOC), so most compounds treated in this study appear better termed as SVOC. Some more clarifications with clear definition would help improve presentation quality.

**Author response**: We thank the referee for pointing out this inconsistency in our terminology. We agree that this was misleading as volatility is not the main criterion for the partitioning of species between the gas and aqueous phases, respectively. The categorization into VOC, NVOC and SVOC is mostly relevant for the partitioning of compounds between the gas and a condensed phase, such as present in aerosol. However, as the volatility does not (necessarily) correlate with water-solubility (e.g. alkanes and small aldehydes exhibit similar vapor pressures but vastly different Henry's law constants), these categories do not seem useful for the categorization of water-soluble compounds into the aqueous phase of cloud droplets.

Therefore, we categorize the various compounds now exclusively based on their solubility in water, i.e. on their Henry's law constant. We distinguish now four compound groups and we will add the following text at the end of the section 2.1.:

[revised manuscript text omitted]

*d) $L_{bact}$ for CCN-derived organics (Table 1). The contour plot is identical to Figure 8 b) .Note that $R_{chemaq}$ and $R_{bact}$ for succinate (pH = 6) is nearly identical to the values for succinic acid (pH = 3) and were therefore omitted from the figure.*

*- Header of Section 3.1.1:  $fr_{bact}$ for*  *: $fr_{bact}$ for water-soluble organic gases*

*- Header of section 3.1.2: $fr_{bact}$ of*  *and comparison to $fr_{bact}$ of*  *: $fr_{bact}$ of CCN-derived compounds and comparison to $fr_{bact}$ of water-soluble organic gases.*

*- Header of Section 3.2.1:  $L_t$ for* *: $L_t$ for water-soluble organic gases*

*- Header od Section 3.2.2.: $L_{bact}$ for* *: $L_{bact}$ for water-soluble organic gases*

*- Header od Section 3.2.3: $L_t$ and $L_{bact}$ for* *: $L_t$ and $L_{bact}$ for CCN-derived compounds*

**Referee comment**: - It was not clear to me how you treat phase transfer in your model. It is just simply stated that it is based on the resistance model (P3), but unclear how exactly you treat (Fuchs-Sutugin correction?) and what values you use for critical parameters such as gas diffusivity and mass accommodation coefficient.

**Author response:** Since also the other referee asked for more details on the model, we added three tables into the supplement with details of our aqueous phase chemical mechanism (Table S1), phase transfer parameters (Table S2) and initial mixing ratios and concentrations (Table S3). In addition, we added the following set of equations for the description of the phase transfer to the supplement:

*The mass transfer coefficient $k_{mt}$ [$s^{-1}$] can be expressed as (Seinfeld and Pandis, 1998)*

$$k_{mt} = \left[ \frac{r_d^2}{3D_g} + \frac{r_d}{3\alpha} \sqrt{\frac{2\pi M_g}{RT}} \right]^{-1} \qquad (Eq\text{-}R1)$$

*Whereas*

*$r_d$ = cloud droplet radius [cm]*

*$D_g$ = gas phase diffusion coefficient [cm $s^{-1}$]*

*$\alpha$ = mass accommodation coefficient (dimensionless)*

*$M_g$ = molecular weight of gas [g mol$^{-1}$]*

*$R$ = constant for ideal gases (8.314 ·10$^7$ erg mol$^{-1}$ K$^{-1}$)*

*$T$ = temperature [K]*

*The mass transfer coefficient is then applied to determine the sink and source terms of soluble species:*

$$\frac{dc_g}{dt} = k_{mt} \cdot LWC \cdot \left[ \frac{c_{aq}}{LWC \cdot K_{H(eff)} \cdot R' \cdot T} - c_g \right] + (P_{gaschem} - L_{gaschem}) \qquad \text{(Eq-R2)}$$

$$\frac{dc_{aq}}{dt} = k_{mt} \cdot LWC \cdot \left[ c_g - \frac{c_{aq}}{LWC \cdot K_{H(eff)} \cdot R' \cdot T} \right] + (P_{aqchem} - L_{aqchem}) \qquad \text{(Eq-R3)}$$

*whereas both the gas and the aqueous phase concentrations have units of mol g(air)$^{-1}$; LWC is the liquid water content in g/m$^3$, $K_{H(eff)}$ is the effective Henry's law constant in M atm$^{-1}$, $R'$ is the ideal-gas constant (8.314.10$^7$ erg mol$^{-1}$ K$^{-1}$), T is the temperature [K] and P and L are the rates of the chemical production and loss reactions in the gas and aqueous phases, respectively.*

In brief, we do not use the Fuchs and Sutugin correction as this applies for the transition regime of transport of gas molecules towards particles, i.e. when the mean free path length of molecules ($\lambda$) is approximately equal to the particle diameter (Dp), i.e. when the Knudsen number is Kn ~ 1. The transport of gas molecules towards droplets with typical diameters of 10 μm or larger falls into the continuum regime. For this regime, the various processes that affect the transport of gas molecules into droplets were expressed in terms of 'resistances' by (Schwartz, 1986). These processes generally include gas phase diffusion, interfacial mass transport, aqueous phase diffusion and aqueous phase chemical reaction. In previous sensitivity studies, we have shown that aqueous phase diffusion is usually sufficiently fast compared to the interfacial mass transfer chemical reaction in cloud droplets so its term can be neglected (Ervens et al., 2003b, 2014).

We add the following text at the beginning of Section 2.1:

*We use a multiphase box model with detailed gas and aqueous phase chemistry (75 species, 44 gas phase reactions, 31 aqueous reactions).* The chemical aqueous phase mechanism with rate constants is listed in **Table S1.** The chemical gas phase mechanism is based on the NCAR Master mechanism (Aumont et al., 2000; Madronich and Calvert, 1989). *The two phases are coupled by 26 phase transfer processes which is described kinetically based on the resistance model by Schwartz (1986). The parameters describing the phase transfer of the soluble species are presented in* **Table S2**. *In addition, the initial mixing ratios of gas phase species are included in* **Table S3**.

*The equations for the mass transfer coefficient $k_{mt}$ and the differential equations for the aqueous and gas phase concentrations were added to the supplement (**Eq-R1-R3;** Section 'Description of phase transfer) (Seinfeld and Pandis, 1998).*

**Referee Comment**: - I wonder what is cell viability and how long bacteria would survive in cloud droplets. Would this be something should be discussed and considered in the model with some sensitivity studies?

**Author response:** We thank the referee for pointing to the uncertainties associated with the biological activity of bacteria in clouds vs lab studies. We did not perform any sensitivity studies on the viability and survival time of bacteria cells in cloud water due to the large uncertainties associated with their estimates. Viable cells have been isolated from aerosol samples (Bovallius et al., 1978; Lighthart, 1997). Given that

living bacteria cells have been found in the atmosphere at many different places around the world, not only in clouds but also in particles, it seems reasonable to assume that cells survive for several hours or days (Fahlgren et al., 2010; Lighthart and Shaffer, 1994). Metabolic activity in cloud water was confirmed by assaying ATP (Adenosine 5'triphosphate) in cloud water samples (Amato et al., 2007b). ATP is considered as a key molecule of the energetic cell metabolism. We also would like to refer to a discussion on uncertainties in determining viable, cultivable and living cell fractions in our previous publication (Ervens and Amato, 2020). We will add some information on this topic throughout the manuscript; see our response to the next comment.

**Referee Comment**: Cloud droplets contain oxidants (e.g., OH , $H_2O_2$, etc.), so I wonder if they experience oxidative stress and eventually degraded? In other words, would $k_{bact}$ be time-dependent?

**Author response:** The referee is correct that high oxidant concentrations in the atmosphere and in particular in cloud water lead to stressful conditions for the bacteria cells, in addition to other factors such as UV exposure, low pH and temperature (Sattler et al., 2001). In addition, other effects such as substrate limitation may impact $k_{bact}$. Thus, given these different aspects, we respond to this referee comment in three parts:

a) Previous experiments in our research group have shown that cloud microorganisms can resist the oxidative stress due to the presence of reactive oxygen species, such as OH. The ADP/ATP ratio was monitored during such experiments which showed that even after exposure to oxidants over several hours, biodegradation rates were not significantly affected (Vaïtilingom et al., 2013). Based on these experiments, we can conclude that the presence of oxidants at typical levels as found in cloud water probably does not significantly affect $k_{bact}$ during the cloud cycles.

We add to the introduction (l.48, 52):

*The atmosphere is a stressful environment for microorganisms (low temperature, UV exposure, acidic pH, quick hydration/drying cycles and the presence of oxidants such as OH, $H_2O_2$) which might limit the survival time of cells in the atmosphere. Marker compounds such as adenosine 5'-triphosphate (ATP) (Amato et al., 2007c), rRNA 50 (Krumins et al., 2014) or mRNA (Amato et al., 2019) have been used to demonstrate metabolic activity in the atmosphere. The ADP/ATP ratio was monitored during such experiments which showed that even after exposure to oxidants over several hours, biodegradation rates were not significantly affected (Vaïtilingom et al., 2013).*

b) Recent experiments of airborne bacteria suggested that metabolic rates are a function of substrate availability (Krumins et al., 2014). Cloud water can be considered an oligotrophic medium. As the biodegradation experiments by (Husárová et al., 2011; Vaïtilingom et al., 2010, 2013) were performed in real or artificial cloud water with realistic solute concentrations, it is assumed that biodegradation occurs at the same rates in clouds. In the case of the CCN-derived water-soluble substrates that are not replenished within a cloud cycle, such substrate limitation might occur. However, as we do not have any data on the change of biodegradation rates as a function of substrate limitation, we do not perform any additional sensitivity studies. Our conclusions would not change, namely that the amount of CCN-derived organics that can be consumed by biodegradation is constrained by the fraction of cloud droplets that contain bacteria cells. Soluble gases are continuously replenished in the cloud water due to uptake processes from the gas phase (cf our discussion of Figure 5 in Section 3.1.2). If the availability of substrates with intermediate water-solubility decreases, $k_{bact}$ may decrease as well. However, as a response, the substrate concentration might increase due to a relatively more efficient replenishment from the gas phase, which may, in turn, lead to an increase in $k_{bact}$ again. Currently, no data is available on the absolute changes in $k_{bact}$ at different levels of substrate availability and/or on the time scales during which $k_{bact}$ might adjust to such conditions. Therefore, we do not think sensitivity studies on would be useful at this point due to the lack of parameter

constraints. We will add some discussion of such potential feedbacks in Section 4, highlighting the need of future experiments.

We add in the introduction (line 66):

*The biodegradation rate likely depends on the availability of the substrate in the aqueous phase. In the case of soluble substrates, it can be expected that uptake of soluble substrates from the gas phase leads to a continuous replenishment of the organics.*

We add in Section 5 (l. 591):

*Recent experiments of airborne bacteria suggested that metabolic rates are a function of substrate availability in oligotrophic media such as cloud droplets (Krumins et al., 2014). In the case of CCN-derived organics or those that are inefficiently replenished by uptake from the gas phase, substrate limitation might result in lower metabolic rates. Currently, no data is available on the absolute changes in $k_{bact}$ at different levels of substrate availability and/or on the time scales during which $k_{bact}$ might adjust to such conditions.*

c) The initial biodegradation rates of acetic and formic acid in lab experiments were observed to be very small if detectable at all (Herlihy et al., 1987). Only after incubation of several hours or days, biodegradation was observed. However, such incubation periods are likely due to the adjustment of the bacteria cells to the ambient conditions in the lab experiments. As bacteria cells are continuously exposed to water, solutes, substrates etc in the atmosphere, such incubation periods are unlikely to occur in clouds. Indeed, experiments in our lab performed in real cloud water did not exhibit such lag periods (e.g. (Amato et al., 2007a; Vaïtilingom et al., 2010, 2013) which supports this hypothesis. While we touched briefly on this topic in Section 4, we will extend this discussion in the revised manuscript adding the latter references.

While some text on the incubation effects was already included in Section 4.3, the following references will be added there (line 535):

*While the latter was regarded as being inefficient due to long incubation time as observed in lab experiments, more recent experiments suggest that such incubation times are likely not occurring in the atmosphere where bacteria cells are continuously exposed to water and substrates* (Amato et al., 2007a; Vaïtilingom et al., 2010, 2013).

We would like to thank the referee for their positive and insightful comments on the manuscript. Below is our point-by-point response to the comments. Our responses are in black, manuscript text is in italic with new/modified text is marked in blue. The line numbers cited here refer to those in the clean manuscript version.

**Referee comment:** Section 2.1: The model description is too concise and could be improved a bit. Which 26 species are transferred between the gas and aqueous phases? It would be worth showing the coupled mass transfer ODEs with Schwartz's treatment.

**Author response:** We agree with the referee that the model description in the section 2.1 was very short. Since also the other referee asked for more detail on the model, we added three tables into the supplement with details on our aqueous phase chemical mechanism (Table S1), phase transfer parameters (Table S2) and initial mixing ratios and concentrations (Table S3).

We will add the following text at the beginning of Section 2.1:

*We use a multiphase box model with detailed gas and aqueous phase chemistry (75 species, 44 gas phase reactions, 31 aqueous reactions).* *The chemical aqueous phase mechanism with rate constants is listed in **Table S1**. The chemical gas phase mechanism is based on the NCAR Master mechanism (Aumont et al., 2000; Madronich and Calvert, 1989). The two phases are coupled by 26 phase transfer processes which is described kinetically based on the resistance model by Schwartz (1986). The parameters describing the phase transfer of the soluble species are presented in **Table S2**. In addition, the initial gas phase mixing ratios are listed in **Table S3**.The equations for the mass transfer coefficient k_{mt} and the differential equations for the aqueous and gas phase concentrations can be found in the supplement **(Eq-R1-R3; Section 'Description of phase transfer)** (Seinfeld and Pandis, 1998).*

In addition, we add the following set of equations for the description of the phase transfer to the supplement:

$$ k_{mt} = \left[ \frac{r_d^2}{3D_g} + \frac{r_d}{3\alpha} \sqrt{\frac{2\pi M_g}{RT}} \right]^{-1} \qquad \text{(Eq-R1)} $$

*Whereas*

*$r_d$ = cloud droplet radius [cm]*

*$D_g$ = gas phase diffusion coefficient [cm s$^{-1}$]*

*$\alpha$ = mass accommodation coefficient (dimensionless)*

*$M_g$ = molecular weight of gas [g mol$^{-1}$]*

*$R$ = constant for ideal gases (8.314 ·10$^7$ erg mol$^{-1}$ K$^{-1}$)*

*$T$ = temperature [K]*

*The mass transfer coefficient is then applied to determine the sink and source terms due to phase transfer of soluble species:*

$$ \frac{dc_g}{dt} = k_{mt} \cdot LWC \cdot \left[ \frac{c_{aq}}{LWC \cdot K_{H(eff)} \cdot R' \cdot T} - c_g \right] + (P_{gaschem} - L_{gaschem}) \qquad \text{(Eq-R2)} $$

$$\frac{dc_{aq}}{dt} = k_{mt} \cdot LWC \cdot \left[ c_g - \frac{c_{aq}}{LWC \cdot K_{H(eff)} \cdot R' \cdot T} \right] + (P_{aqchem} - L_{aqchem}) \qquad (Eq\text{-}R3)$$

*whereas both the gas and the aqueous phase concentrations have units of mol g(air)$^{-1}$; LWC is the liquid water content in g/m$^3$, $K_{H(eff)}$ is the effective Henry's law constant in M atm$^{-1}$, $R'$ is the ideal-gas constant (8.314.10$^7$ erg mol$^{-1}$ K$^{-1}$), T is the temperature [K] and P and L are the rates of the chemical production and loss reactions in the gas and aqueous phases, respectively.*

**Referee comment:** Is the size class same as the size bin? On line 101, it is stated that the 5 um < Ddroplet < 20 um, but then one droplet size class has D$_{droplet}$ = 20 um. Should the range be changed to 5 um < D$_{droplet}$ ≤ 20 um? Also, why is only the last size class allowed to have bacteria cells?

**Author response:** The referee is correct that the size range of the droplets should be written as 5 μm ≤ D$_{droplet}$ ≤ 30 μm .

In our box model, we chose a drop size spectrum with diameters of 5 to 30 μm. The bacteria are only in one of the drop classes (diameter 20 μm). Note that other box model studies usually only consider a single drop size (Deguillaume et al., 2004; Ervens et al., 2003; Tilgner et al., 2013). However, as we have shown previously that the drop size may impact the OH(aq) concentration and distribution (Ervens et al., 2014), we used a polydisperse drop size distribution. While also the uptake of organic compounds may be drop-size-dependent, we did not further explore this effect as it would not add significantly to our conclusions.

**Referee comment:** Line 105-106: Which organic compound? Is the model run separately for each individual compound? What other inorganic species were considered in the model? What was the cloud water pH in the model simulations discussed in section 3?

**Author response:** We removed the sentence in line 105/106 (now line: 116) and clarified now which species are initialized in our model. We provide more detail on our model in Table S1 (detailed aqueous phase chemical mechanism) ,Table S2 (phase transfer parameters) and Table S3 (initial concentration of different species).

**Referee comment**: While the degradation rate constant remains constant between pH values 5 and 8, could it change at lower pH?

**Author response:**

The referee is correct to point out the possible effects of acidity on bacterial activity. Several studies investigated the acid tolerance mechanism of different microorganisms, e.g., (Casal et al., 2016; Patel et al., 2006) by decarboxylation, deamination (Noh et al., 2018), or cell membrane modification (Zhang et al., 2011). In addition, some bacteria can develop an acid resistance system to survive under acidic conditions (pH=2.5) (Lu et al., 2013). However, these strategies do not necessarily imply that the bacteria maintain the same biodegradation activities at high acidity but they allow survival of the cells in the atmosphere.

When exposed to very broad ranges of external pHs, bacteria can control their intracellular pH (~6.5 -7) by internal buffering (Delort et al., 2017). As biodegradation occurs inside the cell, it takes place at these (nearly) neutral conditions. The efficiency of buffering decreases at extreme conditions, e.g., pH < 2 or pH > 10 (Guan and Liu, 2020). However, such pH range is not representative for cloud water where more moderate pH values (~ 3 – 6) are typically found (Deguillaume et al., 2014).

We will modify the text in Section 2.2.2. as follows:

*Experiments with 17 different cloud bacteria in artificial cloud water with pH = 5.0 and pH = 6.5 showed also nearly identical results* (Vaïtilingom et al., 2011). . *Similar results were shown by Razika et al. (2010) who demonstrated that biodegradation rates of phenol by Pseudomonas aeruginosa were very similar when incubated at pH = 5.8, 7.0 and 8.0, respectively. When exposed to very broad ranges of external pHs, bacteria can control their intracellular pH (~6.5 -7) by internal buffering (Delort et al., 2017). As biodegradation occurs inside the cell, it takes place at these (nearly) neutral conditions. The efficiency of buffering decreases at extreme conditions, e.g., pH < 2 or pH > 10 (Guan and Liu, 2020). However, such pH range is not representative for cloud water where more moderate pH values (~ 3 – 6)(Deguillaume et al., 2014) are typically found.*

In addition, we add in the conclusion section (l.596):

*In addition, the biodegradation rates may be affected under highly acidic conditions. Some studies demonstrated that some bacteria can develop an acid resistance to survive under acidic conditions ((Lu et al., 2013). However, these strategies do not necessarily imply that the bacteria maintain the same biodegradation activities at high acidity but they allow survival of the cells in the atmosphere. It can be expected that internal buffering of bacteria cells allows them to maintain their metabolic activity over wide pH ranges (~ 3 to 6) as found in cloud water. Therefore, we do not consider a potential pH dependency of biodegradation rates in our model studies.*

**Referee comment:** Figures 2, 3, 6, 7, 8, 10 are presently displayed as three-dimensional plots, which I found a little difficult to read as parts of the plots are obscured by curved surfaces. Since the Z-axis and colors represent the same dimension, I suggest replotting these figures as two-dimensional color contour plots. This would greatly improve the quality and readability of the figures.

**Author response:** Thank you for this suggestion, we agree with this suggestion and show in the revised version the figures 2,3,6,7,8 and 10 and Figures S1 and S2 as two-dimensional color contour plots to improve the quality and readability of the figures as suggested.

**Referee comment**: Table 2: There's an extra multiplication symbol in the 5 C Experimental rate column on 12th row.

**Author response:** Corrected

Aumont, B., Madronich, S., Bey, I. and Tyndall, G.: Contribution of Secondary VOC to the Composition of Aqueous Atmospheric Particles: A Modeling Approach, J. Atmos. Chem., 35(1), 59–75, doi:10.1023/a:1006243509840, 2000.

Casal, M., Queirós, O., Talaia, G., Ribas, D. and Paiva, S.: Carboxylic acids plasma membrane transporters in saccharomyces cerevisiae, in Advances in Experimental Medicine and Biology., 2016.

Deguillaume, L., Leriche, M., Monod, A. and Chaumerliac, N.: The role of transition metal ions on HOx radicals in clouds: a numerical evaluation of its impact on multiphase chemistry, Atmos. Chem. Phys., doi:10.5194/acp-4-95-2004, 2004.

Deguillaume, L., Charbouillot, T., Joly, M., Vaïtilingom, M., Parazols, M., Marinoni, A., Amato, P., Delort, A. M., Vinatier, V., Flossmann, A., Chaumerliac, N., Pichon, J. M., Houdier, S., Laj, P., Sellegri, K., Colomb, A., Brigante, M. and Mailhot, G.: Classification of clouds sampled at the puy de Dôme (France) based on 10 yr of monitoring of their physicochemical properties,

Atmos. Chem. Phys., doi:10.5194/acp-14-1485-2014, 2014.

Delort, A.-M., Deguillaume, L., Renard, P., Vinatier, V., Canet, I., Vaïtilingom, M. and Chaumerliac, N.: Impacts on Cloud Chemistry, in Microbiology of Aerosols., 2017.

Ervens, B., George, C., Williams, J. E., Buxton, G. V., Salmon, G. A., Bydder, M., Wilkinson, F., Dentener, F., Mirabel, P., Wolke, R. and Herrmann, H.: CAPRAM 2.4 (MODAC mechanism): An extended and condensed tropospheric aqueous phase mechanism and its application, J. Geophys. Res. Atmos., 108(D14), doi:10.1029/2002jd002202, 2003.

Ervens, B., Carlton, A. G., Turpin, B. J., Altieri, K. E., Kreidenweis, S. M. and Feingold, G.: Secondary organic aerosol yields from cloud-processing of isoprene oxidation products, Geophys. Res. Lett., 35(2), doi:10.1029/2007GL031828, 2008.

Ervens, B., Sorooshian, A., Lim, Y. B. and Turpin, B. J.: Key parameters controlling OH-initiated formation of secondary organic aerosol in the aqueous phase (aqSOA), J. Geophys. Res. - Atmos., 119(7), 3997–4016, doi:10.1002/2013JD021021, 2014.

Guan, N. and Liu, L.: Microbial response to acid stress: mechanisms and applications, Appl. Microbiol. Biotechnol., 51–65, doi:10.1007/s00253-019-10226-1, 2020.

Ip, H. S. S., Huang, X. H. H. and Yu, J. Z.: Effective Henry's law constants of glyoxal, glyoxylic acid, and glycolic acid, Geophys. Res. Lett., doi:10.1029/2008GL036212, 2009.

Johnson, B. J., Betterton, E. A. and Craig, D.: Henry's Law coefficients of formic and acetic acids, J. Atmos. Chem., 24(2), 113--119, doi:10.1007/BF00162406, 1996.

Lu, P., Ma, D., Chen, Y., Guo, Y., Chen, G. Q., Deng, H. and Shi, Y.: L-glutamine provides acid resistance for Escherichia coli through enzymatic release of ammonia, Cell Res., doi:10.1038/cr.2013.13, 2013.

Madronich, S. and Calvert, J. G.: No Title, NCAR Tech. Note TN-474+STR, 1989.

Noh, M. H., Lim, H. G., Woo, S. H., Song, J. and Jung, G. Y.: Production of itaconic acid from acetate by engineering acid-tolerant Escherichia coli W, Biotechnol. Bioeng., doi:10.1002/bit.26508, 2018.

Patel, M. A., Ou, M. S., Harbrucker, R., Aldrich, H. C., Buszko, M. L., Ingram, L. O. and Shanmugam, K. T.: Isolation and characterization of acid-tolerant, thermophilic bacteria for effective fermentation of biomass-derived sugars to lactic acid, Appl. Environ. Microbiol., doi:10.1128/AEM.72.5.3228-3235.2006, 2006.

Sander, R.: Compilation of Henry's law constants (version 4.0) for water as solvent, Atmos. Chem. Phys., 15(8), 4399–4981, doi:10.5194/acp-15-4399-2015, 2015.

Seinfeld, J. H. and Pandis, S. N.: Atmospheric Chemistry and Physics, John Wiley & Sons, New York., 1998.

Tilgner, A., Bräuer, P., Wolke, R. and Herrmann, H.: Modelling multiphase chemistry in deliquescent aerosols and clouds using CAPRAM3.0i, J. Atmos. Chem., 70(3), 221–256, doi:10.1007/s10874-013-9267-4, 2013.

Zhang, J. G., Liu, X. Y., He, X. P., Guo, X. N., Lu, Y. and Zhang, B. run: Improvement of acetic

acid tolerance and fermentation performance of Saccharomyces cerevisiae by disruption of the FPS1 aquaglyceroporin gene, Biotechnol. Lett., doi:10.1007/s10529-010-0433-3, 2011.

[revised manuscript text omitted]
* | 8×10⁴ | 0.02 | 1.9×10⁻²⁰ | 8.1×10⁻²¹ | 7.7×10⁻¹¹ | 3.2×10⁻¹¹ | 4.8×10⁻¹⁸ | 2.1×10⁻¹⁸ | 1 |
| | *Pseudomonas* sp. | | | 1.4×10⁻¹⁹ | 8.6×10⁻²⁰ | 5.6×10⁻¹⁰ | 3.4×10⁻¹⁰ | 3.7×10⁻¹⁷ | 2.3×10⁻¹⁷ | |
| | *Frigoribacterium* sp. | | | 6.4×10⁻²¹ | 6.4×10⁻²¹ | 2.6×10⁻¹¹ | 2.6×10⁻¹¹ | 1.7×10⁻¹⁸ | 1.7×10⁻¹⁸ | |
| | *Bacillus* sp. | | | 2.0×10⁻²⁰ | 3.1×10⁻²¹ | 8.1×10⁻¹¹ | 1.2×10⁻¹¹ | 5.4×10⁻¹⁸ | 8.4×10⁻¹⁹ | |
| Formate | *Sphingomonas* sp. | 10⁹ | 0.02 | 9.2×10⁻²¹ | 3.1×10⁻²⁰ | 4.6×10⁻⁷ | 1.6×10⁻⁶ | 3.0×10⁻¹⁴ | 1.0x×0⁻¹³ | 2 |
| | *P graminis* | | | 1.3×10⁻¹⁹ | 9.6×10⁻²⁰ | 6.5×10⁻⁶ | 4.8×10⁻⁶ | 4.3×10⁻¹³ | 3.2×10⁻¹³ | |
| | *Pseudomonas* sp. | | | 4.6×10⁻²⁰ | 8.6×10⁻²¹ | 2.3×10⁻⁶ | 4.3×10⁻⁷ | 1.5×10⁻¹³ | 2.8×10⁻¹⁴ | |
| | *P viridiflava* | | | 1.6×10⁻¹⁹ | 4.7×10⁻²⁰ | 8.1×10⁻⁶ | 2.3×10⁻⁶ | 5.4×10⁻¹³ | 1.5×10⁻¹³ | |
| | *Rhodococcus* sp. | 10⁶ | 2×10⁻⁵ | 8.0×10⁻¹⁹ | 4.0×10⁻¹⁹ | 4.0×10⁻⁵ | 2.0×10⁻⁵ | 2.6×10⁻¹² | 1.3×10⁻¹² | 3 |
| | *Pseudomonas* sp. | | | 1.5×10⁻¹⁸ | 8.0×10⁻¹⁹ | 7.5×10⁻⁶ | 4.0×10⁻⁵ | 5.0×10⁻¹³ | 3.4×10⁻¹³ | |
| | *P syringae* | | | 2.3×10⁻¹⁸ | 2.0×10⁻¹⁸ | 1.1×10⁻⁴ | 1.0×10⁻⁴ | 7.3×10⁻¹² | 6.6×10⁻¹² | |
| | *P graminis* | | | 5.0×10⁻¹⁸ | 1.0x×10⁻¹⁸ | 2.5×10⁻⁴ | 5.0×10⁻⁵ | 1.6×10⁻¹¹ | 3.3×10⁻¹² | |
| | Various microorganisms | 8×10⁴ | 43×10⁻⁶ | 2.1×10⁻¹⁸ | | 4.1×10⁻⁶ | | 2.7×10⁻¹³ | | 4 |
| Acetate | *Sphingomonas* sp. | 10⁹ | 0.02 | 2.7×10⁻²⁰ | 1.6×10⁻²² | 1.3×10⁻⁶ | 8.2×10⁻⁹ | 1.1×10⁻¹³ | 5.4×10⁻¹⁶ | 2 |
| | *P graminis* | | | 3.0×10⁻¹⁹ | 3.0×10⁻²⁰ | 1.5×10⁻⁵ | 1.5×10⁻⁶ | 1.0×10⁻¹² | 1.0×10⁻¹³ | |
| | *Pseudomonas* sp. | | | 2.6×10⁻²⁰ | 1.7×10⁻²⁰ | 1.3×10⁻⁶ | 8.8×10⁻⁷ | 8.7×10⁻¹⁴ | 5.3×10⁻¹⁴ | |
| | *P viridiflava* | | | 5.6×10⁻²⁰ | 1.1×10⁻²⁰ | 2.8×10⁻⁶ | 6.0×10⁻⁷ | 1.8×10⁻¹³ | 4.0×10⁻¹⁴ | |
| | *Rhodococcus* sp. | 10⁶ | 2×10⁻⁵ | 5.0×10⁻¹⁸ | 1.0×10⁻¹⁸ | 2.5×10⁻⁴ | 5.0×10⁻⁵ | 1.6×10⁻¹¹ | 3.3×10⁻¹² | 3 |
| | *Pseudomonas* sp. | | | 1.3×10⁻¹⁸ | 6.0×10⁻¹⁹ | 6.7×10⁻⁵ | 3.0×10⁻⁵ | 4.5×10⁻¹² | 2.0×10⁻¹² | |
| | *P syringae* | | | 8.6×10⁻¹⁹ | 2.0×10⁻¹⁹ | 4.3×10⁻⁵ | 1.0×10⁻⁵ | 2.8×10⁻¹² | 6.6×10⁻¹³ | |
| | *P graminis* | | | 4.0×10⁻¹⁹ | 1.0×10⁻¹⁹ | 2.0×10⁻⁵ | 5.0×10⁻⁶ | 1.3×10⁻¹² | 3.3×10⁻¹³ | |
| | Various microorganisms | 8×10⁴ | 2.5×10⁻⁶ | 1.94×10⁻¹⁸ | | 6.25×10⁻³ | | 4.16×10⁻¹⁰ | | 4 |
| Succinate | *Sphingomonas* sp. | 10⁹ | 0.02 | 2.6×10⁻²⁰ | 1.0×10⁻²⁰ | 1.3×10⁻⁶ | 5.4×10⁻⁷ | 8.6×10⁻¹⁴ | 3.6×10⁻¹⁴ | 2 |
| | *P graminis* | | | 1.0×10⁻¹⁹ | 9.8×10⁻²⁰ | 5.2×10⁻⁶ | 4.9×10⁻⁶ | 3.4×10⁻¹³ | 3.2×10⁻¹³ | |
| | *Pseudomonas* sp. | | | 1.4×10⁻²⁰ | 1.8×10⁻²⁰ | 7.0×10⁻⁷ | 9.2×10⁻⁷ | 4.7×10⁻¹⁴ | 6.1×10⁻¹⁴ | |
| | *P viridiflava* | | | 2.9×10⁻²⁰ | 6.1×10⁻²⁰ | 1.4×10⁻⁶ | 3.0×10⁻⁶ | 9.7×10⁻¹⁴ | 2.0×10⁻¹³ | |
| | *Rhodococcus* sp. | 10⁶ | 2×10⁻⁵ | 5.0×10⁻²⁰ | 4.0×10⁻²⁰ | 2.5×10⁻⁶ | 2.0×10⁻⁶ | 1.6×10⁻¹³ | 1.3×10⁻¹³ | 3 |
| | *Pseudomonas* sp. | | | 1.5×10⁻¹⁹ | 2.0×10⁻²⁰ | 7.5×10⁻⁶ | 1.0×10⁻⁶ | 5.0×10⁻¹³ | 6.6×10⁻¹⁴ | |
| | *P syringae* | | | 6.8×10⁻¹⁹ | 1.7×10⁻¹⁹ | 3.4×10⁻⁵ | 8.8×10⁻⁶ | 2.2×10⁻¹² | 5.8×10⁻¹³ | |
| | *P graminis* | | | 5.0×10⁻¹⁹ | 1.0×10⁻¹⁹ | 2.5×10⁻⁵ | 5.0×10⁻⁶ | 1.6×10⁻¹² | 3.3×10⁻¹³ | |
| | Various microorganisms | 8×10⁴ | 3.1×10⁻⁶ | 5.6×10⁻¹⁹ | | 1.4×10⁻⁵ | | 9.6×10⁻¹³ | | 4 |
| Malonate | Various microorganisms | 8×10⁴ | 3.1×10⁻⁶ | 5.2×10⁻¹⁹ | | 1.3×10⁻⁵ | | 9.0×10⁻¹³ | | 4 |
| Catechol | *R enclensis* | 10⁷ | 0.0001 | 4.1×10⁻¹⁹ | | 4.1×10⁻⁸ | | 2.7×10⁻¹⁵ | | 5 |
| Phenol | *R enclensis* | 10⁹ | 0.0001 | 5.0×10⁻²⁰ | | 5.0×10⁻⁴ | | 3.3×10⁻¹¹ | | 5 |
| Methanol | *P graminis* | 8×10⁴ | 0.02 | 5.6×10⁻²² | - | 2.2×10⁻¹² | - | 1.5×10⁻¹⁹ | 0 | 1 |
| | *P syringae* | | | 5.7×10⁻²¹ | 5.8×10⁻²² | 2.3×10⁻¹¹ | 2.3×10⁻¹² | 1.5×10⁻¹⁸ | 1.5×10⁻¹⁹ | |
| | *Frigoribacterium* sp. | | | 3.5×10⁻²³ | 2.5×10⁻²³ | 1.4×10⁻¹³ | 1.0×10⁻¹³ | 9.4×10⁻²¹ | 6.7×10⁻²¹ | |
| | *Bacillus* sp. | | | 2.9×10⁻²¹ | - | 1.1×10⁻¹¹ | - | 7.8×10⁻¹⁹ | 0 | |

[revised manuscript text omitted]

**Description of phase transfer in the box model**

The mass transfer coefficient $k_{mt}$ [s$^{-1}$] can be expressed as (Seinfeld and Pandis, 1998)

$$k_{mt} = \left[ \frac{r_d^2}{3D_g} + \frac{r_d}{3\alpha} \sqrt{\frac{2\pi M_g}{RT}} \right]^{-1} \qquad\qquad\text{(Eq-R1)}$$

Whereas

$r_d$ = cloud droplet radius [cm]

$D_g$ = gas phase diffusion coefficient [cm s$^{-1}$]

$\alpha$ = mass accommodation coefficient (dimensionless)

$M_g$ = molecular weight of gas [g mol$^{-1}$]

R = constant for ideal gases (8.314 $\cdot 10^7$ erg mol$^{-1}$ K$^{-1}$)

T = temperature [K]

The mass transfer coefficient is then applied to determine the sink and source terms due to phase transfer of soluble species:

$$\frac{dc_g}{dt} = k_{mt} \cdot LWC \cdot \left[ \frac{c_{aq}}{LWC \cdot K_{H(eff)} \cdot R' \cdot T} - c_g \right] + (P_{gaschem} - L_{gaschem}) \qquad\text{(Eq-R2)}$$

$$\frac{dc_{aq}}{dt} = k_{mt} \cdot LWC \cdot \left[ c_g - \frac{c_{aq}}{LWC \cdot K_{H(eff)} \cdot R' \cdot T} \right] + (P_{aqchem} - L_{aqchem}) \qquad \text{(Eq-R3)}$$

whereas both the gas and the aqueous phase concentrations have units of mol g(air)$^{-1}$; LWC is the liquid water content in g/m$^3$, $K_{H(eff)}$ is the effective Henry's law constant in M atm$^{-1}$, R' is the ideal-gas constant (8.314.10$^7$ erg mol$^{-1}$ K$^{-1}$), T is the temperature [K] and P and L are the rates of the chemical production and loss reactions in the gas and aqueous phases, respectively.

**Description of model parameters**

**Table S1**. Aqueous phase chemical mechanism : aqueous phase irreversible reactions with rate constants k and temperature dependencies Ea/R where available and aqueous phase equilibria reactions (Ervens et al., 2003, 2008) .

| Reactions | Reactants | Products | k [M$^{-1}$ s$^{-1}$] | Ea/R [K] |
|---|---|---|---|---|
| | | Aqueous phase irreversible reactions | | |
| R$_1$ | SO$_2$ + O$_3$ | S(VI) + O$_2$ | 2.4×10$^4$ | |
| R$_2$ | HSO$_3^-$ + O$_3$ | S(VI) + O$_2$ | 3.7×10$^5$ | 5530 |
| R$_3$ | SO$_3^{2-}$ + O$_3$ | S(VI) + O$_2$ | 1.5×10$^9$ | 5280 |
| R$_4$ | H$_2$O$_2$ + HSO$_3$ | S(VI) + H$_2$O | 7.2×10$^7$ | 14000 |
| R$_5$ | HO$_2$ + HO$_2$ | H$_2$O$_2$ + O$_2$ | 8.3×10$^5$ | 2720 |
| R$_6$ | O$_2^-$ + HO$_2$ | H$_2$O$_2$ + O$_2$ | 9.7×10$^7$ | 1060 |
| R$_7$ | OH + CH$_2$O | HO$_2$ + HCOOH | 1×10$^9$ | 1020 |
| R$_8$ | OH + CH$_3$OOH | CH$_3$O$_2$ + H$_2$O | 2.4×10$^7$ | 1680 |
| R$_9$ | OH + CH$_3$OOH | HO$_2$ + HCOOH | 6×10$^6$ | 1680 |
| R$_{10}$ | O$_3$ + O$_2^-$ (+ H$^+$) | OH + 2 O$_2$ | 1.5×10$^9$ | 2200 |
| R$_{11}$ | OH + CHOCHO | HO$_2$ + CHOCOOH | 1.1×10$^9$ | 1516 |
| R$_{12}$ | OH + CHOCOOH | HO$_2$ + H$_2$C$_2$O$_4$ | 3.6×10$^8$ | 1000 |
| R$_{13}$ | OH + CHOCOO$^-$ | H$_2$C$_2$O$_4^-$ | 2.9×10$^9$ | 4300 |
| R$_{14}$ | OH + CH$_3$COCHO | HO$_2$ + 0.92 CH$_3$COCOOH + 0.08 CHOCOOH | 1.1×10$^9$ | 1600 |
| R$_{15}$ | OH + CH$_2$(OH)CHO | HO$_2$ + CH$_2$OHCOOH | 1.2×10$^9$ | |
| R$_{16}$ | OH + CH$_2$OHCOOH | HO$_2$ + CHOCOOH | 1.2×10$^9$ | |
| R$_{17}$ | OH + C$_2$O$_4^{2-}$ | O$_2^-$ + 2 CO$_2$ + OH$^-$ | 1.6×10$^8$ | 4300 |
| R$_{18}$ | OH + HC$_2$O$_4^-$ | HO$_2$ + 2 CO$_2$ + OH$^-$ | 1.9×10$^8$ | 2800 |
| R$_{19}$ | OH + H$_2$C$_2$O$_4$ | HO$_2$ + 2 CO$_2$ + H$_2$O | 1.4×10$^6$ | |
| R$_{20}$ | OH + CH$_3$C(O)COOH | HO$_2$ + CO$_2$ + CH$_3$COO$^-$ | 7×10$^8$ | |
| R$_{21}$ | OH + CH$_3$COCOOH | HO$_2$ + H$_2$O + CH$_3$COOH | 1.2×10$^8$ | |
| R$_{22}$ | OH + HCOO$^-$ | HO$_2$ + CO$_2$ + H$_2$O | 3.2×10$^9$ | 1000 |
| R$_{23}$ | OH + HCOOH | HO$_2$ + CO$_2$ + H$_2$O | 1.3×10$^8$ | 1000 |
| R$_{24}$ | OH + CH$_3$COO$^-$ | HO$_2$ + OH$^-$ + 0.15 CH$_2$O + 0.85 CHOCOOH | 1×10$^8$ | 1800 |
| R$_{25}$ | OH + CH$_3$COOH | HO$_2$ + H$_2$O + 0.15 CH$_2$O+ 0.85 CHOCOOH | 1.5×10$^7$ | 1330 |
| R$_{26}$ | CH$_3$O$_2$ + CH$_3$O$_2$ | CH$_2$O + CH$_3$OH + HO$_2$ | 1.7×10$^8$ | 2200 |
| R$_{27}$ | H$_2$O$_2$ + OH | HO$_2$ + H$_2$O | 3×10$^7$ | 1680 |
| R$_{28}$ | OH + CH$_3$CHO | HO$_2$ + H$_2$O + CH$_3$COOH | 3.6×10$^9$ | 580 |
| R$_{29}$ | O$_2^-$ + CH$_3$(CO)OO | CH$_3$COOH | 1×10$^9$ | |
| R$_{30}$ | CH$_3$C(O)OO + CH3C(O)OO | 2 CH$_3$O$_2$ + 2 CO$_2$ | 1.5×10$^8$ | |
| R$_{31}$ | OH + HO-CH$_2$-CHO | HO$_2$ + CH$_2$(OH)COOH | 5×10$^8$ | |

| $R_{32}$ | OH + WSOC | WSOC + HO$_2$ | $3.8\times10^8$ | |
|---|---|---|---|---|
| \multicolumn Aqueous phase equilibria | | | | |
| E | | | $K_a$ [M] | |
| $E_1$ | H$_2$O | OH$^-$+H$^+$ | $1.0\times10^{-14}$ | |
| $E_2$ | HO$_2$ | O$_2^-$ + H$^+$ | $1.60\times10^{-5}$ | |
| $E_3$ | CHOCOOH | CHOCOO$^-$+H$^+$ | $6.60\times10^{-4}$ | |
| $E_4$ | HCOOH | HCOO$^-$ + H$^+$ | $1.77\times10^{-4}$ | |
| $E_5$ | CH$_3$COCOOH | CH$_3$COCOO$^-$+ H$^+$ | $4.07\times10^{-3}$ | |
| $E_6$ | CH$_3$COOH | CH$_3$COO$^-$+ H$^+$ | $1.77\times10^{-5}$ | |
| $E_7$ | H$_2$C$_2$O$_4$ | HC$_2$O$_4^-$+H$^+$ | $6.40\times10^{-2}$ | |
| $E_8$ | HC$_2$O$_4^-$ | C$_2$O$_4^{2-}$+ H$^+$ | $5.25\times10^{-5}$ | |
| $E_9$ | HNO$_3$ | NO$_3^-$+ H$^+$ | 22 | |
| $E_{10}$ | HOCH$_2$CH$_2$OH | HOCH$_2$CH$_2$O$^-$+ H$^+$ | $1.54\times10^{-4}$ | |
| $E_{11}$ | SO$_2$ +H$_2$O | HSO$_3^-$+H$^+$ | 0.013 | |
| $E_{12}$ | HSO$_3^-$ | SO$_3^{2-}$+H$^+$ | $6.60\times10^{-8}$ | |
| $E_{13}$ | H$_2$SO$_4$ | HSO$_4^-$ + H$^+$ | 1000 | |
| $E_{14}$ | HSO$_4^-$ | SO$_4^{2-}$ + H$^+$ | 0.102 | |
| $E_{15}$ | CO$_2$(aq)+H$_2$O | HCO$_3^-$ + H$^+$ | $7.70\times10^{-7}$ | |
| $E_{16}$ | HCO$_3^-$ | CO$_3^{2-}$ + H$^+$ | $4.84\times10^{-11}$ | |
| $E_{17}$ | NH$_3$ | NH$_4^+$ + OH$^-$ | $1.76\times10^{-5}$ | |
| $E_{18}$ | H$_2$O | H$^+$ + OH$^-$ | $1\times10^{-14}$ | |

**Table S2.** Phase transfer parameters of soluble species to calculate the mass transfer coefficient (Eq-S1). M$_g$: molecular weight (g/mol), α: mass accommodation coefficient (dimension less), D$_g$: gas phase diffusion coefficients (cm s$^{-1}$) and K$_H$: Henry's law constant [M atm$^{-1}$]

| Species | M$_g$ [g mol$^{-1}$] | α$^{[1]}$ | D$_g$ [cm s$^{-1}$]$^{[1]}$ | K$_H$ [M atm$^{-1}$] |
|---|---|---|---|---|
| O$_3$ | 48 | 0.05 | 0.148 | $1.14\times10^{-2}$[1] |
| H$_2$O$_2$ | 34 | 0.1 | 0.11 | $1.02\times10^5$[1] |
| HO | 17 | 0.05 | 0.153 | 25[1] |
| HO$_2$ | 33 | 0.01 | 0.104 | $9\times10^3$[1] |
| HCHO | 30 | 0.02 | 0.164 | $4.99\times10^3$[1] |
| CH$_3$O$_2$ | 47 | 0.0038 | 0.135 | 310[1] |
| CH$_3$OOH | 48 | 0.0038 | 0.135 | 310 |
| HNO$_3$ | 63 | 0.054 | 0.132 | $2.1\times10^5$[1] |
| N$_2$O$_5$ | 108 | 0.0037 | 0.110 | 1.4[1] |
| Hydroxyaldehyde | 60 | 0.03 | 0.195 | $4.1\times10^4$[3] |
| Glyoxal | 58 | 0.023 | 0.115 | $4.19\times10^5$[1] |
| Methylglyoxal | 72 | 0.1 | 0.115 | $3.2\times10^4$ [3] |
| HCOOH | 74 | 0.012 | 0.153 | $1.77\times10^{-4}$[1] |
| Acetic acid | 46 | 0.1 | 0.1 | $4\times10^3$[4] |
| Glyoxylic acid | 60 | 0.019 | 0.124 | $9\times10^3$[2] |
| Glycolic acid | 76 | 0.1 | 0.1 | $9\times10^3$[2] |
| Pyruvate | 88 | 0.1 | 0.1 | $3.11\times10^5$[3] |
| Oxalate | 90 | 0.1 | 0.1 | $9\times10^3$[3] |
| Hydroxyketone | 88 | 0.1 | 0.1 | 100[3] |

| | | | | |
|---|---|---|---|---|
| $CH_3O_3$ | 75 | 0.1 | 0.1 | $669$[3] |
| Aldehyde | 44 | 0.1 | 0.1 | $11.4$[3] |
| $SO_2$ | 64 | 0.035 | 0.128 | $1.23$[3] |
| $CO_2$ | 44 | $2.e^{-4}$ | 0.155 | $3.11 \times 10^{-2}$[1] |
| Glycolaldehyde | 58 | 0.1 | 0.1 | $4.1 \times 10^{4}$[3] |
| $C_2H_5OOH$ | 62 | 0.1 | 0.1 | $310$[1] |
| Organic compound that reacts with OH and is consumed by bacteria | 150 | 0.1 | 0.1 | $10^2 - 10^9$ |

[1]: (Ervens et al., 2003), [2]: (Ip et al., 2009), [3]: (Sander, 2015), [4]: (Johnson et al., 1996)

**Table S3**. Initial gas phase mixing ratios of gas phase species and the concentration of species only in the aqueous phase; all other compounds considered in the mechanism were not initialized

| Species formula | Species name | Mixing ratio [ppb] |
|---|---|---|
| Gases with phase transfer into the aqueous phase | | |
| $O_3$ | ozone | 39 |
| $H_2O_2$ | hydrogen peroxide | 1.202 |
| OH | hydroxyl radical | $3.05 \times 10^{-11}$ |
| $HO_2$ | hydroperoxyl radical | $9.07 \times 10^{-3}$ |
| HCHO | formaldehyde | 2.519 |
| $CH_3O_2$ | methylperoxy radical | $1.38 \times 10^{-3}$ |
| $CH_3OOH$ | methyl hydrogen peroxide | 0.211 |
| $HNO_3$ | nitric acid | 0.397 |
| $N_2O_5$ | dinitrogen pentoxide | $4.806 \times 10^{-4}$ |
| $CH_2(OH)CHO$ | hydroxy acetaldehyde | 0.437 |
| CHOCHO | glyoxal | 0.218 |
| $CH_3COCHO$ | methyl glyoxal | 0.190 |
| HCOOH | formic acid | $2.239 \times 10^{-3}$ |
| HAc | Acetic acid | 0.198 |
| $CH_3CO(OO)$ | acetylperoxy radical | $5.5 \times 10^{-5}$ |
| $SO_2$ | sulfur dioxide | 0.150 |
| $CH_3CHO$ | acetaldehyde | 0.409 |
| $CO_2$ | carbon dioxide | $3.96 \times 10^{5}$ |
| $CH_2(OH)CHO$ | glycolaldehyde | 0.4273 |
| $CH_3CH_2(OOH)$ | ethyl hydrogen peroxide | $2.423 \times 10^{-3}$ |
| Organic compound | | 1 |
| compounds without phase transfer into the gas phase | | |
| NO | nitric oxide | 0.0429 |
| $NO_2$ | nitrogen dioxide | 0.179 |
| $NO_3$ | nitrate radical | $3.5 \times 10^{-4}$ |
| $HNO_4$ | Peroxynitric acid | $8.83 \times 10^{-3}$ |
| CO | carbon monoxide | 140.3 |
| $C_5H_8$ | isoprene | 1.031 |
| MACR | methacrolein | 0.282 |
| MVK | methyl vinyl ketone | 0.113 |
| $C_2H_6$ | ethane | 0.846 |
| $C_2H_4$ | ethene | 0.469 |
| $C_3H_6$ | propene | 0.118 |
| $H_2$ | hydrogen | 550 |
| $CH_3CO(OOH)$ | peracetic acid | 0.240 |

| | | |
|---|---|---|
| $C_4H_{10}$ | butane | 0.142 |
| $CH_3CH_2(OO)$ | ethylperoxy radical | $7.144\times10^{-6}$ |
| $PO_2$ | Other peroxy radicals ($C_2$, $C_3$) | $1.414\times10^{-4}$ |
| POOH | Hydroperoxides of $PO_2$ | $2.076\times10^{-3}$ |
| PAN | peroxy acetyl nitrate | 0.586 |
| Isop-OO | | $4.181\times10^{-3}$ |
| $CH_2=C(CH_3)CO(OO)$ | MCO3, peroxy radical from MACR * | $1.690\times10^{-4}$ |
| $CH_2=CHC(OO)(CH_3)CH_2(ONO_2)$ | peroxy radical from NO3+ISOP | $2.840\times10^{-5}$ |
| ONITR | Organic nitrates | 0.0921 |
| $CH_3OH$ | Methanol | 3.403 |
| $HOCH_2C(OOH)CH_3CHCH_2$ | ISOPOOH, peroxide from isoprene | 0.0318 |
| $XO_2$ | Additional peroxy radicals | $5.318\times10^{-4}$ |
| XOOH | Hydroperoxides of $XO_2$ | $8.885\times10^{-3}$ |
| $C_5H_8$ | terpene | 0.0407 |
| terpene-OO | a-pinene peroxy radical | $2.020\times10^{-4}$ |
| terpene-OOH | a-pinene peroxide | $2.499\times10^{-3}$ |
| MACR-OO | methacrolein peroxy radical | $1.678\times10^{-4}$ |
| MACR-OOH | methacrolein peroxide | $3.322\times10^{-3}$ |
| MPAN | methacryloyl peroxynitrate | 0.072 |
| $RO_2$ | Peroxy radicals from acetone | $3.614\times10^{-5}$ |
| ROOH | Peroxyde from $RO_2$ | $3.220\times10^{-3}$ |
| $CH_3COCH_3$ | acetone | 1.437 |
| $CH_4$ | methane | 1850 (constant) |
| Only in the aqueous phase (constant throughout the simulation) | | |
| WSOC (water soluble organic carbon) | | 0.626 |
| pH | | 4.5 |

---

## Author Response (AR2)

We thank the editor for her positive comments on our manuscript.

As we had uploaded our revised manuscript previously and the editor did not have any additional comments, we did not make any significant changes.

The only change was made in Table 3, its caption and in line 508:

Following up on technical correction suggested by the editorial office, the shading in the cells in Table 3 was removed. These values are now by asterisks (*). Accordingly, we changed the text in line 508 and in the caption of Table 3.